



# Regularization and L-curves in ice sheet inverse models, a case study in the Filchner-Ronne catchment

Michael Wolovick[1], Angelika Humbert[1,2], Thomas Kleiner[1], and Martin Rückamp[3]

[1]Glaciology Section, Alfred-Wegener-Institut Helmholtz-Zentrum für Polar- und Meeresforschung, Bremerhaven, Germany
[2]University of Bremen, Department of Geosciences, Bremen, Germany
[3]Bavarian Academy of Sciences and Humanities, Munich, Germany

**Correspondence:** Michael Wolovick (michael.wolovick@awi.de)

**Abstract.** Over the past three decades, inversions for ice sheet basal drag have become commonplace in glaciological modeling. Such inversions require regularization to prevent over—fitting and ensure that the structure they recover is a robust inference from the observations, confidence which is required if they are to be used to draw conclusions about processes and properties of the ice base. While L-curve analysis can be used to select the optimal regularization level, the treatment of L-curve analysis in glaciological inverse modeling has been highly variable. Building on the history of glaciological inverse modeling, we demonstrate general best practices for regularizing glaciological inverse problems, using a domain in the Filchner-Ronne catchment of Antarctica as our test bed. We show a step by step approach to cost function normalization and L-curve analysis. We show that the optimal regularization level converges towards a finite non-zero limit in the continuous problem, associated with a best knowable basal drag field. We find that, when inversion results are judged by a metric that accounts for both the variance of the result and the quality of the fit, then they support nonlinear as opposed to linear sliding laws. We also find that geometry—based approximations for effective pressure degrade inversion performance, but that an actual hydrology model may marginally improve performance in some cases. Our results with 3D inversions suggest that the additional model complexity is not justified by the 2D nature of the velocity data, but we are hopeful that inversions of 3D models may be better situated to take advantage of new constraints in the future. We conclude with recommendations for best practices in glaciological inversions moving forward.

## 1 Introduction

> With four parameters I can fit an elephant, and with five I can make him wiggle his trunk.
>
> -John von Neumann

Ever since the pioneering "tutorial" of MacAyeal (1993), inversions for ice sheet basal drag- and, less commonly, englacial rheology- have been an important part of the ice sheet modeler's toolbox. In these inversions, a numerical model of ice sheet



flow is compared to observations of ice surface velocity using a cost function, and the variable to be constrained (a field of either drag or rheology) is iteratively adjusted until the cost function has dropped to an acceptable level. In the three decades since MacAyeal's original paper, remotely sensed observations of ice sheet surface velocity have become available for nearly the entirety of the Greenland and Antarctic ice sheets, along with many smaller ice caps and mountain glaciers (Joughin et al., 2010; Rignot et al., 2011, 2017a; Fahnestock et al., 2016), while knowledge of ice sheet geometry has improved dramatically as well (Lythe and Vaughan, 2001; Fretwell et al., 2013; Morlighem et al., 2014, 2020; Bamber et al., 2001, 2009, 2013). Concurrently, the computing power available to run these models has increased enormously. Thus, it has now become routine in glaciological modeling to use inversions to study the spatial- and, with repeat observations of surface velocity, temporal-variability of the ice sheet bed and to initialize transient models of the ice sheet future (Joughin et al., 2004, 2006, 2009; Morlighem et al., 2010, 2013; Gillet-Chaulet et al., 2012, 2016; Habermann et al., 2013; Sergienko and Hindmarsh, 2013; Sergienko et al., 2014; Shapero et al., 2016).

However, the question remains of just how much information can truly be gleamed from ice model inversions. Fundamentally, the trouble is that, in the continuous problem, a spatially resolved inversion has an infinite number of free parameters. In a numerical implementation of the problem, the number of free parameters is equal to the number of 2D (map view) grid nodes. Furthermore, the transfer function relating variations in basal shear stress to surface velocity or slope is essentially a low-pass filter (Gudmundsson, 2003), allowing dramatically different fine-scale drag fields to produce similar velocity observations (Habermann et al., 2012), and ensuring that the inverse problem (using surface velocity to infer basal stress) is ill-posed. Ill-posed inversions require regularization to create a well-posed problem and avoid overfitting noise in the data (Tikhonov and Arsenin, 1977). Regularization involves adding an additional term to the cost function that penalizes variability in the inversion target field, usually through a measure of gradient or curvature, and combining this regularization cost ($J_{\mathrm{reg}}$) with the original data cost term ($J_{\mathrm{obs}}$) to create a combined cost function ($J$) that is minimized by the inversion. A weighting parameter, which we write as $\lambda$, controls the relative contribution of the data and regularization terms to the overall cost function that is minimized in the inversion.

Yet this introduces a new free parameter into the problem: if $\lambda$ is too high, the inversion will under-fit the data, producing very little structure even if the observations could be used to infer greater detail, while if $\lambda$ is too low, the inversion will over-fit the data, producing spurious structure that is not truly required by the observations. One way to determine the optimal value of $\lambda$ is through an L-curve analysis, in which $J_{\mathrm{obs}}$ and $J_{\mathrm{reg}}$ are plotted against one another for different values of $\lambda$, ideally showing a corner (the "L" shape) at the value of $\lambda$ that best balances between the extremes of over-fitting and under-fitting (Hansen, 1992; Hansen and O'Leary, 1993). This corner can be rigorously identified by measuring the curvature of the L-curve in log-log space (Hansen and O'Leary, 1993). However, L-curve analysis is not perfect. It can be non-convergent for problems whose solutions are sufficiently "rough": specifically, L-curve analyses are non-convergent if the singular values of the operator decay more rapidly at short wavelengths than the Fourier coefficients of the solution (Vogel, 1996). It is not clear whether or not that condition applies to glaciological inversions; more fundamentally, it is also not clear whether the definition of "convergence" used by Vogel (1996) is of practical use in glaciological modeling.



In this paper, we first give a brief review of the ways in which regularization and L-curve analysis have been treated in glaciological inverse models, and then we give a tutorial on how to regularize and perform L-curve analyses for glaciological inverse problems. Using an inversion of the Filchner-Ronne catchment in Antarctica as our test bed, we demonstrate how to scale misfit components, compute L-curve curvature, and select both the optimal $\lambda$ value as well as a range of acceptable $\lambda$ values bracketing the corner. We then show practical convergence of the L-curve analysis with respect to mesh resolution, and we test for the effects of both sliding nonlinearity and effective pressure dependence within the sliding law. We use a comparative L-curve analysis to evaluate the relative performance of a 2D vertically integrated model as compared to a 3D model. Finally, we combine our various model results into a single best weighted-average estimate of the basal drag within our domain. We conclude with a discussion about what this approach to glaciological inverse problems enables us to learn about basal sliding and with recommended best practices for future modelers.

## 2 Review

The original work of MacAyeal (1993) was a pioneering addition to the toolbox of glaciological modeling at the time. However, that work included no explicit regularization at all (MacAyeal, 1993, Eq. 17), instead avoiding overfitting by parameterizing the drag coefficient as a sum of Fourier basis functions and truncating the sum at low order. Indeed, in the synthetic example used in that paper, the "true" basal drag was given by only two Fourier terms, and the inversion consisted of an optimization to recover those same two coefficients. Subsequent papers using the MacAyeal method with real velocity data instead of synthetic examples obviously included a much greater number of Fourier coefficients, but they still relied on truncation of the series to prevent over-fitting rather than including explicit regularization within the cost function (Joughin et al., 2004, 2006, 2009). The question of where exactly to truncate this series was not investigated in detail other than to note that the governing equations were not accurate below wavelengths of a few ice thicknesses.

Later, Arthern and Gudmundsson (2010) introduced the method of solving the inverse problem by simultaneously solving Neumann and Dirichlet versions of the stress balance problem (ie, solving one forward problem where the surface is traction-free and another forward problem where the surface has a Dirichlet condition given by the observed velocity, and minimizing the difference between these two solutions). This method produced a major advance in ice sheet inversions, allowing the Full Stokes (FS) equations to be inverted without the need to derive the adjoint. In addition, and unlike the models in the direct MacAyeal lineage (Joughin et al., 2004, 2006, 2009), they did not rely on a series expansion representation of the drag coefficient in their derivation. However, they did not put any explicit regularization in their cost function either (Arthern and Gudmundsson, 2010, Eq. 3) and, consequently, they found that their algorithm produced erroneous spatial structure when given noisy velocity data (Arthern and Gudmundsson, 2010, Fig. 3). Rather than introduce regularization, they settled on a "heuristic" method of stopping their algorithm at a small number of iterations before the erroneous spatial structure had a chance to grow. A similar strategy was employed by Habermann et al. (2012), who developed a method for achieving implicit regularization through selection of iteration stopping criteria in synthetic inversions. However, when those authors later applied their methods



to the inversion of real data, they employed explicit regularization (Habermann et al., 2013, Eq. 4), as did later users of the Neumann-Dirichlet approach (e.g. Shapero et al., 2016).

Indeed, it is rare to find inversions of real data and real glacier geometries performed without either explicit regularization or series truncation. However, that regularization is unfortunately not always discussed in detail, and L-curve analyses are not always performed, steps which we suggest ought to become a standard for establishing the validity and reliability of inversion results. For instance, Morlighem et al. (2010) studied spatial patterns in basal drag under Pine Island Glacier determined from inverse models, yet they gave no details on regularization other than a single sentence (in their Section 2.4) confirming that it

was in fact present. To be fair, the same group of authors later published a paper using the same model for all of Antarctica in which they discussed regularization in detail and presented an L-curve (Morlighem et al., 2013, Fig. 1). However, it is not clear whether the regularization and L-curve discussed in the later paper also applies to the earlier paper; certainly, it is not possible to evaluate based on the results presented in the first paper alone whether or not conclusions drawn about the detailed structure of basal drag near the grounding line may have been influenced by the choice of regularization.

In other papers, regularization is performed and explained in detail, but L-curve analysis is not performed. For instance, Sergienko and Hindmarsh (2013) and Sergienko et al. (2014) used inverse models to infer the presence of short-wavelength rib-like spatial structure in both Antarctica and Greenland, and then used those inversion results as evidence for a specific till-water instability mechanism at the ice base. They explicitly showed how they regularized their inversion, and they addressed the question of over- or under-fitting by analyzing inversions with several values of $\lambda$ for both real and synthetic examples.

However, they explicitly chose not to present an L-curve analysis, citing the possibility of L-curve non-convergence (Vogel, 1996) as their reason. While these authors went to considerable lengths to demonstrate that the rib-like pattern they identified was robust to regularization, the larger question they raised about the validity of L-curve analysis in glaciological inverse problems went unanswered.

When an L-curve analysis is performed in glaciological inversions, the presentation of the L-curve can be highly variable.

Hansen and O'Leary (1993) explicitly recommended that L-curves should be presented on log-log axes, and that the curvature computation used to select the corner should also be performed in log-log space. However, while glaciologists have sometimes presented L-curves on log-log axes (e.g. Morlighem et al., 2013, Fig. 1), they have also employed linear-linear axes (Shapero et al. (2016, Fig. 1); Habermann et al. (2013, Fig. 5)) or even mixed log-linear axes (Gillet-Chaulet et al., 2012, Fig. 3). The variable presentation styles for different L-curves is potentially problematic, as L-curve analysis is explicitly predicated on the

ability to perceive and identify shape. Furthermore, viewing an L-curve on a linear scale can potentially be misleading: as an example, consider the case in which the two cost components very inversely with one another, $J_{\mathrm{obs}} \propto J_{\mathrm{reg}}^{-1}$. On linear/linear axes, this plot would look strongly "L"-shaped, with a distinct corner. Yet on log/log axes, it is revealed to be a straight line: a factor of 2 reduction in $J_{\mathrm{obs}}$ can always be purchased for a factor of 2 increase in $J_{\mathrm{reg}}$, and vice versa, and we have no basis for deciding that any one combination is a better tradeoff than any other. Finally, while L-curves in the glaciological literature

have usually been monotonic (Morlighem et al., 2013; Habermann et al., 2013; Shapero et al., 2016), that has not always been the case (Gillet-Chaulet et al., 2012).



We do not claim that this review of regularization and L-curve analysis in the glaciological literature is exhaustive. Our intention here was merely to illustrate the spectrum of ways in which these issues have been treated within glaciological inversions. Inversions are a vital source of boundary conditions for projections of ice sheet dynamics in a changing climate (Seroussi et al., 2019), and inversions also provide a vital window into the conditions and processes at the ice sheet bed, an environment that is difficult to observe directly. L-curve analysis is an important method for ensuring that inversions are well-calibrated between under-fitting and over-fitting the data, yet the treatment of L-curve analysis and the visual presentation of L-curves has been highly variable within the glaciological literature. Even the basic question of whether glaciological inverse problems have a convergent L-curve remains unanswered.

## 3 Methods

### 3.1 Setting

We model a domain covering the catchments of West and East Antarctica that feed into the Filchner-Ronne Ice Shelf (Fig. 1). This domain has a wide variety of glaciological settings, including both slow-flowing regions and fast-flowing ice streams, mountainous subglacial topography and deep subglacial basins, topographically confined outlet glaciers and relatively uncon-fined glaciers, along with a large floating ice shelf containing several ice rises and rumples. We take surface (Fig. 1a) and bed (Fig. 1b) elevations, along with ice thickness and our ocean/shelf/sheet/rock mask from Morlighem et al. (2020). Ice surface velocity observations needed for our inversion (Fig. 1c) are taken from a combined Landsat 8 and TerraSAR-X product created by Hofstede et al. (2021), merged with MEaSUREsv2 (Rignot et al., 2017a). We also can use the observed velocity and geometry to compute the driving stress resolved in the flow direction (Fig. 1d) which we use to estimate potential energy dissipation (Sections 3.4 and 3.3) and to compute our first guess of drag coefficient (Section 3.6).

### 3.2 Ice Sheet Model

We use the Ice-Sheet and Sea-level System Model (ISSM; Larour et al. (2012)) to simulate the catchment of the Filchner-Ronne Ice Shelf. We use the Shelfy-Stream Approximation (SSA, also called Shallow Shelf Approximation; MacAyeal (1989)). SSA reduces the 3D problem of ice flow to a vertically integrated 2D problem in the map-view plane, thus neglecting all but the xx, yy, and xy terms in the stress and strain-rate tensors. SSA is most valid in fast-flowing ice streams and outlet glaciers, where the ice is flowing primarily by basal sliding. The SSA equations are also self-adjoint for linear sliding laws (MacAyeal, 1993), and they were the first version of the ice sheet stress balance equations to be used in an inversion (MacAyeal, 1993). The SSA equations are,

$$\frac{1}{2}\nabla \cdot (H\bar{\mu}(\nabla \boldsymbol{u} + \nabla^{\mathrm{T}} \boldsymbol{u})) = \boldsymbol{\tau}_{\mathrm{d}} - \boldsymbol{\tau}_{\mathrm{b}}, \tag{1}$$

where $H$ is ice thickness, $\bar{\mu}$ is the vertically averaged viscosity, $\boldsymbol{u}$ is the ice velocity vector, $\nabla()$ is the gradient operator and $\nabla^{\mathrm{T}}()$ is its transpose, $\nabla \cdot ()$ is the divergence operator, $\boldsymbol{\tau}_{\mathrm{d}}$ is the gravitational driving stress, and $\boldsymbol{\tau}_{\mathrm{b}}$ is the basal stress,



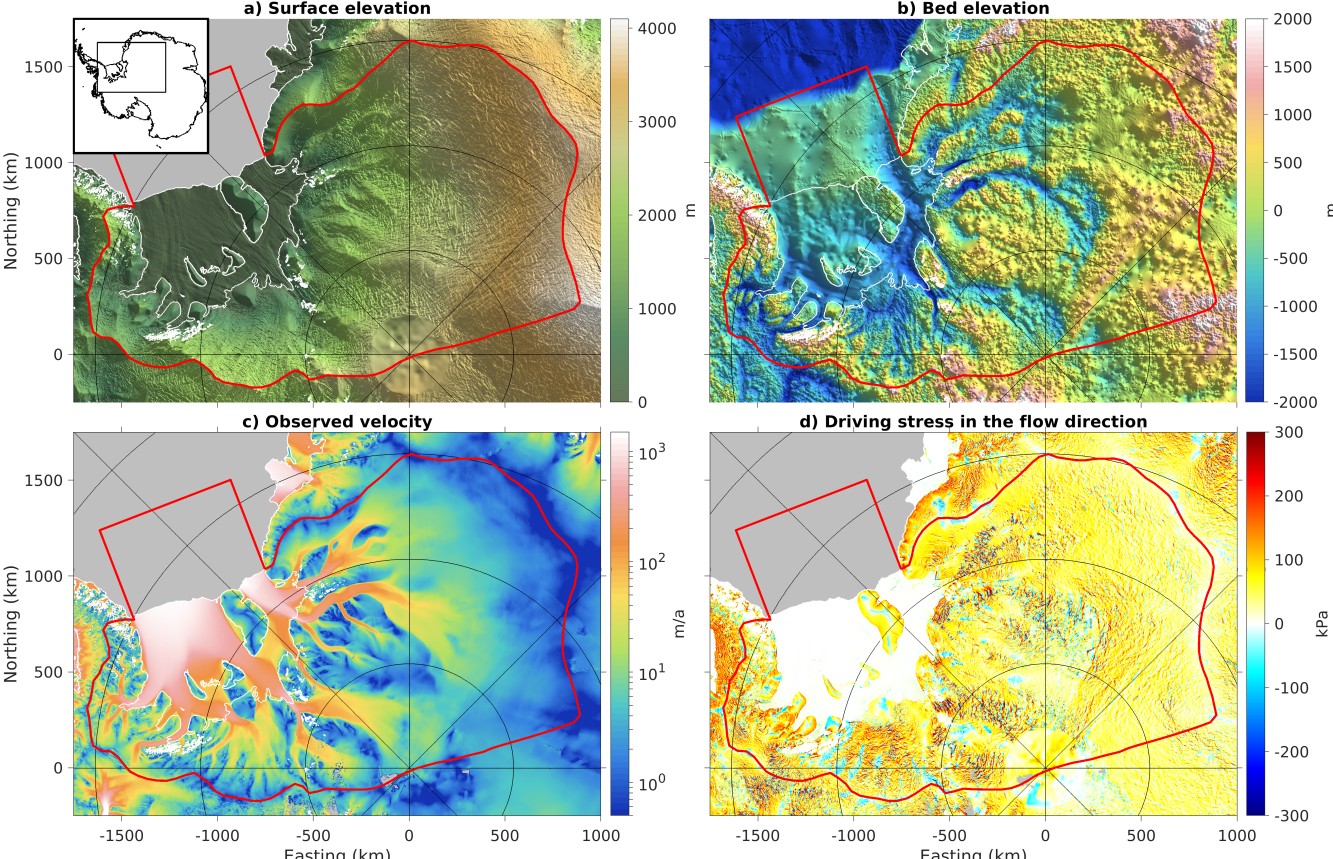

**Figure 1.** Geographic setting of our study area. a) Surface elevation from Morlighem et al. (2020), with inset showing map location in Antarctica. b) Bed elevation from the same source. c) Surface velocity observations from Hofstede et al. (2021) and Rignot et al. (2017b). d) Driving stress in the flow direction computed from ice geometry and observed velocity vectors. Red line in all maps shows outline of model domain. The domain extends beyond the calving front to facilitate transient modeling of ice front motion, which we do not discuss in this paper. White lines show ice edge and grounding line. Graticule uses lines at $5°$ in latitude and $45°$ in longitude. Plots (a) and (b) employ hillshading to emphasize structure and texture, using two perpendicular light sources from grid north and grid east.

which we infer indirectly using our inversion. The driving stress is computed from the ice sheet geometry by, $\boldsymbol{\tau}_\mathrm{d} = -\rho_\mathrm{i} g H \nabla S$, where $\rho_i$ is the density of ice, $g$ is the acceleration due to gravity, and $S$ is the ice sheet surface elevation. At the inland lateral boundaries of the domain we use Dirichlet conditions given by the observed velocity. The effective viscosity is computed from

155 the nonlinear rheology of ice (Glen, 1953) using,

$$\bar{\mu} = \bar{B} \dot{\epsilon}_0^{\frac{1-n}{n}}, \tag{2}$$




where $\bar{B}$ is the vertically averaged rheological stiffness parameter, $\dot{\epsilon}_0$ is the effective strain rate, and $n = 3$ is the rheological exponent. We use the SSA equations for the majority of our experiments, but we also ran several experiments using the 3D Higher Order (HO) equations (Blatter, 1995; Pattyn, 2002) to compare the effect of using a 3D model instead of a 2D one. We do not list those equations here for brevity. We compute the rheological stiffness needed for both sets of equations using a 1D thermal model, described next.

## 3.3 Thermal Structure and Rheology

Our SSA model needs an estimate of the column-average ice rheology $\bar{B}$, our HO models needs an estimate of vertically resolved $B$, and the Confined-Unconfined Aquifer System (CUAS-MPI) hydrology model (Beyer et al., 2018; Fischler et al., 2023) also needs an estimate of basal melt rate. We provided all of these using a 1D vertical steady-state advection-diffusion thermal model. We show the boundary conditions used to force the model in Figure 2a-c. For surface temperature, we use Comiso (2000), and for accumulation rate, we use the mean of Arthern et al. (2006) and Van de Berg et al. (2005). All surface climate inputs were accessed through Le Brocq et al. (2010). We found that early iterations of our model were too warm in the floating ice shelf, resulting in insufficient buttressing and ice flow that was uniformly too fast in the shelf. This likely resulted from the neglect of horizontal advection in the 1D thermal model, thus missing the advection of cold ice into the shelf from higher elevations inland. As an ad hoc fix we reduced surface temperatures by $10°C$ at low elevations (below $\sim 200\,m$), resulting in a mix of positive and negative velocity misfits in the shelf. We do not include velocities from the floating shelf in our cost function, nor did we attempt to optimize ice shelf rheology; the $-10°C$ shift was chosen because it is a round number, and we did not attempt to tune the shift further. The basal heat flux used to force the model (Fig. 2c) is the sum of geothermal heat flow and an estimate of shear heating: we use Martos et al. (2017) for geothermal heat flow, and we estimate shear heating by taking the dot product of driving stress and observed ice velocity (giving us the rate of dissipation of gravitational potential energy) and then smoothing at $10\,km$ wavelength.

The resulting thermal structure is shown in Figure 2d–f. Basal temperature is at the pressure-dependent melting point in most of the domain, including almost all of the fast-flowing regions. Basal melt rate varies between about $0.1\,mm\,a^{-1}$ and $1\,m\,a^{-1}$, with values above $1\,cm\,a^{-1}$ being exclusively due to high rates of shear heating. We use the thermal model to compute a vertically variable rheology stiffness parameter $B$ based on Cuffey and Paterson (2010, p. 75), which we then average vertically to produce the $\bar{B}$ parameter needed by our model (Fig. 2f). $\bar{B}$ only varies by about a factor of 4 between its largest and smallest values, with most of the spatial structure attributable to gradients in surface temperature (Fig. 2, c.f. plots a and f).

## 3.4 Numerical Mesh

For our numerical mesh, we use an unstructured triangular mesh in the map-view plane, constructed using the Bidimensional Anisotropic Mesh Generator (BAMG, Hecht, 2006). As is common in finite element glaciological modeling, we want to refine our mesh in dynamically important regions, such as fast-flowing ice streams and outlet glaciers, or the grounding line and calving front. However, we also wish to investigate the influence of mesh resolution on our inversion results, which means that we need to create multiple meshes with different resolution but a similar spatial pattern of refinement. We do this by



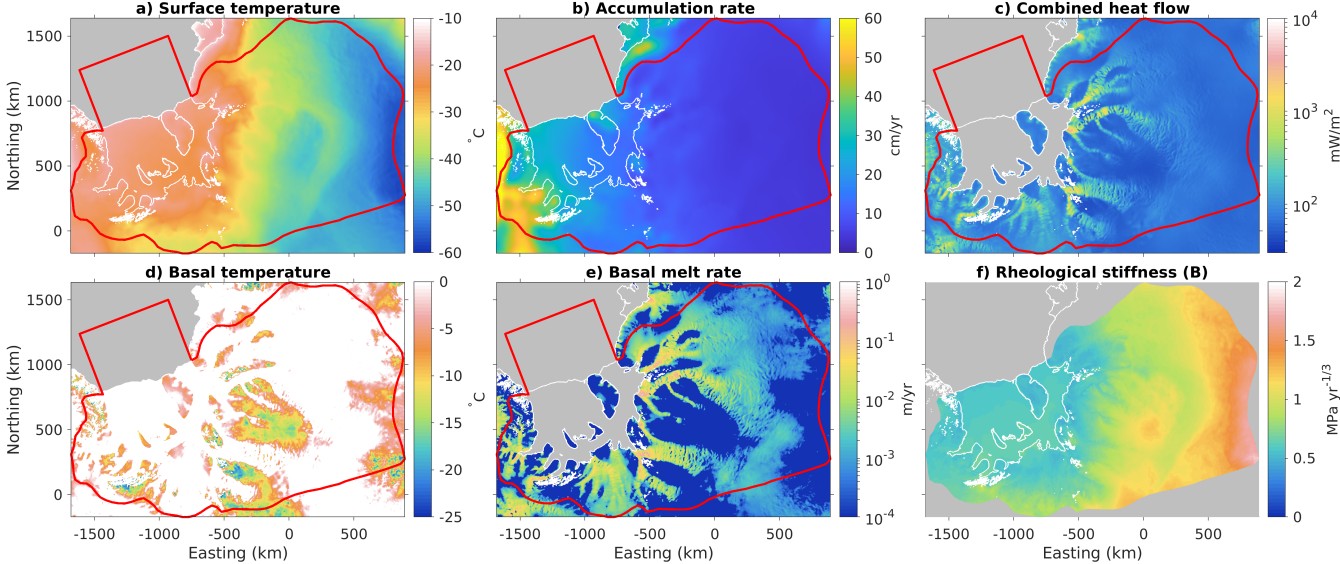

**Figure 2.** Thermal forcing and ice rheology. Top row (a–c) shows the boundary conditions (surface temperature, surface accumulation rate, and basal heat flow) used to force the 1D thermal model. Bottom row (d–f) shows model output: basal temperature, basal melt rate, and column-average rheological stiffness parameter. Basal temperature has been corrected for the pressure dependence of the melting point.

constructing a common control field for all of our meshes, and then systematically varying parameters that control how the mesh is constructed from that control field.

We construct a control field on a 1 km regular grid using the logarithm of potential energy dissipation (the dot product of driving stress and ice velocity) in the grounded domain, the logarithm of strain-rate magnitude in the floating domain, and adding a constant multiple of the mask in order to produce discontinuities at the grounding line and calving front (Fig. 3a).

We use BAMG to iteratively refine the mesh so as to minimize interpolation errors in the control field; thus, the mesh will be refined in areas where the control field has sharp curvature and the mesh will be unrefined in areas where the control field is smooth. We generate ten meshes using values of *err*, the acceptable level of interpolation error, ranging linearly from 0.01 to 0.1; values of *hmin*, the minimum edge length, ranging from 0.1 km to 1.0 km; and values of *hmax*, the maximum edge length, ranging from 10 km to 100 km.

The resulting meshes are summarized in Fig. 3. We use the hydraulic diameter (4 x area/perimeter) to be a representative linear size metric for our anisotropic elements, with the factor of 4 chosen such that the hydraulic diameter of a circle would be equal to the common diameter. All of our meshes share a common spatial pattern of refinement (Fig. 3d-f) that they inherited from the control field (Fig. 3a). We analyze fast-flowing and slow-flowing regions separately, using a smooth weighting function with a nominal 50% cutoff at $15\,\mathrm{m\,a^{-1}}$ and $0\% - 100\%$ limits at $7.5\,\mathrm{m\,a^{-1}}$ and $30\,\mathrm{m\,a^{-1}}$. We use a cosine taper applied to

the logarithm of velocity to generate the weighting function. These velocity thresholds are relatively low, but we choose them as approximate values roughly separating both streaming flow from slow flow, and also separating reliable velocity data from



unreliable data. Mesh size in both the fast- and slow-flowing regions varies smoothly with mesh number, but mesh size in the fast-flow region is consistently $\sim 7$ times smaller than in the slow-flow region, and the model places about $2/3$ of the grounded elements within the fast-flowing region (Fig. 3b), despite the fact that the fast-flowing region is only about $1/5$ of the grounded
area. An analysis of histograms of mesh size in the fast-flowing region reveals that our meshing procedure produces similar statistical distributions of element size for each mesh, simply shifted towards larger or smaller size (Fig. 3c).

Once the meshes were generated, we interpolated all relevant grids (e.g., geometry, velocity) onto each mesh using a multi-grid technique. In this technique, we smoothed each grid at multiple wavelengths, and the value for each mesh element was interpolated from the grid with a smoothing wavelength appropriate for that element's size. This technique is superior to
simple interpolation from a single grid because it prevents aliasing of short-wavelength content into spurious long-wavelength content in regions where the mesh is coarse, while simultaneously preserving short-wavelength content in regions where the mesh is fine. When it came to interpolating the mask, rather than employing nearest neighbor interpolation, we reordered the mask into a sequence that behaves logically under averaging and interpolation operations (0=open ocean, 1=ice shelf, 2=ice sheet, 3=exposed rock) and performed the same multi-wavelength smoothing procedure. Finally, we removed areas with an ice
thickness less than $10\,\mathrm{m}$ from our mesh by setting those areas to either exposed rock or open ocean.

## 3.5  Sliding Law

For our basic model, we use a Weertman-type sliding law (Weertman, 1957),

$$\boldsymbol{\tau}_{\mathrm{b}} = C|\boldsymbol{u}_{\mathrm{b}}|^{\frac{1-m}{m}}\boldsymbol{u}_b, \tag{3}$$

where $\boldsymbol{\tau}_{\mathbf{b}}$ is the basal shear stress vector, $C$ is a spatially variable drag coefficient, $\boldsymbol{u}_{\mathrm{b}}$ is the basal sliding velocity vector,
and $m$ is the slip exponent. For most of our experiments we used linear Weertman ($m = 1$), however, we also ran tests with nonlinear Weertman sliding using $m = 3$ and $m = 5$, the results of which we present in Section 4.6. When testing the effect of sliding laws with effective pressure, we use a Budd-type sliding law (Budd et al., 1979),

$$\boldsymbol{\tau}_{\mathrm{b}} = CN|\boldsymbol{u}_{\mathrm{b}}|^{\frac{1-m}{m}}\boldsymbol{u}_b \tag{4}$$

where $N = P_{\mathrm{i}} - P_{\mathrm{w}}$ is the effective pressure at the ice base, and $P_{\mathrm{i}}$ and $P_{\mathrm{w}}$ are ice and water pressures, respectively. We test
three different sources for $N$:

$$
\begin{aligned}
N_{\mathrm{op}} &= \rho_{\mathrm{i}}gH - \rho_{\mathrm{sw}}g(-B), & (5)\\
N_{\mathrm{opc}} &= \rho_{\mathrm{i}}gH - \max(0, \rho_{\mathrm{sw}}g(-B)), & (6)\\
N_{\mathrm{CUAS}} &= N_{\mathrm{CUAS}}(x, y), & (7)
\end{aligned}
$$

where $N_{\mathrm{op}}$ is $N$ assuming a perfect hydraulic connection to the ocean, and allowing negative water pressures; $N_{\mathrm{opc}}$ is $N$
determined from ocean pressure with a strict cutoff at $B = 0$; and $N_{\mathrm{CUAS}}$ is $N$ taken from the hydrology model CUAS-MPI.



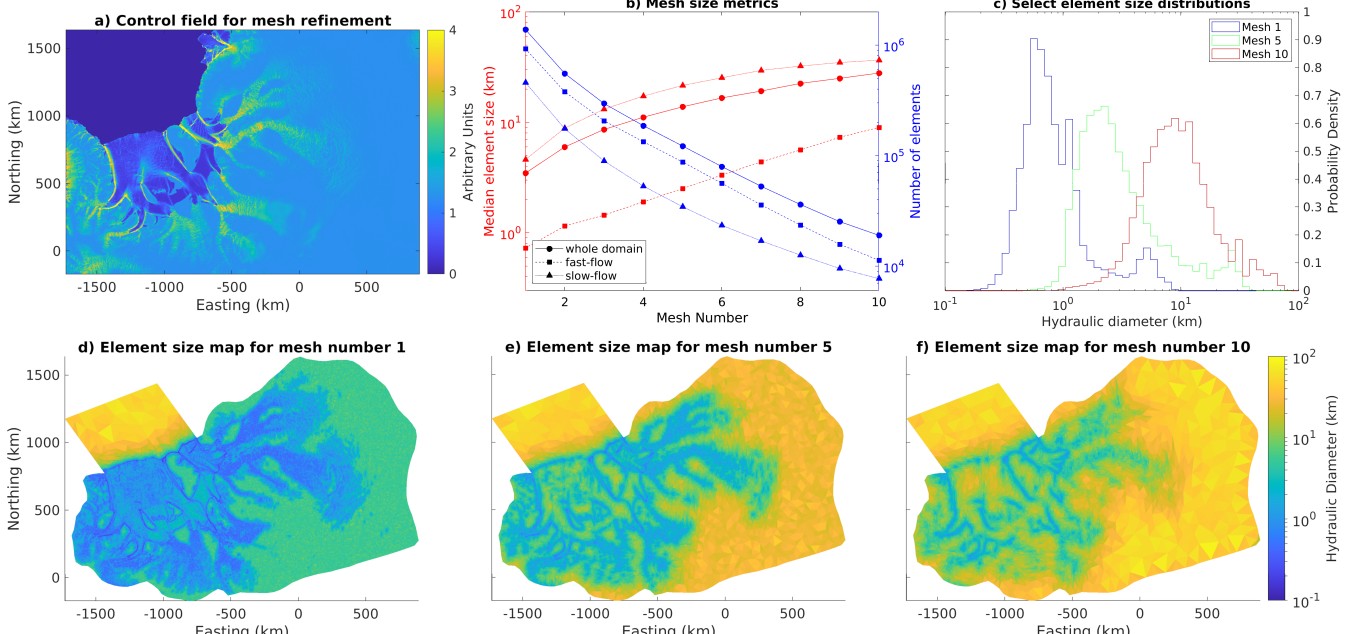

**Figure 3.** Mesh size analysis for our series of meshes. a) The common control field used to refine all of the meshes. b) Analysis of median element size and number of elements for each mesh in the series. Separate plots are shown for the entire grounded domain, as well as the grounded domain split into fast-flowing and slow-flowing regions. We define "fast-flow" as $\gtrsim 15\,\mathrm{m\,a^{-1}}$, with a smooth weighting function between $7.5\,\mathrm{m\,a^{-1}}$ and $30\,\mathrm{m\,a^{-1}}$. c) Selected histograms of element size within the grounded fast-flow domain. d–f) Maps of element size for the same selected meshes. Note that median size in (b) and probability density in (c) are defined with respect to domain area, not element number; ie, a median size of $X$ means that $50\%$ of the domain area is covered by elements smaller than $X$, not that $50\%$ of the elements themselves are smaller than $X$.

CUAS-MPI is an equivalent thin-layer hydrology model using confined-unconfined aquifer equations and parameterizations for efficient drainage and the opening of cavity space by ice sliding (Beyer et al., 2018; Fischler et al., 2023). We force CUAS-MPI using the melt rate estimated by the same 1D thermal model used to estimate ice rheology (Fig. 2e). We run CUAS-MPI on a 1 km regular grid and then interpolate the results onto our ISSM mesh. $N_{\mathrm{opc}}$ is a commonly used approximation in the glaciological literature, and we felt it was important to test it here. However, $N_{\mathrm{opc}}$ contains a sharp discontinuity in gradient around the $B = 0$ contour, so our motivation for allowing negative water pressures in $N_{\mathrm{op}}$ is to eliminate that discontinuity and test for the optimal regularization when $N$ was a simple linear function of $B$. In both approximations $N$ is strictly positive throughout our domain.

We show a comparison of the three $N$ fields in Figure 4. $N_{\mathrm{op}}$ is the smoothest, as it is a simple linear function of ice thickness and bed elevation. It also has the most range, as inland regions with a very high ice surface elevation and bed elevation above zero have a large ice overburden pressure combined with a negative water pressure, producing very high $N$. $N_{\mathrm{opc}}$ has a broadly similar large-scale structure to $N_{\mathrm{op}}$, but with more visible short-wavelength structure introduced by the cutoff in water pressure





at $B = 0$. The range of $N_{\mathrm{opc}}$ is also less than the range of $N_{\mathrm{op}}$, as the high-elevation inland regions are no longer allowed to have negative water pressure and thus have lower $N$. $N_{\mathrm{CUAS}}$ has the lowest range of the three, with a maximum of about 10 MPa, or a factor of 3–4 less than the other two. The spatial structure of $N_{\mathrm{CUAS}}$ is more complex than $N_{\mathrm{op}}$ but also appears more ice dynamically reasonable, at least under visual inspection, than $N_{\mathrm{opc}}$. The spatial structure of $N_{\mathrm{CUAS}}$ is a close visual match to the overall structure of the ice-sheet velocity field (Fig. 4c, c.f. Fig. 1c), likely as a result of the fact that we used estimated shear heating to produce the melt rate forcing for CUAS-MPI (Fig 2e).

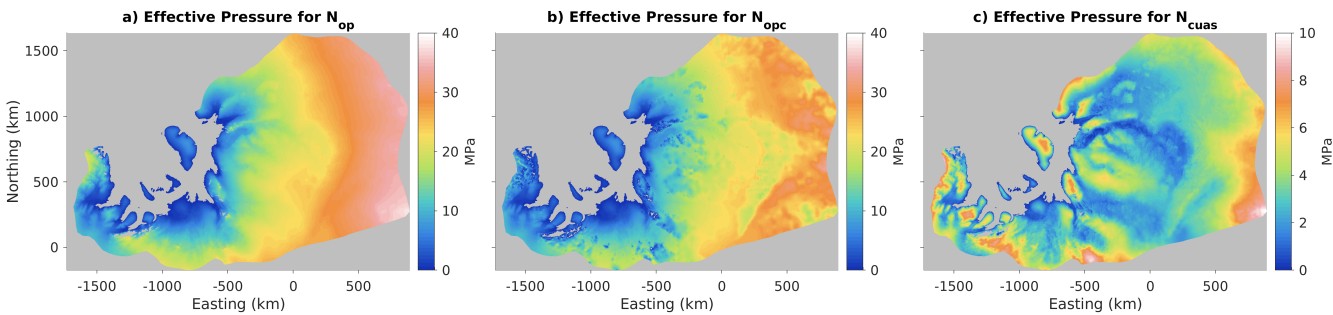

**Figure 4.** Comparative maps of effective pressure $N$. a) $N_{\mathrm{op}}$, or $N$ computed assuming a perfect connection to the ocean and allowing negative water pressures. b) $N_{\mathrm{opc}}$, as $N_{\mathrm{op}}$ but without allowing negative water pressures. c) $N$ computed by the hydrology model CUAS. Note change in color scale in panel (c).

### 3.6 Cost Function

We use a single observational misfit and a single regularization term in our cost function. Our observational misfit is the absolute velocity misfit,

$$J_{\mathrm{obs,raw}} = \int_{\Omega} \frac{1}{2} \left( (u_x - u_{x,obs})^2 + (u_y - u_{y,obs})^2 \right) d\Omega, \tag{8}$$

where $J_{\mathrm{obs,raw}}$ is the raw observational component of the cost function, $\Omega$ is the map-view model domain, and $\boldsymbol{u} = (u_x, u_y)$ are the two components of the horizontal velocity vector. Our regularization misfit penalizes gradients in the friction coefficient,

$$J_{\mathrm{reg,raw}} = \int_{\Omega} \frac{1}{2} |\nabla k|^2 d\Omega, \tag{9}$$

where $J_{\mathrm{reg,raw}}$ is the raw regularization cost and $k = \sqrt{C}$ is the internal ISSM friction coefficient. There is no mathematical or physical reason why the leading constant of the sliding law should be squared; MacAyeal (1993) squared the constant to ensure it remained positive, but this same goal can be achieved by simply setting a lower limit within the inversion code. We therefore exclusively analyze structure in $C$, not $k$, when analyzing our inversion results. However, we do expect $C$ to vary by many orders of magnitude, and the regularization term may therefore under-penalize structure in low-drag areas as compared to high-drag areas. Taking the square root of the drag coefficient before computing the regularization term halves the dynamic range, and thus we expect this problem to be less of an issue for $k$ than for $C$.



Before combining the data and regularization cost terms, it is useful to scale them according to estimates of their characteristic magnitude. Scaling the cost function terms is valuable for two reasons: 1) it allows us to easily identify the region of

parameter space that we need to search in our L-curve analysis, and, 2) it ensures that the regularization parameter is both unitless and readily interpretable in terms of relative weight placed on the two component terms. Without scaling, the regularization parameter would have obscure units, it would be difficult to identify the region of parameter space near the corner of the L-curve to search for the optimal regularization, and there would be no basis for judging whether a given level of regularization was "large" or "small".

The characteristic scale of the data term is readily given by the variance of the velocity observations themselves,

$$S_{\text{obs}} = \int_{\Omega} |\boldsymbol{u}_{\text{obs}}|^2 d\Omega = A\sigma_{\text{obs}}^2, \tag{10}$$

where $S_{\text{obs}}$ is the characteristic scale of $J_{\text{obs,raw}}$, $|\boldsymbol{u}_{\text{obs}}| = \sqrt{u_{\text{x,obs}}^2 + u_{\text{y,obs}}^2}$ is the magnitude of the observed velocity vector, $A$ is the area of the grounded domain, and $\sigma_{\text{obs}}$ is the root-mean-squared (RMS) variability of the observed velocity magnitude.

Computing the characteristic scale of the regularization term is slightly more complex than computing the characteristic

scale of the data term. For the regularization term, we need a first guess of the drag coefficient, and we need that guess to have roughly the same magnitude and range of variability as the final drag coefficient. We get this guess by taking the ratio of driving stress and observed velocity,

$$k_{\text{guess}}^2 = \frac{\max(0, \tau_{\text{d,flow}})}{N|\boldsymbol{u}_{\text{d}}|^{1/m}}, \tag{11}$$

where $k_{\text{guess}}$ is the guess of $k$, and $\tau_{\text{d,flow}} = -\rho_{\text{i}} g H \nabla(S) \cdot \boldsymbol{u}_d/|\boldsymbol{u}_d|$ is the driving stress resolved in the flow direction (Fig. 1d).

For Weertman sliding, we set $N = 1$ in the above equation. For Budd sliding, we use a minimum $N$ value of $100\,\text{Pa}$ to prevent division by zero, and for both sliding laws we use a minimum velocity value of $10\,\text{cm\,a}^{-1}$ for the same purpose. Note that, while the above expression assumes that driving stress and basal drag are locally balanced, we do not use the gradient of $k_{\text{guess}}$ to estimate the scale of the regularization term directly. Instead, we use $k_{\text{guess}}$ to estimate the scale of variability in the true drag coefficient. We compute $\sigma_k$, the standard deviation of $k_{\text{guess}}$, and we then assume a sinusoidal perturbation in drag coefficient

with that amplitude and a wavelength given by the mean ice thickness within the domain, $k = \sigma_k \sin(2\pi x/\bar{H})$. The gradient of this sinusoid is, $\nabla(k) = \frac{2\pi\sigma_k}{\bar{H}} \cos(2\pi x/\bar{H})$, and the mean-square amplitude of the gradient is $2(\pi\sigma_k/\bar{H})^2$. Plugging the mean-square amplitude of this perturbation into Eq. 9 and assuming an area of integration equal to the grounded domain yields the scale estimate,

$$S_{\text{reg}} = A\left(\frac{\pi\sigma_k}{\bar{H}}\right)^2, \tag{12}$$

where $S_{\text{reg}}$ is that characteristic scale of the regularization term $J_{\text{reg,raw}}$.

We then use the characteristic scales to normalize both components of the cost function,

$$J_{\text{obs}} = \frac{J_{\text{obs,raw}}}{S_{\text{obs}}}, \tag{13}$$

$$J_{\text{reg}} = \frac{J_{\text{reg,raw}}}{S_{\text{reg}}}, \tag{14}$$



where $J_{\text{obs}}, J_{\text{reg}}$ are the scaled dimensionless cost function terms. The final cost function is a linear combination of the two
scaled terms,

$$J = J_{\text{obs}} + \lambda J_{\text{reg}}, \tag{15}$$

where $J$ is the total cost function that is minimized by the inversion and $\lambda$ is a dimensionless Tikhonov regularization parameter
(Tikhonov and Arsenin, 1977). We determine the optimal $\lambda$ value through an L-curve analysis, discussed next.

### 3.7  L-Curve Analysis

For our L-curve analysis, we sample the range $10^{-3} \leq \lambda \leq 10^3$ with 25 logarithmically spaced samples, resulting in 4 repeating
samples per decade $(1, 1.8, 3.2, 5.6, 10, 18...)$. For each sample, we run the inverse model to convergence and record $J_{\text{obs}}$ and
$J_{\text{reg}}$. An example of the result is shown by the black dots in Fig. 5a.

Before selecting the optimal $\lambda$ value, we next construct a smoothed tradeoff curve based on our 25 individual model results.
Constructing a smoothed curve is necessary because we identify the corner of the L-curve by computing the curvature of
our cost function components, and taking the 2$^{\text{nd}}$ derivative tends to amplify noise. To make the smoothed tradeoff curve, we
first subsampled our 25 individual model results onto 1000 logarithmically spaced $\lambda$ subsamples. We then tested 50 different
smoothing wavelengths, ranging from a minimum of half the separation of the original model points to a maximum of $\ln(100)$.
Note that all smoothing was performed in $\ln(\lambda)$ space.

The selection of the optimal smoothing wavelength can be regarded as a sort of meta-regularization: increased smoothing
results in a worse fit between the tradeoff curve and the original model points, but reduced smoothing increases the random
fluctuations in the curvature of the tradeoff curve. We would like to select a smoothing wavelength that balances these concerns.
Thus, for each smoothing wavelength we computed the variance due to curvature of the smoothed curve and the variance due
to scatter of the original model points about the smoothed curve. These variances are,

$$V_{\text{s}} = \quad \frac{1}{N} \sum_{i=1}^{N} \left( (J_{\text{obs}} - J_{\text{d,c}})^2 + (J_{\text{reg}} - J_{\text{r,c}})^2 \right), \tag{16}$$

$$V_{\text{c}} = \quad \int_{\lambda_{\min}}^{\lambda_{\max}} \left( \left( \frac{d^2 \ln(J_{\text{d,c}})}{d\ln(\lambda)} \right)^2 + \left( \frac{d^2 \ln(J_{\text{r,c}})}{d\ln(\lambda)} \right)^2 \right) d\ln(\lambda), \tag{17}$$

where $V_{\text{s}}$ and $V_{\text{c}}$ are the scatter and curvature variances of the curves, $(J_{\text{obs}}, J_{\text{reg}})$ are the cost function components from the
$N$ individual inverse models, and $(J_{\text{d,c}}, J_{\text{r,c}})$ are the cost function components from the smoothed tradeoff curve. As in our
original analysis, both $V_{\text{s}}$ and $V_{\text{c}}$ need to be normalized before they can be combined; we simply normalized each by their
respective maximum values, then added them together and selected the wavelength that minimized the combined variance.

Once we have a smoothed tradeoff curve, we computed the total logarithmic curvature from $\frac{d^2 \ln(J_{\text{d,c}})}{d\ln(\lambda)} + \frac{d^2 \ln(J_{\text{r,c}})}{d\ln(\lambda)}$ (Fig. 5d).
The location of maximum curvature represents the sharpest corner in the L-curve and the best-value $\lambda$. We also record where
the total curvature drops to $1/2$ of its maximum value, in order to bracket the full range of the corner. Thus, we have three $\lambda$
values: minimum acceptable, best, and maximum acceptable. The logarithmic uncertainty in these three $\lambda$ values is given by
$1/2$ the smoothing wavelength chosen above, and the equivalent uncertainty ratio on a linear scale is simply the exponential of
that uncertainty.



## 3.8 Experimental Design

In addition to regularization, we also test the sensitivity of our results to mesh size, flow equation, sliding nonlinearity, and the use of effective pressure. We test the influence of regularization through L-curve analysis, and we created separate independent L-curves whenever we tested anything else in our experimental setup: for instance, when we varied mesh size, we created a

separate L-curve for each mesh that we tested. We organized our experimental setup around the linear Weertman sliding law as a reference case: we produced L-curves using linear Weertman inversions for each of our 10 meshes in order to test the dependence on mesh size, while for our highest-resolution mesh, we also ran tests with Budd sliding using our 3 candidate $N$ fields, along with nonlinear Weertman sliding. As we found that $N_{\mathrm{CUAS}}$ performed better than the other $N$ fields for linear sliding, we decided to also test $N_{\mathrm{CUAS}}$ with nonlinear sliding. We performed experiments comparing SSA inversions with HO

inversions using linear Weertman sliding in Meshes # 4, 6, and 8, along with one additional HO experiment in Mesh #6 using constant rheology instead of variable rheology. All in all, the experiments presented in this paper comprise 21 independent L-curves (10x linear Weertman SSA, 2x nonlinear Weertman SSA, 3x linear Budd SSA, 2x nonlinear Budd SSA, and 4x linear Weertman HO), each composed of 25 separate inversions, for a total of 525 individual inversions.

## 4 Results

### 4.1 L-curves

A representative L-curve is shown for the case of mesh #1 and linear Weertman sliding in Figure 5a. Our model results form a well-defined "L" shape in $(\ln(J_{\mathrm{reg}}),\ln(J_{\mathrm{obs}}))$ space, and the visual impression of an "L" shape can be trusted in this figure because we have taken care to scale the plot so that a decade of dynamic range takes the same amount of space on both axes. Both components of the misfit are monotonic functions of $\lambda$, and our smoothed tradeoff curve runs very close to the original

model points (Fig. 5a,b), giving us confidence in our inversion procedure. Previously published inversions have sometimes been vexed by problems such as deviations from L-curve monotonicity or outlier models that fall far from the smooth L-curve (e.g. Gillet-Chaulet et al., 2012, Fig. 3); while we cannot be sure of what caused these results for other authors, during development of our model setup, we found that such issues were often an indication of lower-level numerical problems such as suboptimal choice of solvers, overly lax convergence tolerances, or mask topologies with too much complexity at the scale of individual

mesh elements. Once those issues were fixed, we found that smooth monotonic results such as those in Fig. 5 were typical for all of our inversions.

Far from the corner region, both limbs of our L-curve tend towards straight lines, but for the large-$\lambda$ limb this line is sloping, indicating a power-law relationship between $J_{\mathrm{reg}}$ and $J_{\mathrm{obs}}$, while for the small-$\lambda$ limb this line is horizontal, indicating an asymptotic approach to a best achievable $J_{\mathrm{obs}}$ (Fig. 5a). The straight-line nature of the limbs can be confirmed by looking at

the gradient of the cost terms: both terms approach constant gradient for large $\lambda$, while $J_{\mathrm{obs}}$ smoothly approaches zero gradient for small $\lambda$ (Fig. 5c). The low-$\lambda$ limb corresponds to the "flat" part of the L-curve identified by Hansen and O'Leary (1993, Section 4.1), where the overall cost function is dominated by regularization, while the high-$\lambda$ limb corresponds to the "vertical"





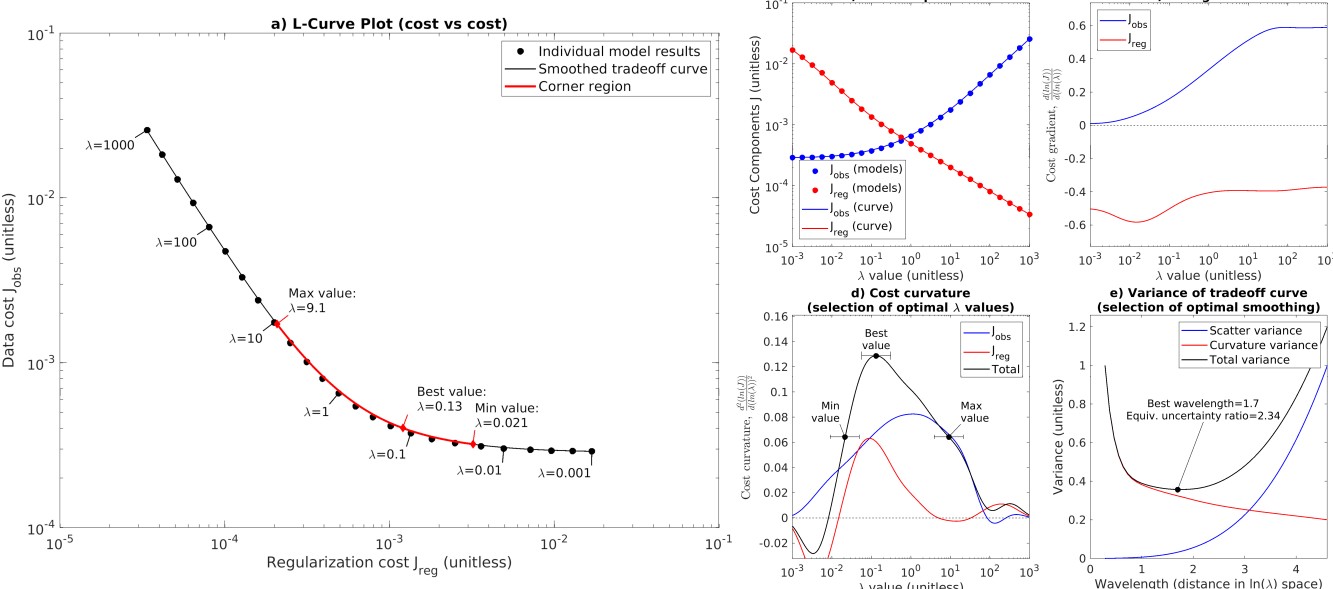

**Figure 5.** Example L-curve for mesh #1. Large plot (a) shows the L-curve, a log/log plot of $J_{\mathrm{reg}}$ vs $J_{\mathrm{obs}}$. Black circles show individual inversions, black line shows the smoothed tradeoff curve fitted to the individual models, and red region highlights the corner of the L-curve. Note that we have scaled the axes in plot (a) so that one order of magnitude in $J_{\mathrm{reg}}$ takes up precisely as much space as one order of magnitude in $J_{\mathrm{obs}}$. b) Plot of both misfit components ($J_{\mathrm{reg}}$ and $J_{\mathrm{obs}}$) against $\lambda$. c) Cost gradient as a function of $\lambda$. d) Cost curvature as a function of $\lambda$, with selection of min, best, and max $\lambda$ values annotated. e) Selection of optimal smoothing wavelength, showing scatter and curvature variance ($V_{\mathrm{s}}$ and $V_{\mathrm{c}}$), selection of optimal smoothing wavelength, and equivalent uncertainty ratio for $\lambda$ on a linear scale.

part where the cost function is dominated by observational misfit, although for linear sliding the slope of this limb is far from vertical (for nonlinear sliding the slope of this limb increases, as we show in Section 4.6). The power-law nature of the large-$\lambda$

limb may potentially explain the results found by Habermann et al. (2013). Those authors found that their L-curve lacked a corner when plotted on log-log axes, motivating them to use linear axes instead (Habermann et al., 2013, Section 3.4). While we cannot be certain what caused them to find this result, our own results suggest that they may have been testing $\lambda$ values in the large-$\lambda$ limb. Normalizing cost function terms by an estimate of their characteristic scale before the L-curve analysis is performed helps to prevent this possibility by ensuring that the L-curve analysis covers a range of $\lambda$ that includes both limbs.

For Weertman sliding, the curvature of the data and regularization terms are offset from each other, with $J_{\mathrm{reg}}$ having peak curvature at smaller $\lambda$ values than $J_{\mathrm{obs}}$, and the total curvature used to select the best $\lambda$ value represents a compromise between these two terms (Fig. 5d); while for Budd sliding, the peak curvature in $J_{\mathrm{obs}}$ and $J_{\mathrm{reg}}$ overlapped more. On the original L-curve plot (Fig. 5a), the best-$\lambda$ value computed from curvature falls roughly at the visual corner of the plot, while the minimum and maximum acceptable $\lambda$ values bracket the transition between the corner regime and the straight-line limbs. In this example,

the smoothing applied to generate the tradeoff curve corresponds to a wavelength of 1.7 in $\ln(\lambda)$ space, or an uncertainty of $\pm$ a factor of about 2 in linear space (Fig. 5e). The uncertainty in $\lambda$ was generally in the range of $\pm$ a factor of 2–3 for all of our





experiments; given that the values of $\lambda$ we tested differed by a factor of $1.8$, this implies that we were generally able to identify $\lambda_{\text{best}}$ to within 2–3 adjacent $\lambda$ samples.

While, for simplicity's sake, we are only showing a single representative L-curve analysis in Figure 5, the features we
described above are generally consistent for all of the L-curves that we produced, although the exact position of the curves (and the corresponding position of the min, max, and best $\lambda$ values) systematically shifted towards smaller $\lambda$ as mesh resolution increased. In addition, the minimum achievable $J_{\text{obs}}$ in the small-$\lambda$ limit dropped as mesh resolution increased. We discuss the convergence of our results as a function of mesh resolution in greater detail in Section 4.4. In Section 4.5, we discuss how our L-curve results differ for Budd sliding laws with different $N$ fields, and in Section 4.6 we discuss how our L-curves differ
for nonlinear sliding. But first we discuss the spatial and spectral characteristics of our results for linear Weertman sliding as a function of regularization.

## 4.2 Spatial Analysis

In Figure 6, we show our inversion results for linear Weertman sliding in our highest resolution mesh for minimum, maximum, and best $\lambda$ values. In general, $C$ values are orders of magnitude lower in fast-flowing ice streams and outlet glaciers than
in slow-flowing regions of the ice sheet (Fig. 6a,b,c). The distribution of $\tau_{\text{b}}$ is more complex, however, with some of the highest drag values occurring in narrow ribs or sticky spots within fast-flowing ice streams (Fig. 6d,e,f). As expected, there is a clear trend towards greater smoothness in $C$ and $\tau_{\text{b}}$ for larger $\lambda$ values, and greater small-scale structure in those fields for smaller $\lambda$ values. Unsurprisingly, velocity misfit declines for smaller $\lambda$ values (Fig. 6g,h,i). Absolute misfits tend to be higher in faster-flowing regions for all $\lambda$ values, reflecting the larger velocity magnitude in those regions. However, the $\lambda_{\text{max}}$
model also contains a large amount of positive misfit (ie, model velocity too high) in the slow-flowing regions in between fast-flowing outlet glaciers and in the islands within the floating shelf (Fig. 6i), likely because the larger regularization prevented the inversion from fully capturing the sharp rise in $C$ across the glacier shear margins and grounding lines.

We also show details of the inverted structure within several fast-flowing ice streams and outlet glaciers. In response to the Sergienko and Hindmarsh (2013) hypothesis of regular rib-like patterns in basal drag, we choose these detail regions
to cover a spectrum from very "rib-like" appearance to very "non-rib-like" appearance: Academy Glacier just upstream of its confluence with Foundation Ice Stream has some of the strongest ribs in our inversion, followed closely by downstream Slessor Glacier, while downstream Recovery Glacier has a mix of rib-like structure and non-rib sticky spots, and downstream Rutford Ice Stream has very little visible ribbing. Unsurprisingly, the ribs are strongest in the inversion with minimum acceptable $\lambda$ (Fig. 6a,d) and weakest in the inversion with maximum acceptable $\lambda$ (Fig. 6c,f), with the best $\lambda$ inversion being intermediate.
The same ribs appear in the $\lambda_{\text{min}}$ and $\lambda_{\text{best}}$ inversions, albeit wider in the latter; while in the $\lambda_{\text{max}}$ inversion, adjacent ribs have begun to combine. In Academy and Slessor glaciers, almost all of the basal resistance is contained within the ribs, while in Recovery and Rutford glaciers ice flow is resisted by a combination of more rounded sticky spots in the middle of the fast-flow region along with stress transmission to higher-drag margins (Fig. 6d,e). We explore the spectral characteristics of these four detail regions next.



**Figure 6.** Comparison of inversion results for the highest-resolution mesh as a function of $\lambda$. Top row (a,b,c) shows drag coefficient $C$ on a logarithmic scale, middle row (d,e,f) displays drag $\tau_b$ on a linear scale, and bottom row (g,h,i) shows velocity misfit on a symmetric logarithmic scale (ie, these plots use separate logarithmic scales for positive and negative errors). Left column (a,d,g) shows minimum acceptable $\lambda$, middle column (b,e,h) shows best $\lambda$, and right column (c,f,i) shows maximum acceptable $\lambda$. All plots show zoom-in panels for Slessor, Recovery, and Academy Glaciers, along with Rutford Ice Stream.

## 4.3 Spectral Analysis

For each region, we first interpolated $\tau_b$ from the model mesh onto a regular grid aligned with the rectangular detail box. In order to minimize spectral artifacts produced by the assumption of periodic boundary conditions in the FFT algorithm, we expanded the grid by 50% in each direction and filled the buffer zone by a smooth extrapolation that approached a constant value around the edges of the expanded grid. We then took the two-dimensional discrete Fourier transform to produce 2D spectra in $(k_x, k_y)$ space, and transformed these into 1D spectra using $k_{\mathrm{mag}} = \sqrt{k_x^2 + k_y^2}$, followed by integration of spectral





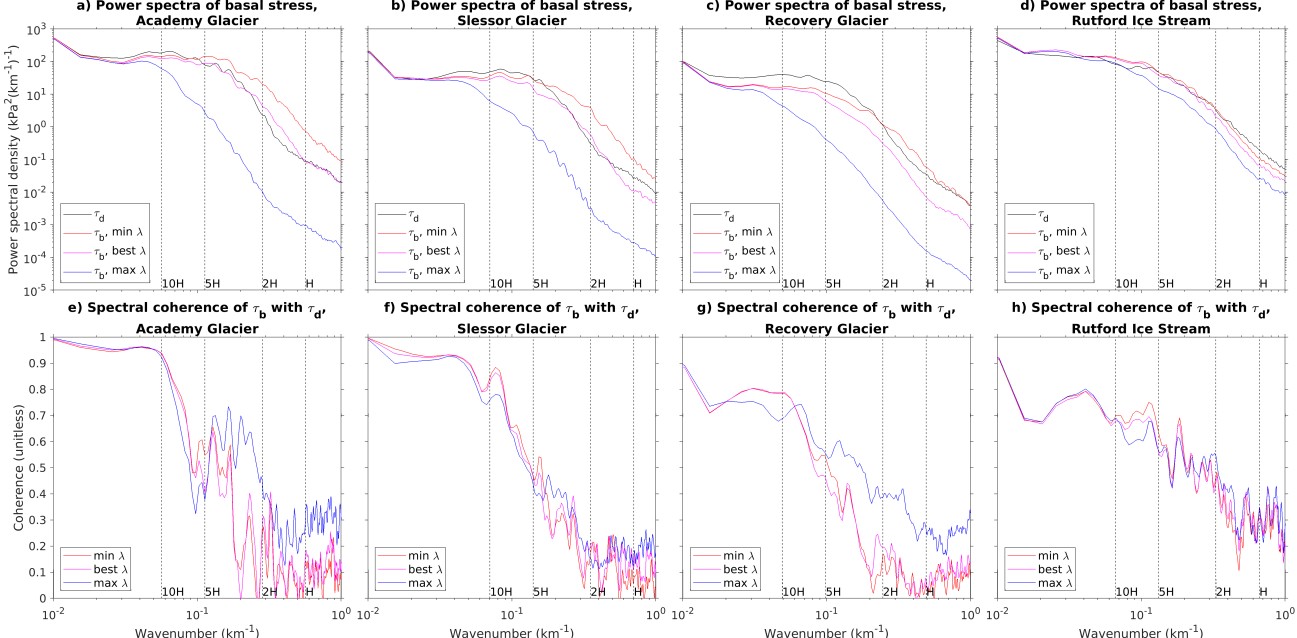

**Figure 7.** Spectral analysis for all four detail regions highlighted in Fig. 6. Top row (a–d) shows power spectral density of basal stress ($\tau$). Colored lines indicate spectra for minimum acceptable, maximum acceptable, and best $\lambda$ values, while black lines indicate spectra for driving stress in the flow direction. Bottom row (e–h) show coherence between basal stress $\tau_b$ and driving stress $\tau_d$. Vertical dashed lines in all plots indicate wavelengths for select multiples of the local average ice thickness.

power around circles of constant $k_{mag}$. We also computed the spectral coherence between basal drag $\tau_b$ and driving stress $\tau_d$ by taking the dot product between the two spectra in $(k_x, k_y)$ space and then integrating in a similar manner.

We show the results of this spectral analysis of basal drag for the four detail regions in Fig. 7. We have organized the regions in order of visual "ribiness", from Academy (Fig. 7a,e) to Rutford (Fig. 7d,h). All four glaciers have roughly constant spectral

power in the $\lambda_{min}$ and $\lambda_{best}$ inversions from wavelengths of about 30 H to 5 H. Below 5 H, spectral power decays rapidly, while for the $\lambda_{max}$ inversion, the decay begins just above 10 H. The three most "ribby" glaciers have large differences in spectral power as a function of $\lambda$ at short wavelengths, especially when comparing $\lambda_{max}$ to the other two (Fig. 7a–c), while Rutford has comparatively small differences as a function of $\lambda$ (Fig. 7d). For all glaciers, the largest spread in spectral power between $\lambda_{min}$ and $\lambda_{max}$ occurs at a wavelength of about twice the ice thickness; for Academy Glacier the difference between $\lambda_{min}$ and $\lambda_{max}$

is about three orders of magnitude at this wavelength, while for Rutford Ice Stream the difference is less than one. For Academy and Slessor glaciers, the $\lambda_{min}$ inversion actually has about an order of magnitude more spectral power than the driving stress at this wavelength, and $\lambda_{min}$ remains above the driving stress for all wavelengths less than 4–5 H (Fig. 7a,b). However, $\lambda_{best}$ has less spectral power than the driving stress at most wavelengths and analysis regions, only slightly exceeding the spectral power in driving stress for Academy and Slessor glaciers near a wavelength of 2 H (Fig. 7a–d). Academy and Slessor glaciers also





have very high ($\gtrsim 90\%$) coherence values between $\tau_\mathrm{b}$ and $\tau_\mathrm{d}$ at long wavelengths, but a steep drop in coherence towards shorter wavelengths (Fig. 7e,f). This indicates that, in the "ribby" regions, driving stress and basal drag are locally balanced at scales greater than about 10 ice thicknesses, but this balance drops rapidly at scales of 5x or 2x the ice thickness. Thus, many of the high-resolution ribs revealed by the inversion in these regions are not mere copies of structure observable in the driving stress; the inversion adds value by revealing structure that cannot be easily inferred from ice sheet geometry. In Recovery Glacier

and Rutford Ice Stream, by contrast, the gradient in coherence between long and short wavelengths is weaker(Fig. 7g,h), with values of $\sim 75\%$ even at wavelengths much larger than $10\,\mathrm{H}$. This is probably a reflection of stress transmission in those glaciers, in which fast-flowing trunks with nearly zero basal drag are supported by long-distance stress transmission to shear margins or isolated sticky spots (Fig. 6d–f).

Overall, these results provide qualified support to the hypothesis advanced in Sergienko and Hindmarsh (2013) and Sergienko

et al. (2014) that basal drag is often concentrated within narrow elongated ribs, at least for some of the ice streams and glaciers within our domain. Our results alone cannot confirm that the specific till-water instability mechanism that they proposed is responsible for creating these ribs, but clearly some process must be invoked to explain this common spatial pattern. However, in contrast to the suggestion in those papers that the finding of ribbed structure was independent of regularization, we find that the narrow ribs are highly sensitive to the choice of $\lambda$ value. Spectral analysis of our inversion results reveals that the

most prominently ribbed regions also display the strongest dependence of short-wavelength structure on $\lambda$, while the least ribbed glacier we analyzed (Rutford Ice Stream) also had the weakest dependence of short-wavelength spectral power on regularization. These results emphasize the importance of choosing the correct regularization before using an inversion to draw conclusions about the short-wavelength structure in the ice sheet basal drag.

### 4.4   Resolution dependence

The definition of numerical convergence of inverse model results is more complicated than the definition of convergence for forward models. Vogel (1996) defines "convergence" to mean that the results of the inversion asymptotically approach the true (continuously resolved) solution as the number of data points and model nodes both approach infinity. While this definition seems sensible at first glance, it implies that $\lambda \to 0$ as model resolution improves; in practical terms, this means that glaciologists would be chasing a moving target, as improvements in the resolution of surface velocity observations, ice

geometry grids, and model meshes would allow ever finer structure of the basal drag to be inferred. Under this definition of "convergence" it would make no sense to directly compare inversion results from different meshes, as we would expect new structure to be revealed with every improvement in resolution. However, the utility of using glaciological inversions to infer spatial structure at scales less than the ice thickness is questionable. Only the FS equations are valid at such short lengthscales, and furthermore the thickness of the ice sheet acts as a filter, attenuating the observable surface effects of short-wavelength

basal structure (Gudmundsson, 2003). While ice sheet basal properties certainly do vary at all lengthscales, it is doubtful that anyone in the long history of glaciological inverse modeling expected to be able to infer structure at scales much smaller than the ice thickness on the basis of surface observations. By contrast, what Vogel (1996) define as "nonconvergence" looks much more like convergence in practical terms: as model resolution improves, $\lambda$ approaches a non-zero limit and the inversion





**Figure 8.** Convergence of L-curve analysis with mesh resolution. a) All L-curves with Weertman sliding overlain on each other. For clarity, only the fitted smooth tradeoff curves are shown, with solid diamonds marking $\lambda_{\text{best}}$ for each mesh. Curves are colored by mesh size. b) Estimated values of $\lambda_{\min}$, $\lambda_{\text{best}}$, and $\lambda_{\max}$, along with their associated uncertainties, as a function of mesh resolution. Best-fit exponentials (linear plots on the semi-logarithmic axes used here) are shown as well. c) RMS misfit in drag coefficient ($\ln(C)$) and drag ($\tau_{\text{b}}$) as a function of mesh resolution. Each data point in this plot represents a comparison between $\lambda_{\text{best}}$ for a particular mesh with $\lambda_{\text{best}}$ in Mesh #1. Solid lines represent the fast-flow domain ($\gtrsim 15\,\mathrm{m\,a^{-1}}$), dashed lines represent the entire grounded domain. d) As (c), but showing correlation coefficients instead of RMS misfits. As discussed in the text, we use the area-weighted median of hydraulic diameter in moderately fast-flow regions ($\gtrsim 15\,\mathrm{m\,a^{-1}}$) to represent mesh resolution for our unstructured refined meshes.

solution approaches the best achievable picture of ice sheet basal drag. As we show below, it is this second sense of practical

convergence that applies to glaciological inversions.



In Figure 8, we show the resolution dependence of our results. As discussed in Section 3.4, we use the area-weighted median hydraulic diameter to represent mesh size for our anisotropic mesh elements, and we here restrict our analysis to mesh size in the fast-flowing part of the grounded ice sheet ($\gtrsim 15\,\mathrm{m\,a^{-1}}$). We find that, in the low-$\lambda$ limit, the minimum achievable $J_{\mathrm{obs}}$ systematically rises with increasing mesh size, while the entire L-curve systematically shifts towards lower $J_{\mathrm{reg}}$ (Fig. 8a). That

is, not only do inversions with coarse meshes produce less structure than inversions with fine meshes at equal values of $\lambda$, but in addition coarse meshes are also unable to fit the data as well as fine meshes, regardless of $\lambda$. The inferred values of $\lambda_{\mathrm{min}}$, $\lambda_{\mathrm{best}}$, and $\lambda_{\mathrm{max}}$ all systematically drop as mesh resolution improves (Fig. 8b).

The dependence of all three $\lambda$ values on mesh size is well fit by an exponential function, indicating that our results are converging towards the following non-zero $\lambda$ values in the continuous solution: $\lambda_{\mathrm{min}} = 0.02$, $\lambda_{\mathrm{best}} = 0.16$, and $\lambda_{\mathrm{max}} = 7.8$.

Because we normalized our cost function components before combining them, we can interpret these $\lambda$ values as "small": at least for high resolution meshes, our best estimate is that the regularization term only needs to be weighted about $1/6$ as strongly as the data term. The mesh size dependence of regularization is described by the e-folding mesh size, which is about 2–3 km for all three $\lambda$ values. (Fig. 8b). Since the average ice thickness in our domain is also about 2 km, this indicates that mesh resolution must be on the order of the ice thickness before an L-curve analysis will produce values of $\lambda$ that are within a

factor of $e$ of their continuous values. Once mesh resolution is better than the ice thickness, diminishing marginal returns set in, and further improvements in resolution will not allow the inversion to resolve finer structure.

In addition to examining variation in $\lambda$ values, we would also like to explore the convergence of our inversion results themselves. We obviously do not have the continuous solution to compare our results with, but we can get an approximate picture by comparing our inversion results for $\lambda_{\mathrm{best}}$ in each mesh with the inversion results in Mesh #1 (Fig. 8c,d). This

comparison reveals that misfits for the entire grounded domain are lower (Fig. 8c), and correlations higher (Fig. 8d), than for the fast-flow domain alone. This is likely because the spatial structure of the entire domain is dominated by the large-scale structure of streaming flow (Fig. 6), which is resolved even in the coarsest mesh we used, while restricting our analysis to the fast-flow domain represents a more difficult test of model convergence. The RMS misfits for $\ln(C)$ and $\tau_{\mathrm{b}}$ track each other closely, with values for the fast-flow domain in the range of 0.6–1.6 for $\ln(C)$ and 15–35 kPa for $\tau_{\mathrm{b}}$ (Fig. 8c). However,

correlation $R^2$ values are consistently higher for $\ln(C)$ than for $\tau_{\mathrm{b}}$ (Fig. 8d).

In addition to comparing inversion results at the (variable) $\lambda_{\mathrm{best}}$ for each mesh, we can also compare our results with Mesh #1 at a constant $\lambda$ value (Fig. 9). This comparison is useful because inversions with different meshes but the same $\lambda$ value should, in principle, be solving the exact same set of equations, and so any differences can be treated as model error. We overlay our exponential fits from Fig. 8b for reference. Above the line representing $\lambda_{\mathrm{best}}$, misfit in both $\ln(C)$ (Fig. 9a) and $\tau_{\mathrm{b}}$

(Fig. 9b) is low, with only a weak dependence on $\lambda$, while correlation in both of those parameters is high (Fig. 9c,d). Between the lines representing $\lambda_{\mathrm{best}}$ and $\lambda_{\mathrm{min}}$, misfit begins to rise (Fig. 9a,b) and correlation begins to drop (Fig. 9c,d), and below $\lambda_{\mathrm{min}}$ both fit metrics worsen rapidly.

These results have a powerful implication: they imply that an L-curve analysis, performed on a single mesh size without reference to other meshes, can be used to separate structure that has converged with respect to mesh resolution from structure

that has not converged. If, for example, numerical constraints limited us to using a mesh with an element size three or four





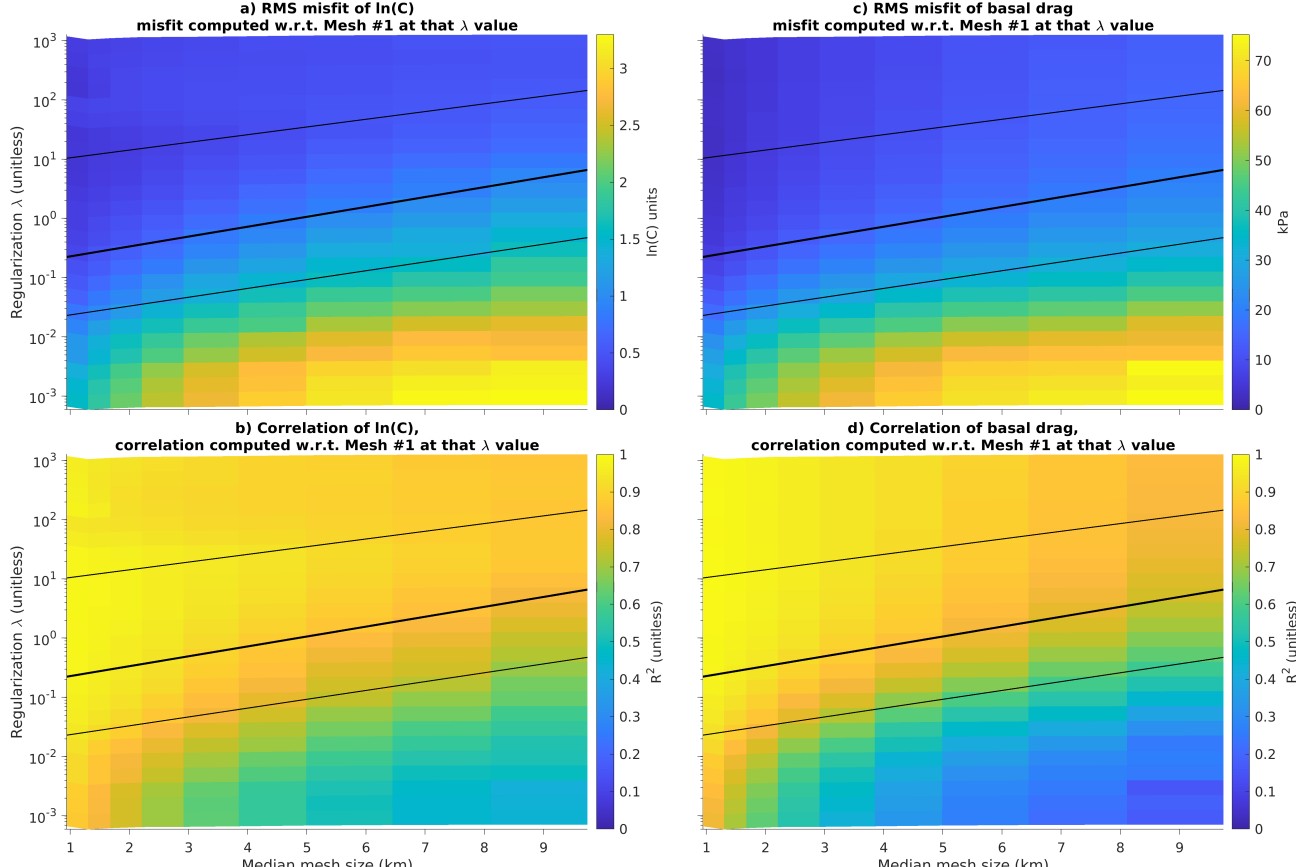

**Figure 9.** Comparison of inversion results as a function of mesh resolution and $\lambda$ value. In these plots, each inverted $C$ or $\tau_b$ field is compared to the corresponding field at the same $\lambda$ value at the highest-resolution mesh. Left column (a,c) shows comparison of drag coefficient $C$ while right column (b,d) shows comparison of drag $\tau_b$. Top row (a,b) shows RMS misfit while bottom row (c,d) shows squared correlation coefficient. Best-fit lines from Fig. 8 overlain for comparison. Comparisons are limited to moderately fast-flowing regions ($\gtrsim 15\,\mathrm{m\,a^{-1}}$).

times the ice thickness, we know from our previous analysis that we would be unable to resolve all of the structure that could be recovered in the continuous problem. Nonetheless, we would still like to have some way of ensuring that the coarser structure we did recover would still be present in the continuous solution, even if in the continuous solution it would also be joined by finer structure we cannot recover. The misfit analysis in Figure 9 suggests that, for any given mesh size, $\lambda_{\mathrm{best}}$ and $\lambda_{\mathrm{min}}$

represent the approximate boundary between spatial structure that has converged with respect to mesh resolution and spatial structure that has not converged. Of course, reducing $\lambda$ will always cause the inversion to add more fine structure and reduce the observational misfit, regardless of mesh resolution; but below $\lambda_{\mathrm{best}}$ (or, at the lowest, $\lambda_{\mathrm{min}}$) this additional structure will not match the structure produced by higher-resolution inversions at the same $\lambda$ value, suggesting that it is erroneous.




## 4.5 Effective pressure

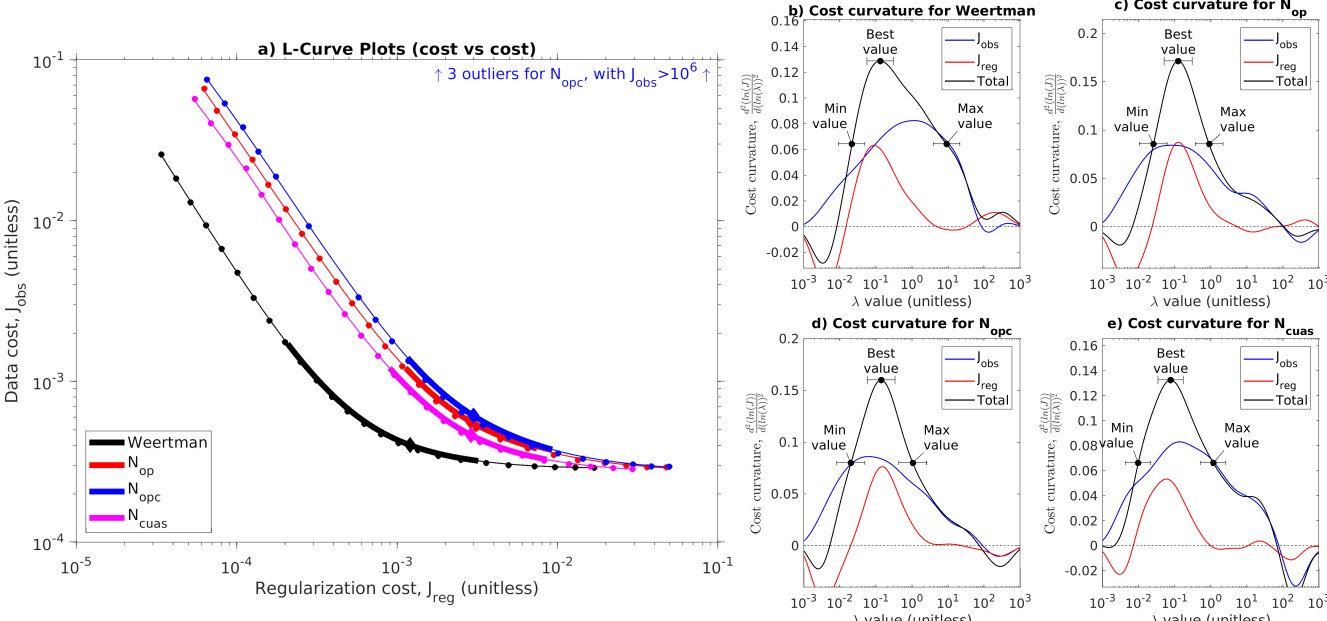

**Figure 10.** Comparative L-curves for different $N$ fields. a) All L-curves overlain on each other. Small circles represent individual models, thin lines represent smoothed curves, thick lines represent inferred corner regions, and large diamonds represent inferred $\lambda_{\text{best}}$ values. b–d) Cost curvature as a function of $\lambda$ for each $N$ field, with selection of min, best, and max $\lambda$ values annotated. Plot (a) is equivalent to plot (a) in Fig. 5, plots (b–d) are equivalent to plot (d) in that figure. Three outlier models from $N_{\text{opc}}$ have been excluded from this analysis.

Before analyzing our results for Budd sliding in detail, it is useful to consider Equation 4: in order to reproduce the stress and velocity fields recovered by the Weertman inversion, the Budd coefficient must satisfy the relationship, $C_{\text{W}} = C_{\text{B}}N$, where $C_{\text{W}}$ is the coefficient in the Weertman sliding law and $C_{\text{B}}$ is the coefficient in the Budd sliding law. From this, we can form the *a priori* expectation that the utility of a given $N$ field will be dependent on the linear correlation between $N$ and $C_{\text{W}}$. If $N$ has a strong positive correlation with $C_{\text{W}}$, then the inversion will not need to produce much structure in $C_{\text{B}}$ in order to fit the observations; in the limit of a perfect $N$ field and no geological variation in the substrate, the $N$ field would explain all of the variation in $C_{\text{W}}$, and $C_{\text{B}}$ would be a constant. Conversely, if the correlation between $N$ and $C_{\text{W}}$ is weak or even negative, then the inversion will need to produce a lot of structure in $C_{\text{B}}$ in order to fit the observations.

All three of our candidate $N$ fields had a positive correlation with $C_{\text{W}}$, but $N_{\text{CUAS}}$ had much better correlation than either of the two geometry-based fields (Table 1). In addition, all three fields had better correlation values when considering the entire grounded domain than when considering the fast-flow region alone (Table 1). For the geometry-based $N$ fields, this is likely due to the physical setting of fast-flowing regions: many fast-flowing ice streams and outlet glaciers are located in subglacial troughs, so a simple geometry-based calculation incorporating the bed topography can approximate the basic





dichotomy between slow flow and fast flow. For $N_{\mathrm{CUAS}}$, this is likely because the melt rate forcing to CUAS incorporated shear heating estimated from the observed velocity field. Restricting the analysis to the fast-flowing region alone represents a
much more stringent test; $N_{\mathrm{CUAS}}$ can explain 33% of the variance of $C_W$ within the fast-flowing region of the domain, but the two geometry-based $N$ fields can explain almost none of it.

Figure 10 shows comparative L-curves for all of them and Table 1 summarizes the results of a comparison between the $\lambda_{\mathrm{best}}$ inversions for each of those $N$ fields with the $\lambda_{\mathrm{best}}$ inversion for Weertman sliding. The L-curves for all three $N$ fields have a similar shape as the L-curve for Weertman sliding, but shifted towards higher $J_{\mathrm{reg}}$ (Fig. 10a). Total curvature was in
the range of 0.13–0.17 for all four models (Fig. 10b–e), providing a quantitative confirmation of the visual similarity in shape. The inferred $\lambda_{\mathrm{best}}$ values are shifted towards lower $\lambda$ values, mostly driven by a shift in the data curvature term (Fig. 10c–e, c.f. b). A visual inspection of the respective L-curves suggests that the asymptotic best achievable $J_{\mathrm{obs}}$ is similar for all sliding laws, but the $\lambda_{\mathrm{best}}$ inversions for inversions with $N$ have higher $J_{\mathrm{obs}}$ than the $\lambda_{\mathrm{best}}$ inversion for Weertman sliding (Fig. 10a). Three of the inversions in the $N_{\mathrm{opc}}$ L-curve were unable to converge, with $J_{\mathrm{obs}}$ many orders of magnitude higher than the other
models (Fig. 10a), but the remaining inversions were sufficient for us to recover a smooth tradeoff curve.

At face value, Figure 10a implies that $N_{\mathrm{opc}}$ required the most structure to fit the data, almost a full order of magnitude more structure than the inversions with Weertman sliding. However, it is possible that the increase in $J_{\mathrm{reg}}$ could have been caused by the different normalizations used, as the estimate of the characteristic scale $S_{\mathrm{reg}}$ depends on the value of $N$ (Eqs 11, 14). To remedy this, we can obtain a scale-independent estimate of the magnitude of the structure produced by the inversion by
computing the variance of $\ln(C)$. Converting $C$ to a logarithmic scale before computing the variance ensures that the change in units and mean magnitude between Weertman and Budd sliding has no effect on the calculation. This calculation reveals that all of the candidate $N$ fields reduced the variance in the inverted $C$ coefficient when considered over the entire domain, but $N_{\mathrm{CUAS}}$ reduced the variance the most (Table 1). When the fast-flow domain is considered alone, $N_{\mathrm{op}}$ produced a slight *increase* in variance, $N_{\mathrm{opc}}$ a slight decrease, and $N_{\mathrm{CUAS}}$ the largest decrease. All of the inversions with $N$ had larger $J_{\mathrm{obs}}$ values than the
inversion with Weertman sliding, regardless of whether or not the analysis is restricted to the fast-flow domain, with $N_{\mathrm{CUAS}}$ having the smallest increase in observational misfit of the three (Table 1). In addition, the scaling of $J_{\mathrm{obs}}$ is determined only by the statistics of the observations (Eq. 14), so we know that differences in $J_{\mathrm{obs}}$ between models are directly comparable with one another. Nonetheless, the absolute magnitude of error for all of the models is relatively low, with $J_{\mathrm{obs}} < 10^{-3}$ over the whole domain and $J_{\mathrm{obs}} < 2 \times 10^{-3}$ over the fast-flow region; converted back to dimensional form using $\Delta u = \sigma_d \sqrt{J_{\mathrm{obs}}}$,
these correspond to characteristic velocity errors of $\sim 5\,\mathrm{m\,a^{-1}}$ over the whole domain and $\sim 10\,\mathrm{m\,a^{-1}}$ in the fast flow. We can evaluate the overall quality of each model by computing the total variance ratio, which is the product of the coefficient variance ratio and the $J_{\mathrm{obs}}$ ratio (Table 1). Doing so reveals that the increase in observational misfit outweighs the reduction in coefficient variance for the two geometry-based $N$ fields ($N_{\mathrm{op}}$ and $N_{\mathrm{opc}}$) regardless of whether the analysis is performed for the whole domain or for the fast-flow alone; for $N_{\mathrm{CUAS}}$, by contrast, we obtain a reduction in total variance for the whole
domain and a very slight increase in the fast flow.



**Table 1.** This Table summarizes inversion performance for the three sources of effective pressure $N$ that we tested on Mesh #1, alongside results for Weertman sliding (no $N$). Each model is evaluated at its own $\lambda_{\text{best}}$ value. We show statistics both for the entire grounded domain and for fast flow regions only ($\gtrsim 15\,\text{m}\,\text{a}^{-1}$).

| Model | $N$ correlation $R^2(N, C_0)$ (unitless) | $C$ variance $\sigma^2(\ln(C))$ (unitless) | Obs Cost $J_{\text{obs}}$ ($\times 10^{-4}$) | Equiv $\Delta u$ $\sigma_d\sqrt{J_{\text{obs}}}$ (ma$^{-1}$) | Var ratio (a) $\frac{\sigma^2(\ln(C))}{\sigma^2(\ln(C_0))}$ (unitless) | Var ratio (b) $\frac{J_{\text{obs}}}{J_{\text{obs},0}}$ (unitless) | Var ratio (tot) $a \times b$ (unitless) |
|---|---|---|---|---|---|---|---|
| \multicolumn Whole grounded domain: | | | | | | | |
| no $N$ | - | 3.22 | 3.96 | 4.98 | 1.00 | 1.00 | 1.00 |
| $N_{\text{op}}$ | 0.18 | 3.07 | 5.33 | 5.78 | 0.95 | 1.35 | 1.28 |
| $N_{\text{opc}}$ | 0.07 | 2.97 | 6.81 | 6.53 | 0.92 | 1.72 | 1.59 |
| $N_{\text{CUAS}}$ | 0.51 | 2.18 | 5.07 | 5.64 | 0.68 | 1.28 | 0.87 |
| \multicolumn Grounded fast flow: | | | | | | | |
| no $N$ | - | 7.04 | 13.8 | 9.31 | 1.00 | 1.00 | 1.00 |
| $N_{\text{op}}$ | 0.00 | 7.27 | 17.5 | 10.48 | 1.03 | 1.27 | 1.31 |
| $N_{\text{opc}}$ | 0.00 | 6.77 | 19.7 | 11.11 | 0.96 | 1.42 | 1.37 |
| $N_{\text{CUAS}}$ | 0.33 | 5.91 | 16.7 | 10.24 | 0.84 | 1.21 | 1.02 |

## 4.6 Nonlinear Sliding

In Figure 11, we show comparative L-curves for linear and nonlinear Weertman sliding laws. These results reveal that, in a nonlinear sliding law, the high-$\lambda$ limb of the L-curve is substantially steeper than in a linear sliding law, producing a sharper corner in the L-curve (Fig. 10a). The peak in cost curvature became narrower as sliding became more nonlinear, reducing the spread between $\lambda_{\text{min}}$ and $\lambda_{\text{max}}$ and allowing us to define $\lambda_{\text{best}}$ more precisely (Fig. 10c,e,g). The increased visual sharpness of the L-curves for nonlinear sliding can be confirmed by looking at the total curvature, which doubles from 0.13 for $m = 1$ to 0.26 for $m = 5$ (Fig. 11c,e,g). The increase in L-curve sharpness for nonlinear sliding is driven by both a steeper increase in $J_{\text{obs}}$ and a shallower reduction in $J_{\text{reg}}$ with increasing $\lambda$ (Fig. 11b,d,f). Both of these effects can be explained by an increasing sensitivity of velocity to the coefficient in nonlinear sliding. Rearranging Eq. 3 to solve for $u$ reveals that $u \propto C^{-m}$; thus, when $C$ is perturbed away from a value that fits the observations closely, the resulting increase in $J_{\text{obs}}$ will be larger for nonlinear sliding than for linear sliding. The increased sensitivity of $J_{\text{obs}}$ to $C$ in turn prevented the inversion from smoothing $C$ as much as it would for linear sliding, producing higher $J_{\text{reg}}$ values at large $\lambda$. The combination of a smaller decrease in $J_{\text{reg}}$ with a bigger increase in $J_{\text{obs}}$ at large $\lambda$ produces a sharper L-curve.

As in our experiments with $N$, a first glance at our L-curves in Figure 11a seems to imply greater inversion structure with higher values of $m$. However, again care needs to be taken when interpreting $J_{\text{reg}}$, as the scaling of $J_{\text{reg}}$ depends on the value of $m$ (Eq. 11). Again, we can obtain a scale-independent estimate of the amount of structure present in the inverted $C$ field by computing the variance on a logarithmic scale. Doing so reveals that the structure produced by the inversion drops drastically as $m$ increases, dropping by a factor of 3–4 over the whole domain and by a factor of 2–2.5 in the fast flow (Table 2). This





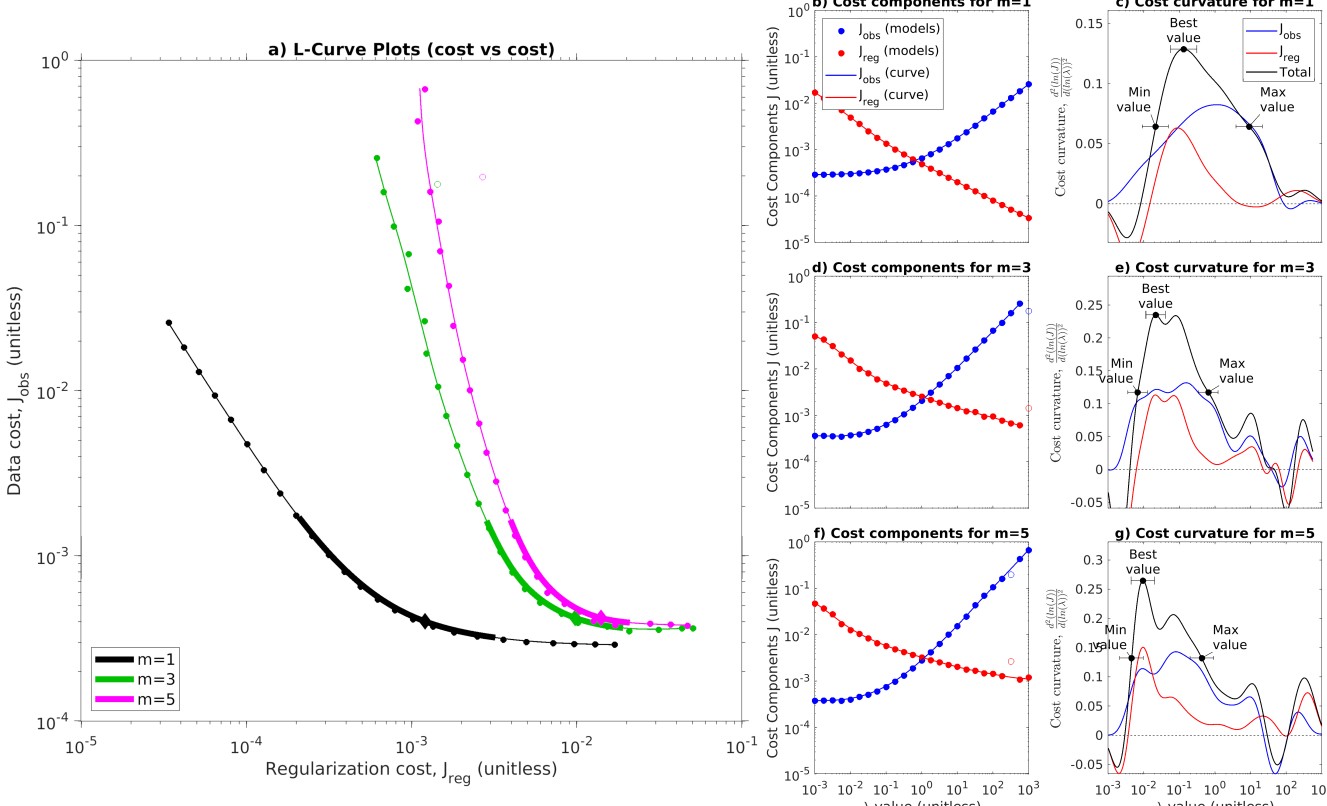

**Figure 11.** Comparative L-curves for different values of the sliding exponent $m$ in Mesh #1. a) All L-curves overlain on each other. Filled circles represent individual models, thin lines represent smoothed curves, thick lines represent inferred corner regions, and large diamonds represent inferred $\lambda_{\text{best}}$ values. Open circles represent two outlier models excluded from the curve-fitting (one each for $m = 3$ and $m = 5$). b) Individual cost components for $m = 1$. Filled circles represent individual models, lines represent fitted curves, and open circles represent outlier models. c) Cost curvature and selection of $\lambda_{\text{min}}$, $\lambda_{\text{best}}$, and $\lambda_{\text{max}}$ for $m = 1$. d–g) As (b) and (c), but for $m = 3$ and $m = 5$.

reduction in variance can also be seen by examining the color scales in Figure 12. At the high end, linear Weertman requires over five and a half orders of magnitude of dynamic range to display the 1%–99% range of its distribution, while at the low end, nonlinear Budd requires less than two and a half orders of magnitude. The spatial structure of $C$ also changes somewhat in nonlinear sliding, with greater ribbing and less of a simple dichotomy between fast and slow regions, producing a $C$ field that starts to have more visual features of the $\tau_{\text{b}}$ field for linear sliding (Fig. 12b,c, c.f. Fig. 6d–f). With nonlinear Budd sliding the large-scale contrast between slow and fast flow is greatly reduced, with much of that distinction being accounted for by $N_{\text{cuas}}$ rather than $C$ (Fig. 12e,f). The coefficient variance can also be compared with the variance of $\ln(\tau_{\text{b}})$, as we expect that $C$ should approach $\tau_{\text{b}}$ as $m$ approaches infinity. For linear Weertman, the variance in the coefficient is nearly three times the variance in basal drag when considered over the whole domain, but this drops to only a 20% increase in variance for $m = 5$. For $N_{\text{CUAS}}$ with $m = 5$, the coefficient variance is only 5% higher than the drag variance over the whole domain, and it is actually



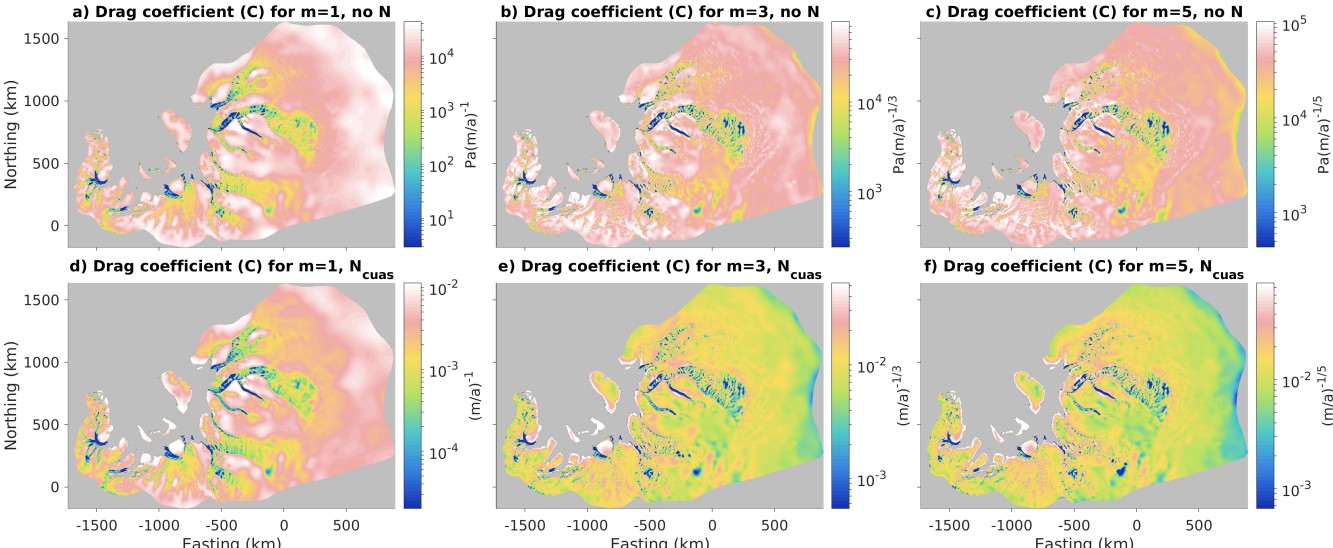

**Figure 12.** Comparative maps of inverted drag coefficient $C$ for different values of the sliding exponent $m$, both with and without effective pressure $N$. Top row shows drag coefficient for Weertman sliding (without $N$), for $m = 1, 3, 5$. Bottom row shows the same values of $m$ for inversions with Budd sliding using $N_{\mathrm{cuas}}$. Each subplot uses a logarithmic color scale with the limits automatically set to the $1^{st}$ and $99^{th}$ percentiles of the distribution.

*7% less* in the fast flow. Observational misfit is slightly higher for nonlinear as compared to linear sliding, although as with our experiments with $N$, the absolute magnitude of $J_{\mathrm{obs}}$ is low for all inversions and the equivalent velocity errors are on the order of $\sim 5\,\mathrm{m\,a^{-1}}$ for the whole domain and $\sim 10\,\mathrm{m\,a^{-1}}$ in the fast flow. However, unlike in our experiments with $N$, for our experiments with $m$ the large reduction in coefficient variance moving to nonlinear sliding more than makes up for the slight increase in misfit, producing total variance ratios that are an unambiguous improvement (Table 2). The improvement produced by moving from $m = 1$ to $m > 1$ in Weertman sliding is about a factor of 3 when measured across the whole domain, and about a factor of 2 within the fast flow. $m = 5$ produced a slight improvement as compared to $m = 3$, but by far the biggest jump came between $m = 1$ and $m = 3$. With $N_{\mathrm{CUAS}}$ the improvement when moving from linear to nonlinear sliding was somewhat less, although still strong (Table 2).

## 4.7 Higher-Order Inversions

We produced L-curves for HO inversions with linear Weertman sliding in Meshes # 4, 6, and 8. We compare these with the corresponding SSA inversions in Figure 13. While we were able to obtain smooth monotonic L-curves with well-defined corner regions for all three experiments, the resulting L-curves clearly indicate that the HO inversions were inferior to SSA inversions using the same mesh size and sliding law. Note that the scalings for both $J_{\mathrm{reg}}$ and $J_{\mathrm{obs}}$ are independent of the flow model used, so the relative positions of the SSA and HO L-curves in Figure 13 are directly comparable to one another. This positioning indicates that the HO inversions performed worse than the SSA inversions in both components of the cost function.



**Table 2.** This table summarizes inversion performance for the three values of the sliding exponent $m$ that we considered in Mesh #1. Each $m$ is evaluated at its own $\lambda_{\text{best}}$ value. We show statistics both for the entire grounded domain and for fast flow regions only ($\gtrsim 15\,\text{m a}^{-1}$).

| Model | $C$ variance $\sigma^2(\ln(C))$ (unitless) | $\tau_{\text{b}}$ variance $\sigma^2(\ln(\tau_{\text{b}}))$ (unitless) | Obs Cost $J_{\text{obs}}$ ($\times 10^{-4}$) | Equiv $\Delta u$ $\sigma_d\sqrt{J_{\text{obs}}}$ (ma$^{-1}$) | Var ratio $\frac{\sigma^2(\ln(C))}{\sigma^2(\ln(\tau))}$ (unitless) | Var ratio (a) $\frac{\sigma^2(\ln(C))}{\sigma^2(\ln(C_0))}$ (unitless) | Var ratio (b) $\frac{J_{\text{obs}}}{J_{obs,0}}$ (unitless) | Var ratio (tot) $a \times b$ (unitless) |
|---|---|---|---|---|---|---|---|---|
| | | | | Whole grounded domain: | | | | |
| $m = 1$, no $N$ | 3.22 | 1.13 | 4.0 | 5.0 | 2.86 | 1.00 | 1.00 | 1.00 |
| $m = 3$, no $N$ | 1.10 | 0.80 | 4.2 | 5.1 | 1.38 | 0.34 | 1.06 | 0.36 |
| $m = 5$, no $N$ | 0.90 | 0.75 | 4.3 | 5.2 | 1.19 | 0.28 | 1.09 | 0.30 |
| $m = 1$, $N_{\text{CUAS}}$ | 2.18 | 0.89 | 5.1 | 5.6 | 2.44 | 0.68 | 1.28 | 0.87 |
| $m = 3$, $N_{\text{CUAS}}$ | 0.87 | 0.80 | 5.8 | 6.0 | 1.09 | 0.27 | 1.47 | 0.40 |
| $m = 5$, $N_{\text{CUAS}}$ | 0.90 | 0.86 | 5.1 | 5.7 | 1.05 | 0.28 | 1.29 | 0.36 |
| | | | | Grounded fast flow: | | | | |
| $m = 1$, no $N$ | 7.0 | 4.4 | 14 | 9.3 | 1.59 | 1.00 | 1.00 | 1.00 |
| $m = 3$, no $N$ | 3.3 | 2.8 | 15 | 9.7 | 1.18 | 0.47 | 1.08 | 0.51 |
| $m = 5$, no $N$ | 2.9 | 2.6 | 15 | 9.7 | 1.11 | 0.41 | 1.08 | 0.45 |
| $m = 1$, $N_{\text{CUAS}}$ | 5.9 | 3.4 | 17 | 10.2 | 1.75 | 0.84 | 1.21 | 1.02 |
| $m = 3$, $N_{\text{CUAS}}$ | 2.9 | 2.9 | 20 | 11.3 | 1.01 | 0.41 | 1.48 | 0.61 |
| $m = 5$, $N_{\text{CUAS}}$ | 2.9 | 3.1 | 19 | 10.8 | 0.93 | 0.41 | 1.35 | 0.56 |

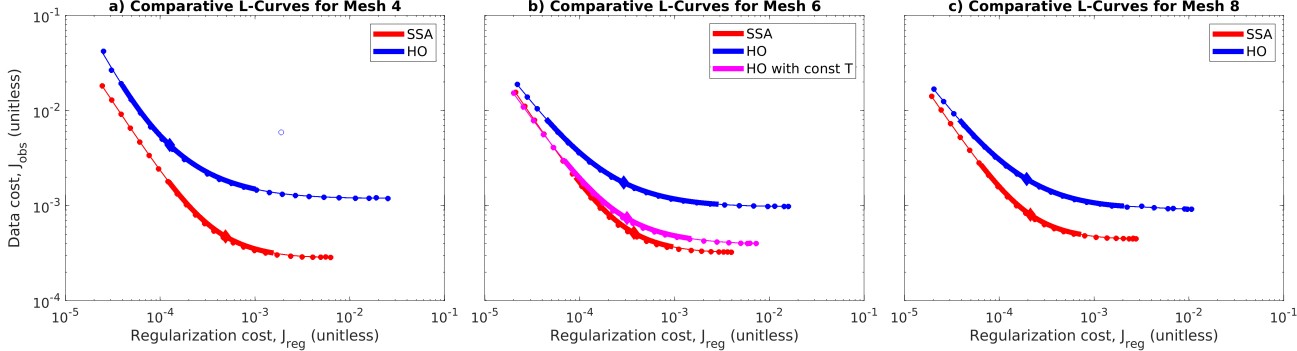

**Figure 13.** Comparative L-curves for HO and SSA inversions. a) Comparative L-curves for Mesh #4. Solid circles indicate model results, thin smooth lines indicate fitted curves, thick lines indicate corner region, diamonds indicate $\lambda_{\text{best}}$ location, and hollow circle indicates an outlier model excluded from the curve-fitting for HO. Red represents SSA L-curve, blue represents HO. b) as (a), but for Mesh #6. Magenta represents HO with constant ice temperature of -25°C. c) as (a), but for Mesh #8.

The difference in the asymptotic best achievable $J_{\text{obs}}$ is particularly notable: the HO inversions were incapable of achieving



values of $J_{\mathrm{obs}}$ less than $10^{-3}$, regardless of mesh resolution. By contrast, the asymptotic best achievable $J_{\mathrm{obs}}$ for the SSA inversions improved with finer mesh resolution, as discussed above.

A detailed examination of the misfit maps for the HO inversions (not shown) revealed that their increased observational misfit was largely due to increased positive misfit (ie, model velocity too high) in slow-flowing regions. Thus, we hypothesized that they may have been unable to fit the data because they contained too much deformation flow: if the temperature model used as input contained too much warm ice near the base, then a vertically resolved HO model would have too much shear deformation in the lower part of the ice column, resulting in model surface velocities that were faster than observed, with no way to bring the model velocities down by adjusting basal slip. We tested this hypothesis by running one additional L-curve experiment in Mesh #6 with HO and a constant ice rheology corresponding to a temperature of -25°C throughout the domain. The results are shown by the magenta line in Fig. 13b. With a constant, and fairly stiff, ice rheology, this HO model was able to make up almost all of the difference with the SSA L-curve.

We summarize the performance of the two HO inversions in Mesh #6 as compared to the SSA inversion for that mesh in Table 3. Basal drag was similar in both the 2D and 3D models, with RMS differences in $\tau_{\mathrm{b}}$ on the order of $10\,\mathrm{kPa}$. The HO model with variable ice temperature required more regularization and thus had $12\%$ less variance in its coefficient than the SSA model, but it also had a big increase in $J_{\mathrm{obs}}$, resulting in a tripling in the total variance ratio (Table 3). The HO inversion with constant ice temperature performed better than this, but still had a $68\%$ increase in total variance as compared to SSA. When the analysis is restricted to the fast-flowing domain, then the HO inversion with variable temperature improves somewhat, dropping down to a variance ratio compared to SSA of "only" 1.91, while the variance ratio for HO with constant temperature is nearly identical within the fast flow as it is in the whole domain (Table 3). These results demonstrate that, while SSA outperforms HO regardless of whether the analysis is performed over the whole domain or in the fast flow alone, the SSA model has the biggest advantage when compared against the HO model with variable temperature in a comparison region that includes both slow and fast flow. We discuss our interpretation of these results and what this finding means for future ice sheet inversions in more detail in Section 5.3.

## 4.8 Best Drag Map

Throughout this paper we have produced many different inversions and competing descriptions of the ice sheet bed. In this subsection, we combine them to produce our single consensus picture of the ice sheet basal drag. We feel that it is more appropriate to produce a consensus view of drag, $\tau_{\mathrm{b}}$, rather than drag coefficient $C$, because drag is directly comparable between different sliding laws, while the units and mean magnitude of $C$ vary with both $m$ and $N$. In this consensus estimate we include all eight of the L-curves that we performed on our highest resolution mesh, that is, Weertman sliding with $m = 1, 3, 5$, Budd sliding with $N_{\mathrm{CUAS}}$ and the same three $m$ values, and linear Budd sliding with $N_{\mathrm{OP}}$ and $N_{\mathrm{OPC}}$. In order to sample the range of uncertainty associated with regularization, for each L-curve we included inversions with $\lambda_{\mathrm{min}}$, $\lambda_{\mathrm{best}}$, and $\lambda_{\mathrm{max}}$. Overall, our combined estimate is built from 24 individual inversions on our highest resolution mesh. Each inversion is weighted according to the inverse of its total variance ratio. For this purpose we use the version of total variance computed for the whole domain.





**Table 3.** This table summarizes inversion performance for HO and SSA models in Meshes #4, 6, and 8. For Mesh #6, we also considered an HO model with constant ice temperature. Each model is evaluated at its own $\lambda_{\text{best}}$ value. Variance ratios are computed with respect to the SSA model at the same mesh size. We show statistics both for the entire grounded domain and for fast flow regions only ($\gtrsim 15\,\text{m a}^{-1}$).

| Model | $C$ variance $\sigma^2(\ln(C))$ (unitless) | $\tau_{\text{b}}$ difference $RMS(\tau_{\text{b}})$ (kPa) | Obs Cost $J_{\text{obs}}$ ($\times 10^{-4}$) | Equiv $\Delta u$ $\sigma_d\sqrt{J_{\text{obs}}}$ (ma$^{-1}$) | Var ratio (a) $\frac{\sigma^2(\ln(C))}{\sigma^2(\ln(C_0))}$ (unitless) | Var ratio (b) $\frac{J_{\text{obs}}}{J_{obs,0}}$ (unitless) | Var ratio (tot) $a \times b$ (unitless) |
|---|---|---|---|---|---|---|---|
| Whole grounded domain: | | | | | | | |
| Mesh #6, SSA | 2.56 | 0.00 | 5.8 | 6.03 | 1.0 | 1.00 | 1.00 |
| Mesh #6, HO | 2.25 | 8.85 | 20.1 | 11.19 | 0.88 | 3.44 | 3.02 |
| Mesh #6, HO, const T | 2.79 | 7.37 | 9.0 | 7.49 | 1.09 | 1.54 | 1.68 |
| Grounded fast flow: | | | | | | | |
| Mesh #6, SSA | 5.33 | 0.00 | 19.2 | 10.9 | 1.00 | 1.00 | 1.00 |
| Mesh #6, HO | 4.36 | 14.2 | 44.8 | 16.7 | 0.82 | 2.33 | 1.91 |
| Mesh #6, HO, const T | 6.38 | 11.4 | 26.5 | 12.9 | 1.20 | 1.38 | 1.66 |

Figure 14 shows the resulting spatial structure of basal drag. A number of faster-flowing regions are supported by rib-like
structures in drag, including upstream Recovery and Slessor glaciers, Academy glacier, and the region of the Robin Subglacial
Basin upstream of Institute and Möller ice streams. Generally these ribby regions are further upstream in the fast flow, while
further downstream many fast-flowing trunks have nearly zero basal drag supported by stress transmission from shear margins
or isolated sticky spots. Fast-flowing trunks with mostly zero basal drag are seen in Recovery and its tributaries, Rutford, Evans,
Foundation, and downstream Slessor glaciers. Stress transmission from fast-flowing trunks to the shear margins produces thin
strips of high drag just outside of the main trunk of many glaciers. In general, more visually ribby structure tends to be located
in moderately fast-flow regions, $\sim 30$–$300\,\text{m a}^{-1}$, where streaming flow is spread out over broad subglacial basins. Where
fast flow is more intense ($\gtrsim 300\,\text{m a}^{-1}$) and concentrated in narrower troughs, the tendency is towards near-zero basal drag
supported by isolated sticky spots or side drag from shear margins. Recovery and Support Force glaciers have regions of high
drag near their grounding lines, but otherwise most glaciers continue their low-drag troughs right into the ice shelf.

**5 Discussion**

Fundamentally, the purpose of regularization is to determine the information content of our observations by managing the
tradeoff between increased complexity in the inversion target field and reduced observational misfit. While one can usually
reduce the observational misfit by adding ever more complexity to the inversion target field, this is equivalent to increasing the
number of free parameters one is allowed to tune, and the meaningfulness of a good observational fit achieved with an excess
of free parameters is questionable. Occam's razor suggests that we should prefer the model that explains the maximum amount
of observational variance with the minimum amount of spatial structure. The ideal $\lambda$ value is thus defined by the point of





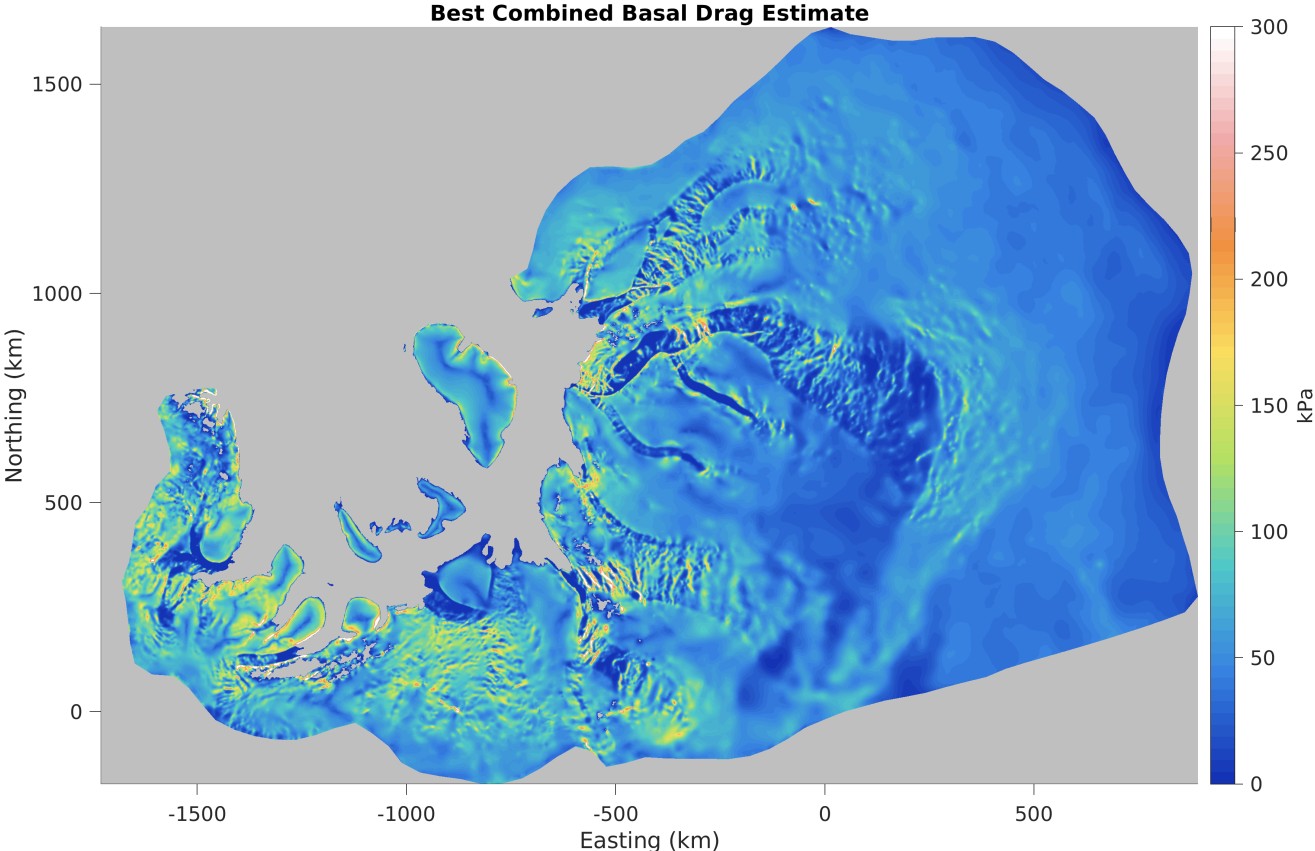

**Figure 14.** Our best combined estimate of the ice sheet basal drag. This map is produced by taking a weighted average of multiple models as described in the text. Each model is weighted by the reciprocal of its total variance ratio.

diminishing marginal returns: starting from a highly regularized state, reducing $\lambda$ improves the observational fit substantially without adding much spatial structure to the target field. At some point, diminishing marginal returns set in, such that further improvements require disproportionately complex spatial structure in the target field to achieve only slight reductions in misfit.

Thus, regularization is not an ad hoc or arbitrary addition to the inversion process, nor should regularization be treated as an inconvenient or unfortunate necessity required to produce numerical stability. Rather, regularization is a fundamental measure of the information content achievable in geophysical inverse problems, and the selection of optimal regularization through L-curve analysis should be regarded as an essential part of glaciological inversions. A well-formed L-curve allows us to answer the question: how much structure is actually required to fit the data? Conversely, a poorly-formed L-curve casts doubt on the

meaningfulness of all of the spatial structure recovered by an inversion.

This focus on information content also provides a useful framework for thinking about the question of convergence in inverse models. We found that our L-curve analysis was not convergent in the sense used by Vogel (1996). This fact is unsurprising, as





their style of convergence is only achievable when the solution is smooth in comparison to the singular values of the operator, and ice sheet basal drag is likely to be highly variable at small scales while the bed to surface transfer function strongly
attenuates short wavelengths (Gudmundsson, 2003; Habermann et al., 2012). However, while Sergienko and Hindmarsh (2013) and Sergienko et al. (2014) argued that the possibility of this form of nonconvergence removed the need for L-curve analysis, we argue the opposite. We argue that a model which is nonconvergent in the Vogel (1996) sense is in fact convergent in the practically useful sense needed by glaciological inverse modelers. To be specific, we postulate that there exists a best knowable basal drag field for the inverse problem using surface velocity data as input. This best knowable basal drag field is not the
same as the true basal drag field, although it is probably related to the true field through a low-pass filtering operation. There are an infinite number of short-wavelength perturbations that can be added to the best knowable drag field that will produce roughly equivalent surface velocity observations; the distinguishing feature of the best knowable drag field is that it has the least amount of structure of any of the fields that are consistent with the observations. Our findings in Section 4.4 indicate that, for any given mesh, $\lambda_{\mathrm{best}}$ determined from an L-curve will produce the best approximation to the best knowable drag field
that can be achieved by a mesh of that size. As mesh size approaches zero, $\lambda_{\mathrm{best}}$ converges towards a non-zero limit and the inversion result converges towards the best knowable drag field. Thus, we conclude that the glaciological L-curve is convergent in the practically useful sense.

## 5.1 Sliding Nonlinearity

It is common in the inverse modeling literature to read some version of the sentiment that, because it is possible to fit the data
and achieve force balance with any sliding law, we therefore cannot use inverse models to distinguish between different sliding laws. For instance, Joughin et al. (2004) state "the inversion results alone do not distinguish which is the most appropriate bed model", wording which is repeated almost verbatim in Joughin et al. (2006). However, we argue that this sentiment fundamentally misunderstands the purpose of inverse modeling, which is not merely to fit the data, but to fit the data using the least amount of structure. All sliding laws are by necessity a parameterization of unresolved small-scale processes, such as
sediment deformation or subglacial cavity formation. In order to be useful, a parameterization must simplify reality in a way that helps us gain understanding. A parameterization that makes the world more complex has failed in this purpose.

In the case of a sliding law, we are creating a parameterization that relates two quantities that each have about 2–3 orders of magnitude of dynamic range: velocity varies between about 1 to about $1000\,\mathrm{m\,a^{-1}}$, and basal drag varies between a few kPa and a few hundred kPa. If we require a parameter with over 5 orders of magnitude of dynamic range to explain the relationship
between two variables that individually have only 2–3 orders of magnitude of range, as is the case for linear Weertman, then our sliding law has fundamentally failed to be a useful parameterization. Of course, it is a mathematical requirement that $C$ must approach $\tau_{\mathrm{b}}$ as $m$ approaches infinity, so the free parameter in a Coulomb plastic sliding law will always have the exact same dynamic range as $\tau_{\mathrm{b}}$, and indeed we found that the variance of $\ln(C)$ was only 10–20% higher than that of $\ln(\tau_{\mathrm{b}})$ for $m = 5$. However, it is not a mathematical requirement that the variance of $\ln(C)$ must approach this limit from the high side.
If linear Weertman were a good description of basal sliding, then we might expect to be able to fit the data with very little variance in $C$. In that case, the variance of $\ln(C)$ would be less than that of $\ln(\tau_{\mathrm{b}})$ for $m = 1$, and that variance would increase





as $m$ approached infinity. But that is not what we found; we found that linear Weertman requires a coefficient with much more variance than either of the two variables it seeks to relate, and this variance declines as $m$ increases.

Thus, we would argue that, rather than being agnostic about the value of the sliding exponent, inverse models actually provide
strong evidence in favor of nonlinear sliding. While it is true that we were able to fit the data almost as well with any value of $m$, having a good observational fit is only half the story. The amount of variance in our inverted coefficient required to fit the data dropped sharply as $m$ rose from 1 to 3, and dropped a little more as $m$ rose from 3 to 5. Our total variance measure, which combines both observational misfit and coefficient variance, was better for nonlinear sliding as compared to linear sliding by about a factor of 3 over the whole domain, or a factor of 2 when the analysis was limited to the more challenging fast-flow
region. In addition, our L-curves for nonlinear sliding were more sharply curved, producing a narrower and more well-defined range of acceptable $\lambda$ values.

This evidence from numerical inverse modeling is in agreement with other lines of evidence as well. Laboratory tests on actual samples of subglacial till consistently reveal coulomb plastic rheology for soft basal sediments (e.g. Kamb, 1991; Tulaczyk et al., 2000). Additionally, analytical and numerical process models for hard bed sliding with cavitation show coulomb-like
behavior with basal traction bounded by a maximum yield stress (Iken, 1981; Schoof, 2007; Gagliardini et al., 2007). Transient numerical modeling of the visco-elastic response of glaciers to tidal forcing suggests that nonlinear sliding is required to fit the response recorded with GPS observations (Gudmundsson, 2011). Inversions of time-dependent velocity and geometry data suggest that the observed acceleration of Pine Island Glacier is best fit by a sliding law with $m \geq 5$ (Gillet-Chaulet et al., 2016). Thus, we argue that the glaciological community actually has many converging lines of evidence indicating that the best
description of basal sliding is a nonlinear one.

Fortunately, our results suggest that the requirement that modelers use nonlinear sliding is not as numerically onerous as it would first appear. One reason for the popularity of linear Weertman sliding is doubtless that it is numerically easiest. Within our own inversions, we had to tighten the iteration stopping tolerance by an order of magnitude in order to produce a monotonic L-curve when moving from $m = 1$ to $m = 3$, and we had to tighten the tolerance by an additional order of magnitude when
moving to $m = 5$. However, while some of the evidence referenced above would suggest that $m \to \infty$ is appropriate, implying that modelers should always use the largest value of $m$ that they can, our results suggest that most of the benefits of nonlinear sliding can be realised by merely increasing $m$ from 1 to 3. This makes sense when considering $1/m$ rather than $m$ itself: $m$ has the range $[1, \infty)$, but $1/m$ has the more well-defined range $[0, 1]$. From this perspective, $m = 3$ has already moved 67% of the way to infinity, and $m = 5$ has moved 80% of the way there. The fact that most of the improvement in inversion performance
can be realized with relatively mild nonlinearity means that the transition to nonlinear sliding should be easy for modelers to make, and it also means that future modelers have no excuse for continuing to use linear Weertman sliding laws.

This is because the distinction between linear and nonlinear basal sliding is not merely a theoretical manner; when the basal boundary conditions constrained by an inversion are used to drive a transient model into the future, the nonlinearity of the sliding law also has a strong influence on our predictions of future ice sheet dynamics. It has long been known that the
diffusivity of ice sheet surface perturbations is proportional to $m$ (e.g. Nye, 1959, Eq. 18). This surface diffusivity controls the rate at which dynamic changes — for instance, grounding line retreat triggered by the intrusion of warm ocean waters into an



ice shelf cavity, the cause of potential ice sheet destabilization in the Amundsen Sea (Turner et al., 2017) — propagate into the interior of the ice sheet. In the limit of Coulomb plastic sliding throughout the ice sheet, this propagation is instantaneous, while for linear Weertman sliding, this propagation could take millennia. While it is tempting to therefore conclude that nonlinear sliding makes the ice sheet more unstable, the reality is more complex than this. With nonlinear sliding, the rapid inland propagation of dynamic thinning following grounding line retreat draws mass from the interior towards the margins more rapidly and in greater quantity than for linear sliding. One result of this is more mass loss and sea level rise at an equivalent amount of grounding line retreat; however, another result of this is that the increased mass flux from the interior towards the margins acts to reinforce the area of the ice sheet near the retreating grounding line, thus slowing the retreat (Barnes and Gudmundsson, 2022). The balance between these effects (slower grounding line retreat, but more sea level rise at an equivalent level of retreat) can result in either an increase or a decrease in sea level rise, potentially even having opposite sign for neighboring glaciers (Barnes and Gudmundsson, 2022). As making projections of future sea level rise is the single most societally important contribution that glaciologists can make, it is therefore vital that we use the right sliding law when making predictions of ice sheet evolution in a warming climate. Unfortunately, at least 5 of the 13 models that participated in the ISMIP6 projections for Antarctica used linear sliding relations (Seroussi et al., 2020).

## 5.2 Effective Pressure

While our inverse model results are strongly in favor of nonlinear basal sliding, they are more ambiguous on the utility of including effective pressure in a sliding rule. Other lines of evidence, including laboratory tests (Tulaczyk et al., 2000), field observations (Kamb et al., 1985; Andrews et al., 2014), and analytic and numerical process models (Iken, 1981; Schoof, 2007; Gagliardini et al., 2007), argue that basal water pressure plays an important role in regulating basal sliding. The importance of subglacial water in regulating basal sliding is by now the unambiguous consensus of the glaciological community (e.g. Cuffey and Paterson, 2010, Ch. 7) However, the question addressed by inverse models is a more subtle one: not whether basal water pressure plays a role in the real physics of ice sliding, which it undoubtedly does, but whether our models of basal water pressure are good enough to make $N$ a useful addition to a large-scale sliding law.

When it comes to simple geometry-based calculations for $N$, our answer is an unequivocal *no*: the slight reduction in inversion variance produced by these $N$ fields is insufficient to compensate for the increase in observational misfit, regardless of whether the analysis is performed for the whole grounded domain or for the fast-flow region alone. When it comes to the physically based $N_{\mathrm{CUAS}}$, however, our results are more ambiguous. When considered over the whole domain, using $N_{\mathrm{CUAS}}$ with linear sliding reduced total variance by 13%, but when considered in the more challenging fast-flow region alone, it increased total variance by 2%. Both of these changes are small compared with the large reductions in total variance achieved by using $m > 1$. Using $m > 1$ along with $N_{\mathrm{CUAS}}$ produced improvements when compared with either linear Weertman or linear Budd sliding, but not when compared with nonlinear Weertman at the same value of $m$ (Table 2). Thus, our results do not provide support for using $N$ in basal sliding laws- yet. If one is forced to use a linear sliding relationship, and one is mostly concerned with capturing the large-scale pattern of slow and fast flow, then our results suggest that one could improve one's model by including a physically based $N$ field like $N_{\mathrm{CUAS}}$. However, the resulting improvement is much less than the



improvement that one could get by moving from linear to nonlinear sliding, and if nonlinear sliding is combined with $N_{\mathrm{CUAS}}$, the improvement vanishes.

Nonetheless, there remains hope that further improvements in hydrological models may change the situation. $N_{\mathrm{CUAS}}$ was unambiguously better than either of the two geometry-based $N$ fields, demonstrating that the performance of inversions using
Budd sliding is sensitive to the quality of the $N$ field; it is thus reasonable to assume that further improvements in $N$ will improve inversion performance further. Given that we found that the performance of nonlinear Budd laws with $N_{\mathrm{CUAS}}$ was very close to the performance of nonlinear Weertman laws at the same value of $m$ (within 5–10%, measured in units of the total variance in the linear Weertman inversion, Table 2), it would not take much improvement for nonlinear Budd inversions to overtake nonlinear Weertman inversions. Furthermore, we know from the basic math of our sliding laws that there is a hard
limit to the performance of nonlinear Weertman inversions: the coefficient in a nonlinear Weertman sliding law must always have at least as much dynamic range as the dynamic range of basal drag, which for typical ice sheet settings is at least two orders of magnitude. By including $N$, Budd sliding laws open up the possibility that the variance of the coefficient can drop below this limit, and indeed we found that in our inversion with $N_{\mathrm{CUAS}}$ and $m = 5$, the variance of the coefficient was slightly less than the variance of basal drag in the fast-flow region. The total variance for this model was still higher than for Weertman
sliding with $m = 5$ (Table 2), but the fact that one of our Budd models was able to dip below this limit is a hopeful sign that future improvements in subglacial hydrology models may explain more of the variance in basal drag. Put simply, nonlinear Weertman may be better than nonlinear Budd right now, but it also has a much lower ceiling. With $m = 5$, nonlinear Weertman is already within 10–20% of the best possible performance it can achieve, whereas sliding laws that incorporate effective pressure can be as good (or as bad) as the hydrology model they use as input.

A key part of improving the use of hydrology models in glaciological inversions will likely be the development of improved procedures for ensuring internal consistency between the ice and water models. Hydrology models require melt forcing as input, and basal melt rates are highly dependent on shear heating. In fast flowing glaciers and ice streams, the heat released from frictional dissipation can exceed the geothermal heat flow by an order of magnitude or more. In our experiments with one-way coupling to CUAS, we estimated shear heating by smoothing the dot product of the observed surface velocity and
the gravitational driving stress. An improvement on this approach would be to use the velocity and stress fields computed by the inverse model to produce a better estimate of shear heating, and to use that estimate to force CUAS. It will be a challenge to develop a computationally efficient algorithm that can do this in a way that produces self-consistency without circularity. One possibility could be to iterate between the inverse model and the hydrology model at a fixed $\lambda$ value, and hope that the result converges to a single self-consistent solution. It would be computationally expensive to do this for each each individual
$\lambda$ sample in an L-curve, but this approach might be practical for refining the solution once $\lambda_{\mathrm{best}}$ has been constrained through an L-curve analysis with a fixed $N$ field. An alternate approach could be to do a full L-curve at each iteration and rerun the hydrology model using shear heating computed from the resulting $\lambda_{\mathrm{best}}$ model, but of course this approach may introduce some circularity into the argument, in that using $\lambda_{\mathrm{best}}$ to force CUAS in one iteration may bias the next L-curve towards keeping the same $\lambda_{\mathrm{best}}$ value. It will take careful thought by future modelers to figure out where the line between internal consistency and
circular reasoning is when coupling basal hydrology models with ice flow inverse models.

journal article preprint
en
Author(s) 2023. CC BY 4.0 License.



## 5.3 Full Stokes, Higher Order, and Prospects for Future Inversons

In almost all of our experiments in this paper, we used the SSA stress balance equations, which are the simplest approximation that can be used in an inversion. It is tempting to assume that inversions using more advanced stress balance equations, such as HO or Full Stokes (FS), are inherently more powerful than inversions performed with the SSA equations. However, that is forward model thinking, not inverse model thinking. From an inverse modeling perspective, the extra stress and strain rate components found in HO and FS models are a liability, not an asset. If a simpler SSA model is able to obtain a similar fit to the data as a more complex HO or FS model, then it is the simpler model that should be preferred. But "similar fit" may actually be an optimistic case for the more complex models: in our experiments comparing HO and SSA using the same mesh size and sliding law, we found that the SSA inversion was able to obtain a better fit to the data, using less inverted structure, than the HO model. There is thus very little justification, from our results at least, for using the more complex stress balance equations in an inversion. Yet it is still worth discussing the reasons why the more advanced equation performed poorly in our experiments, and what that means for the prospects of future improvements in ice sheet inversions.

The proximal cause of the poor performance of our HO inversions was that the basal ice rheology was too warm in large portions of the domain. Remedying this by using a constant cold rheology at all depths throughout the domain brought the inversion performance closer to the performance of the SSA inversion. However, a spatially invariant constant is a very poor approximation of the ice sheet thermal structure; it ignores very real variations in thermal boundary conditions such as surface temperature, accumulation rate, geothermal flux, ice thickness, and strain heating. While we probably could have produced a realistic colder temperature field that enabled our HO inversions to match the performance of our SSA inversions with a bit of work, the mere fact that this tuning is required is a mark against the HO inversions. In addition, using a stiffer ice rheology near the bed has the effect of greatly reducing shear deformation, forcing the entire domain to shift towards a regime of plug flow. In other words, in order to improve the performance of our HO inversion, we had to force it into a regime where it mimicked SSA! This explanation may also help to explain why some previous studies, such as Shapero et al. (2016), have found that the results of 3D inversions were insensitive to ice rheology. In that study, the authors performed FS inversions in restricted domains around three of the largest and fastest outlet glaciers in Greenland: Jakobshavn, Helheim, and Kangerdlugssuaq. While they did include a small amount of stagnant ice around the margins, the overwhelming majority of their three domains was composed of ice moving at least several hundred meters per year, which usually indicates that flow is dominated by basal slip, or at least, that they are unlikely to be in the regime where slip has dropped to zero and the sliding inversion is powerless to reduce velocity further. While Shapero et al. (2016) did not compare the performance of their FS inversion with an SSA inversion, our prediction would be that, if they had performed a comparative L-curve analysis, they probably would have found that their FS inversion had roughly the same performance as an SSA inversion.

Fundamentally, the problem is that, by allowing for vertical variations in rheology, strain rate, and therefore velocity, HO and FS introduce an enormous amount of variability into the problem that cannot be constrained by the available datasets. Sliding inversions typically take the rheological structure of the ice sheet to be fixed, but properly speaking, an FS or HO inversion should be simultaneously adjusting the sliding coefficient and the rheology of the lower ice column. While glaciological



inversions for vertically averaged rheology have been performed (e.g. Larour et al., 2005), we are not aware of any cases
       of glaciologists performing mechanical inversions for vertically resolved rheology. This is unsurprising, because the data we
       use to constrain ice sheet inverse models are essentially two dimensional: we have maps of the two components of horizontal
       velocity on a single vertical level of the ice sheet. If the rheology of the lower ice column is soft enough to permit a deformation
       velocity greater than the observed surface velocity, then an HO or FS inversion has no remedy to fix this by adjusting sliding,
and it will be incapable of fitting the data. Conversely, if the rheology of the lower ice column is too stiff to permit a rapid
       deformation velocity, then the ice sheet will move primarily by plug flow and the HO or FS inversion will merely recover
       the same performance that could have been achieved by an SSA inversion with far less internal complexity. In the absence of
       any constraint on variation in the third dimension, a two dimensional SSA model works best because it is a match to the two
       dimensional nature of the data.

It is interesting that we found that our SSA inversions even outperformed our HO inversions in slow-flowing areas of
       the ice sheet, where the SSA equations should not be a good approximation to the ice sheet stress balance. However, this
       is less surprising when considering the framework of the "hybrid" approximation to the stress balance equations (Schoof
       and Hindmarsh, 2010; Goldberg, 2011). In this approximation, shear deformation in the lower ice column is parameterized
       as a component of the sliding law, with ice velocity being decomposed into the sum of sliding and deformation flow, and
basal drag being determined by whichever process (slip or deformation) is weaker. Crucially, the vertically integrated stress
       balance equations of this approximation are identical to the SSA equations; the only difference is that the effective viscosity
       computation is modified to include the contribution of vertical shear to the effective strain rate, and the sliding law is modified
       to reflect the fact that resistance to flow could come from shear deformation in the lower ice column instead of sliding over
       the bed. The hybrid approximation applies throughout the ice sheet, not merely in fast-flowing streams and shelves moving by
plug flow, and because its stress balance equations are identical to those of SSA, an SSA inversion can therefore be regarded
       as representing a smooth transition between a sliding inversion in fast-flowing areas and an inversion for near-basal rheology
       in slow-flowing areas. Under this framework, our SSA inversion results for drag coefficient in slow-flowing regions could be
       reinterpreted as a measure of the integrated stiffness of the lower ice column rather than as a measure of sliding itself. But
       regardless of how we choose to interpret our results, the SSA equations themselves are agnostic as to whether the resistance
to horizontal flow comes from slip across the basal interface or from shear deformation a short distance above, and this very
       agnosticism is the key to their success at fitting surface velocity data, which cannot distinguish between sliding and deformation
       either.

           Nonetheless, we do not wish to give the impression that 3D models are useless in inversions. While 2D SSA equations
       may be superior for classic glaciological inversions of 2D velocity datasets, they will struggle to incorporate 3D constraints,
if those become available. In our opinion, the most likely source of 3D constraint in ice sheet stress balance inversions will
       be remotely sensed estimates of vertical velocity at the ice sheet surface. Altimetry measurements of surface height change
       ($dH/dt$) are routinely used in conjunction with firn densification and surface mass balance models in order to estimate ice
       sheet mass balance or the basal melt rate of floating shelves (e.g. Pritchard et al., 2012). This same data stream could easily be
       used to estimate the vertical component of velocity at the ice sheet surface. Indeed, using this data stream to estimate surface





vertical velocity would actually be easier than using it to estimate basal melting, since we do not need to perform corrections for advection or flux divergence in order to estimate surface vertical velocity. While this would still leave us with a two dimensional map of velocity at the ice sheet surface, at least we would then have access to all three components of velocity. The framework that we have introduced in this paper of normalizing misfit components by their characteristic scale before combining them would be invaluable in such an inversion, as horizontal velocity is typically several orders of magnitude larger than vertical 875 velocity.

An SSA model would likely struggle to fit a dataset of surface vertical velocity while simultaneously preserving the fit to observations of surface horizontal velocity. The reason is that an SSA inversion is required to produce a field of ice flux closely approximating the product of observed ice thickness and observed horizontal velocity. If the divergence of this flux field does not match observations of $dH/dt$, then the SSA model has nothing left with which to correct the mismatch. An 880 HO or FS inverse model, by contrast, can adjust ice velocity in the lower part of the ice sheet, giving it some freedom to change the vertically integrated flux divergence while only producing a small change in surface horizontal velocity. Thus, it may be possible for future inversions to leverage satellite altimetry measurements to learn more about the distribution of ice flow between basal slip and shear deformation. In addition, an inversion constrained by altimetry measurements provides a better starting point for projections of future sea level rise without the need for a drift correction, as such a model will also be 885 constrained to fit the present-day rate of mass change.

The other promising source of three dimensional constraints are observations using phase-sensitive Radio Echo Sounding (pRES) systems (e.g. Corr et al., 2002). Such systems use repeat radar observations at the same location to produce vertically resolved profiles of vertical velocity within the ice column. These measurements could provide a valuable compliment to the remotely sensed estimates of surface vertical velocity considered above: satellites can give a wide spatial distribution of 890 vertical velocity, but only at the surface, while pRES can give depth-resolved profiles of vertical velocity, but only at a select few locations. However, the limited spatial extent of pRES measurements is a major drawback. Inverse models work best when their data constraints are evenly distributed throughout the domain, but pRES data are sparse point measurements in the map-view plane. Since acquiring each pRES profile requires two costly *in situ* visits to the same location, we will probably never have a sufficiently widespread distribution of pRES measurements to include them in large-scale inversions such as those we 895 performed in this paper. Nonetheless, 3D stress balance inversions fitting pRES data could have an important role in high-resolution studies of select detail areas. If a targeted field campaign generates a local region with a relatively high density of pRES measurements, then an HO or FS inversion might be able to leverage those data to learn much more about the interior of the ice column than could be revealed using surface data alone.

Regardless of whether 3D inversions are performed in the classic context or incorporating constraints on vertical velocity, 900 they are likely to require the ability to adjust both basal drag and near-bed ice rheology. Procedures are needed to ensure that inverse models have a way of doing so that is both well-posed and physically realistic. For instance, it might be possible to partition the adjoint such that near-basal rheology is adjusted downward rather than the sliding coefficient when model velocity is higher than observed and sliding velocity is already very low. Another possibility might be to run a coupled thermo-mechanical inversion with an added cost function term penalizing basal slip where the basal temperature is below the melting




point. However, these are merely suggestions, and it will likely require careful work to produce a well-posed simultaneous inversion for both basal drag and vertically resolved rheology.

## 6  Best Practices

On the basis of the results and discussion above, we suggest the following principles for guiding the use and analysis of regularization in glaciological inversions:

– Whenever possible, modelers should use an *a priori* estimate of characteristic scale to normalize their individual cost function terms before those terms are combined. Doing so ensures that $\lambda$ values are unitless and of order unity, making it easy to identify the region of parameter space to explore in the L-curve analysis. In addition, normalization is invaluable if multiple data types or misfit metrics are included in $J_{\mathrm{obs}}$, or if multiple parameters are being solved for and therefore multiple regularization terms must be included in $J_{\mathrm{reg}}$.

– L-curves should always be presented on log/log axes with equal axis scaling (ie, one order of magnitude on the y-axis should be the same size as one order of magnitude on the x-axis). Given that the entire L-curve analysis is predicated on being able to identify a "corner", it is vital that the L-curve is presented in such a way that the visual impression of shape can be trusted. Linear/linear axes are not reliable for this purpose, as inverse power laws ($J_{\mathrm{obs}} \propto J_{\mathrm{reg}}^{-p}$ for $p > 0$) always appear "L"-shaped on linear axes despite having no intrinsically preferable value. A mix of linear and logarithmic axes
is, of course, completely meaningless.

– In addition to an honest visual presentation of L-curve shape, it is vital that the optimal $\lambda$ value be chosen through a quantitative measure of curvature. The specific curve-fitting procedure we presented here is, of course, not the only way to do this, nor do we even claim that it is necessarily optimal. However, it is vital that some sort of quantitative measure of curvature be used to identify the corner of the L-curve, rather than simple visual inspection. This curvature analysis
should be performed in log/log space rather than linear space.

– Inversions with noisy, uncurved, nonmonotonic, or otherwise ill-formed L-curves should be regarded as suspect. Rather than simply selecting one $\lambda$ value and moving on, modelers should view a poor L-curve as a sign that something needs to be fixed "under the hood": incomplete numerical convergence, suboptimal choice of solvers, inconsistent input datasets, excessively complex mask topology, a choice of $\lambda$ samples that does not include both limbs of the L-curve, or other
problems in the model set-up can all contribute to a poor L-curve.

– If performing a classic inversion- that is, fitting observations of surface horizontal velocity alone- then modelers should use the SSA equations first. If the modeler is performing an inversion because they need boundary conditions to force an HO or FS transient model, and they are not particularly concerned with the inversion results themselves, then an HO or FS inversion may be appropriate, especially if the transient experiment is formulated in terms of perturbations relative
to a control run. However, if the modeler is performing an HO or FS inversion for the purpose of analysing the inversion





results themselves, then they should perform a comparative L-curve analysis (as in our Fig. 13) to demonstrate that their 3D inversion outperforms a 2D SSA inversion. What they cannot do is to simply assume that the more complex equation is superior just because it is a more complete representation of the stress tensor; in an inverse modeling context, precisely the opposite may be true.

– The utility of sliding laws that incorporate effective pressure $N$ is dependent on the quality of the $N$ field used as input. Commonly used geometry-based calculations for $N$ add no value, producing a slight reduction in coefficient variance that is not sufficient to compensate for their slight increase in observational misfit. In contrast, we found that a physically based $N$ field added value when considered over the entire domain, although not when the analysis is restricted to the more challenging fast-flow region. Our recommendation is that glaciological modelers should only use sliding laws that incorporate $N$ if that $N$ field is based on the results of an actual hydrology model.


    – In contrast to the small marginal utility added by including $N$, moving from linear Weertman sliding to nonlinear sliding improves inversion performance substantially. Nonlinear sliding laws have more sharply cornered L-curves, a reduced range between $\lambda_{\max}$ and $\lambda_{\min}$, and up to a factor of three reduction in total inversion variance. Our recommendation is that glaciological modelers should always use nonlinear sliding if they are able.

**7   Conclusions**

Fundamentally, inverse models are a mathematical expression of principles of knowledge and inference. We have some aspects of the ice sheet that we can observe, such as the surface velocity, and we would like to use those observations to infer aspects that we cannot observe, such as the basal drag. However, we cannot infer all variables everywhere from a finite set of observations on a single surface of the ice sheet. It is therefore vital that we structure our inference engine to be skeptical of the inferred 955   structure that it generates. Regularization is the mathematical expression of that skepticism, ensuring that our inversions only produce structure that is actually required to fit the data.

    In this paper, we have given a tutorial on how to regularize glaciological inverse models, including normalization of cost function components and L-curve analysis. We have shown that the glaciological L-curve converges towards finite non-zero regularization values and a best knowable basal drag field in the continuous problem. We have shown how an L-curve analysis 960   can enable a modeler to draw more rigorous conclusions about the short-wavelength structure of basal drag, and we have shown how the optimal regularization level on coarse meshes is intimately connected with numerical convergence of spatial structure with respect to mesh size. We have also advocated a change in philosophy for glaciological inverse models that centers the role of regularization in the process, with the goal of inverse modeling being explicitly understood as not merely fitting the data, but fitting the data using the least amount of structure. We have shown how this shift in philosophy allows inverse models to break 965   their agnosticism on the question of sliding nonlinearity, coming down strongly on the side of nonlinear sliding laws while providing more ambiguous conclusions on the utility of incorporating effective pressure into the sliding law. This philosophy

also provides a framework for thinking about the relative performance of different types of inversions, with more complex models being required to justify their increased degrees of freedom with an improved observational fit.

But while this shift in philosophy may favor simpler 2D models for inversions of 2D datasets, we believe that a fascinating future of inverse modeling may lie in using 3D models to assimilate a greater variety of information than that which can be included in 2D models. Surface horizontal velocity is just one of many observations glaciologists have that give us information about the ice sheet state. The next step may be to include surface vertical velocities estimated from altimetry measurements; beyond that, we have a wealth of information derived from radar, such as pRES observations of depth-dependent vertical velocity, and RES measurements of internal layer geometry or basal reflection characteristics, not to mention geologic and geomorphic evidence of past ice sheet dynamics. Ultimately, we have a wealth of different data streams illuminating different aspects of ice sheet dynamics. The philosophy of fitting each data type according to its characteristic scale while simultaneously minimizing complexity in the model that does so provides the best framework for integrating the many sources of our knowledge into a coherent picture of the ice sheet.

*Code and data availability.* Inversion scripts, inverse model results, L-curve summaries, and our best combined drag estimate are available at DOI:10.5281/zenodo.7798650. The best combined drag estimate is presented interpolated onto a regular grid in netcdf format for ease of use. The ice flow model ISSM is open source and freely available at: https://issm.jpl.nasa.gov/ (Larour et al., 2012). Here ISSM version 4.18 was used.

*Author contributions.* MJW wrote inversion scripts, devised experimental design, made figures, and prepared the manuscript text. TK ran the CUAS hydrology model. AH and MR acquired funding and supervised the work. MR installed and maintained the ISSM installation on AWI's HPC system. All authors contributed to revising and improvements of the manuscript as well as discussions about ideas and conclusions.

*Competing interests.* The authors declare that we have no competing interests that would influence the conclusions drawn in this paper.

*Acknowledgements.* This work has been funded by AWI's INSPIRES project FRISio.



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
