# Peer review of "Regularization and L-curves in ice sheet inverse models, a case study in the Filchner-Ronne catchment"

_EGUsphere, 2023_

## Referee Comment (RC1)

**Comments on "Regularization and L-curves in ice sheet inverse models, a case study in the Filchner-Ronne catchment"**

**General comments**
Wolovick et al. proposes best-practices for achieving a well-balanced regularization in the context of basal drag inversion. Inversions are commonly used to initialize (and calibrate boundary conditions on) ice sheet models before running numerical experiments (e.g., transient simulations). Inversions based on minimizing a cost function (as is the case of this manuscript) is an ill-posed problem. Adding some regularization tends to make the minimization problem less ill-posed and converge to a global minimum. The type of regularization is somewhat subjective. The commonly employed one is the Tikhonov regularization of first order (as used in this manuscript), but many others have been proposed. The weight of the regularization term itself may be adjusted depending on the level of smoothness that the user wants to enforce. One way to determine this level of smoothness is to perform an L-curve analysis, which is frequently used in the glaciological community. In this paper, the authors performed several numerical experiments (inversions) to infer the friction coefficient varying some modeling choices, such as the mesh resolution, type of friction law, stress balance model etc.

Overall, the paper is well-written, the figures are well done and their captions are clear. However, the description of the numerical experiments could be written right in the introduction - it is hard to follow what exactly was done and why before digging into the methodology and results sections. Also, there are unnecessary technical details, discussions and speculations in several sections, which makes the text unnecessarily long. Most of them could be removed or moved to a supporting information file.

The biggest issue with the manuscript is the authors claiming they have "the best" recommendation for performing the friction coefficient inversion (by minimization). I would argue that it is simply impossible to validate any inversion without any observation of what we are trying to invert for. How can one claim that an inversion is better if there is no information about the "real" friction coefficient or basal drag? Affirming that they have the "best" inversion results (or knowledgeable friction coefficient, or Best Friction Map) only based on the L-curve shape alone is problematic. One could change the norm of the cost function (e.g., L1 instead of L2 - the norm used by the authors), the regularization term, the descent algorithm, line-search scheme, the inversion method, and find completely different recommendations based on the shape of the L-curve (with or not minimal values of cost functions), whatever "better" means here. Also, minimization problems have lots of local minima; so if the descent algorithm constantly achieves global minima, the results are biased, and not the "best".

There is not much to say about the methodology itself without comparing its results with a known friction coefficient and other methodologies. It would be better if the authors employed a simpler ice sheet domain, with a flat bed, and a *known* friction coefficient (e.g., based on the ISMIP-HOM setups for example) to compare with their inversion results. The resolution of BedMachine is 450 m. It means that any element of size h > 450 m (considering linear finite elements) will not catch the complete geometry (i.e., bed elevation), which adds another factor to the current results. Changing mesh resolution also changes

geometry (bed), so you are not solving the same mathematical problem (for any h > 450 m). Also, it is hard to say the regularization parameter \lambda goes to a nonzero number based on the extrapolation of a fitted curve (Figure 8b). Is there any guarantee of that? Could one have similar results if using another inversion scheme, descent algorithm, regularization term, cost function norm (L1, L2, Linf), cost function (log vel) or numerical setup? Additionally, affirming that the exponent m (in Weertman's friction law) should be >1 based on the shape of L-curve alone is simply impossible: can we really select the correct physics without direct observations of that, based only on one kind of ill-posed inversion methodology? A reasonable approach (but not the best) is matching the complexity of the model with the accuracy of the observed data (Smyl and Liu, 2019). But is it possible to be tested in a glaciological context considering the current lack of observed data below ice sheets?

With a simpler control model (synthetic model with a simple geometry and known friction coefficient) one could isolate and compare the effects of different practices and methodologies, regularization terms, cost function scaling, mesh resolution, errors in observed data, friction law, different wavelengths of friction coefficient, stress balance, ice temperature, etc, and then point out which approach achieves a reasonable result considering the known friction coefficient (see for example Smyl and Liu, 2019).

I do agree with the "fitting the data using the least amount of structure" message of the paper, but is it not already known by the glaciological community?

With that said, and based on the TC criteria (novelty, rigour, impact, presentation quality) it is not possible to support publication. However, I do feel the manuscript, after several changes to the text, removing unnecessary discussions and speculations and with focus on a single message, could be a good educational paper for beginners in ice sheet modeling (I am not sure if TC is the right journal). It brings some interesting approaches that could (but not necessarily guarantee) improve the way a corner in L-curve could be estimated, shows the impact of mesh resolution on the inversion (considering that bed elevation changes as practical mesh resolution changes), compares nonlinearly in friction laws, etc, which could help others to understand and run inversion problems in glaciology.

**Specific comments**

lines 8-12: For the abstract, I believe it should be a little bit clearer, mainly for a broad audience. The results or conclusions are written without any mention of the methodology - for instance, a hydrological model was used? So one can not catch the main results without reading the methodology part in the paper. A suggestion is to briefly describe the methodology, and then briefly take home messages. A shorter abstract in general is more readable than a long one.

line 13 velocity data. Do you mean ice surface velocities?

line 14 new constraints. Do you have any examples of possible constraints?

lines 29-30. Not sure if these studies show temporal variability of the bed. Do you mean basal drag or bed elevation?

line 34. continuous problem?

line 35-36. Note that this is true if the discretization of the basal drag is P1 over a mesh, for example. But one can model the basal drag using any other polynomial order, even piecewise constant. Maybe changes to "the number of free parameters is finite and depends on the discretization method (for example, in a grid….)", or something like that.

line 40-44 I believe this part "Regularization involves adding .. minimized in the inversion." could be shortened. There is "adding" and "combining", which basically is the same.

line 51. What is the meaning of non-convergent here? There is no an L-shape or corner?

line 52: solutions: solution of the PDE, or the inversion process (inverted field)?

line 52-53 the phrase "specifically, L-curve analyses are … Fourier coefficients of the solution" is not clear. For example, which "operator"? Is it performed a Fourier analysis? If the technical detail here does not add relevant information, maybe remove it. Stick to the point that in some problems (which ones?) the L-curve does not converge (what means converge?).

lines 56-65 The paragraph is what is expected to be (briefly) written in the abstract (please, see my comment about missing a few phrases about the methodology), so the reader can at a glance know what was done and the resulting conclusions.

line 80 "without the need to derive the adjoint". Maybe it is worth mentioning the meaning of adjoint here (some readers may not be familiar with).

line 81-82 "they did not rely on a series expansion representation of the drag coefficient in their derivation". Maybe it is worth adding after that phrase how the authors discretized the basal drag.

lines 90-99 I think that paragraph should be shortened. There is a discussion that is beyond the scope of "review". There is not need to say that one group did something in the second paper and not in the first. My suggestion is to stick to what is relevant of the current manuscript: there is not much detail of the regularization? Does the choice of regularization impact the basal drag near the grounding line? (note that the meaning of "grounding line" was not mentioned before - it is important for readers not familiar with specific terms of the cryosphere community);

lines 100-104. The same – the paragraph should be shortened.

lines 109-121. The same – the paragraph should be shortened.

line 138. The references "Rignot et al., 2017a" and "Rignot et al., 2017b" are the same.

line 150-151. Maybe it is worth mentioning that the operations ($\nabla$) are on the map-view plane. The same for $u = ux, uy$

line 164. There is no detailed mention about how the hydrology model is used in the inversions. It is used to compute the effective pressure? Maybe it is worth mentioning that a hydrological model is employed in the last paragraph of the introduction.

line 168. "of our model". Which one? Thermal model?

line 182. "by our model" => by the SSA model (or something similar)

line 182. "varies by about" => varies spatially (or on map view plane) by about

Figure 3c. What is hydraulic diameter (x-axis label of panel c)

line 194. "of the mask". Which one?

lines 195- 199. Maybe it is worth mentioning that err, hmin and hmax are Bamg parameters.

line 201. "anisotropic elements". Which anisotropy ratio? Note that anisotropic elements might introduce numerical errors.

line 202. "common diameter". What do you mean?

Line 203-205. "We analyze fast-flowing …to generate the weighting function". Maybe it is better to invert those phrases: explain first how you compute the weighting function, then the nominal cutoff.

line 206-2067. I am not sure what it means: "and also separating reliable velocity data from unreliable data". If it is not relevant, maybe cut that phrase.

line 214-215. "This technique is superior to simple interpolation". Well, I am not sure if it is superior, since you are performing more operations than a simple interpolation and smooth

things around. Maybe just mention that "this technique prevents aliasing …". Also, as said above, most of the details here could be moved to a supporting information file.

line 217-220 Does this smoothing procedure produce artifacts in terms of grounding line position? Any change (or parameterization) of the grounding line impacts the stress balance.

line 231-232 Equations (5) and (6). "B" here is the bed elevation, right? Note that you use the same letter for the rheological parameter.

lines 286-293. It is not clear how the $S_{reg}$ is obtained.

line 306. "inverse model to convergence". Which descent algorithm did you employ? LBFGS (M1QN3)? Which parameters to stop? Maybe it is worth mentioning it in the text.
line 309-324. I think most of the details here could be moved to supporting information. Maybe just say you applied a fitting curve to the sample points and the reason you did that (needs of a smooth 2nd derivative etc).

line 327. "curvature drops to 1/2 of its maximum value". Is 1/2 a heuristic value?

line 329-330 I think the phrase "and the equivalent uncertainty ratio on a linear scale is simply the exponential of that uncertainty" could be deleted.

lines 332-343. Entire paragraph. This is what I expected to read somewhere above, such that the reader could have a big picture of all the experiments you did. Not sure if moving it or writing something similar above would improve the manuscript. Nevertheless, my suggestion is to keep it as simple and clearer as possible, with few words.

line 350 "giving us confidence in our inversion procedure". I am note sure if achieving L-curve monotonicity will give you a confidence of the inversion itself (i.e., achieving global minima); there are other factors involved (eg. descent algorithm convergence criterion). What the monotonicity could say is that your algorithm is not introducing potential spurious or numerical errors, as you mentioned below. Maybe remove or rephrase that phrase.

line 362-363 - is it not the contrary?

Line 364-369 – "The power-law nature… that includes both limbs". I am not sure if this discussion should be here. Maybe remove it.

line 385-386. I believe it is important to mention you also performed a spatial and spectral analysis of the inversion results somewhere above (maybe in the introduction).

line 447-448. "These results emphasize the importance of choosing the correct regularization before using an inversion to draw conclusions about the short-wavelength structure in the ice sheet basal drag". I believe this phrase could be removed.

line 480. "Once mesh resolution is better than the ice thickness" better you mean "finer"? Maybe it is worth rephrasing it.

line 504. lambda_best, lambda_min (is it the opposite? lambda_best. lambda_max) decreasing resolution, increasing lambda

line 515 What exactly do you by "no geological variation in the substrate"?

line 624 Best Drag Map. Combining several inversions to produce a single basal drag map seems a reasonable approach; however, I would not say "best" map. Unfortunately, we do not have measures of the "real" basal drag to compare with. Better say a combined map (an averaged or mean map) that includes some modeling "uncertainties" or different physical parameters (note that uncertainties in data sets were not included - e.g., bed elevation, surface elevation, ice velocity etc.)

line 632-633 "Each inversion is weighted according to the inverse of its total variance ratio." Is there a reason to use that weight?

line 663 "singular values of the operator". Which operator?

line 675 by a mesh of that size => by a mesh of that resolution
As mesh size approaches zero => As element size approaches zero or As mesh resolution approaches infinity (or anything better)

line 726. "future modelers have no excuse for continuing to use linear Weertman sliding laws." It sounds like the whole community is using m=1 nowadays. I am not sure if this is really true, although you mentioned some modelers employed m=1 in the ISMIP6 effort. The most common is m=3 and new studies say that m>3 seems more appropriate. I would rephrase it appropriately.

line 731-732 - I believe the delimiters "—" in that phrase is misplaced. Please check it.

line 727-745. Note that in unconfined ice sheets, any change in the ice shelf will not propagate inland (Schoof, 2007). Probably the retreat of the grounding line is caused by loss of buttressing (which is caused by shelf thinning due to increased basal melting).
Also, the phrase: "the nonlinearity of the sliding law also has a strong influence on our predictions of future ice sheet dynamics". From Barnes and Gudmundsson (2022): "In broad terms, the system always reacts in the same way to a particular change in forcing, no matter which sliding law or parameter values are used. The magnitude and speed of these reactions vary, but the response is clearly bounded." That said, I suggest rephrasing or removing that discussion, which I think is unnecessary here.

line 785-800. Again, not sure if this whole discussion is really needed. Better shortening or removing it.

**Technical corrections**

line 62 vertically integrated model => vertically integrated ice sheet model

line 63 our various model results => maybe "our various inversion results"

line 84 Rather than introduce => Rather than introducing

line 116. components very inversely => components vary inversely

line 139. driving stress resolved in the flow direction => driving stress projected on the flow direction

line 149. check the equation

line 163 HO models => HO model

line 207. with mesh number => in all meshes (maybe?)

line 208. the model places => the mesh generator (or mesher or Bamg or our meshing strategy) places

line 212. all relevant grids => all relevant fields (I know the fields are defined on grids, but better saying fields (defined over equally spaced grids) )

line 224. please, un    bold subscript "b" in "\tau_b"

line 264 analyze structure in C => analyze the structure in C

line 265 under-penalize structure => under-penalize C (maybe?)

line 281. We get this guess => We compute this guess

line 284. driving stress resolved in the flow direction => driving stress projected on the flow direction

line 387. we tested differed by => we tested differ by

line 390: please, check if "\tau" should be bold here (same in lines 392, 411, and elsewhere)

line 431 driving stress and basal drag are locally balanced => driving stress and basal drag are virtually locally balanced (not sure if coherence in power spectra means exactly \tau_d = \tau_b)

line 467 represent mesh size => represent element size (and elsewhere in this subsection)

line 483 we can get an approximate = > we can estimate an approximate

Figure 8. "with solid diamonds marking λbest for each mesh." it is hard to see the diamonds. Maybe changing their colors to a single color? (e.g., black) (same for Figures 10, 11 and 13)

Table 1. "C_0" should it be C_W?

line 519. geometry-based fields => geometry-based N fields. Same for "all three fields" => all three N fields

line 559 Fig. 10a => Fig. 11a

line 560 Fig. 10c,e,g => Fig. 11c,e,g

line 625 descriptions of the ice sheet bed => descriptions of the ice sheet basal drag (or friction coefficient)

line 633. we use the version of total variance => we use the value of total variance

line 708. coulomb => Coulomb (and elsewhere)

line 962 mesh size => mesh resolution

**References**

Barnes, J. M. and Gudmundsson, G. H.: The predictive power of ice sheet models and the regional sensitivity of ice loss to basal sliding parameterisations: a case study of Pine Island and Thwaites glaciers, West Antarctica, *The Cryosphere,* 16, 4291–4304, https://doi.org/10.5194/tc-16-4291-2022, 2022.

Schoof, C. (2007), Ice sheet grounding line dynamics: Steady states, stability, and hysteresis, *J. Geophys. Res.*, 112, F03S28, doi:10.1029/2006JF000664.

Smyl, D., Liu, D. Less is often more: Applied inverse problems using hp-forward models, Journal of Computational Physics 399 (2019) 108949. https://www.sciencedirect.com/science/article/pii/S0021999119306540. doi:https://doi.org/10.1016/j.jcp.2019.108949.

---

## Referee Comment (RC2)

**General Comments**

Wolovick et al present a detailed study centred on the regularization of an ill-posed problem in glaciology, namely the estimation of a basal traction field from surface velocity data. As the authors say, many ice sheet modellers rely on these techniques and improved discussion is welcome. I approve of the choice to confront real-world data rather than only recover pre-defined fields. The numerical experiments are sound and the analysis thoughtful. The demonstrations of scaling the cost function are particularly useful.

The paper is quite long: I don't think much is added by the inclusion of Budd sliding experiments, which are not the pressure dependent laws of interest today. I recommend removing these sections (they could go in a supplement?)

This is a good paper and should be published on the basis of its results and the detailed exploration of interesting aspects of the inverse problem, but sometimes seems to insist that its methods are optimal without rigorous proof, while using language that seems quite acerbic when referring to the work of others. I think the authors should word their points more carefully. I don't think that will require much work or change the paper in a major scientific sense.

For example, starting in L680, "It is common in the inverse modeling literature to read some version of the **sentiment** [I object to this word] that we … cannot use inverse models to distinguish between different sliding laws.", and says that the "the purpose of inverse modeling,… is not merely to fit the data, but to fit the data using the least amount of structure". In my view Joughin is correct to say that even complete knowledge of $Tb$ alone at a single time provides no information on the relationship between $Tb$ and $u$: **it is not a sentiment and does not fundamentally misunderstand anything.** Note that Joughin 2010 and Joughin 2019 make use of data at multiple times to determine that $m$ is not 1, which you should cite in the paragraph around line 710. I do agree that having more structure in C than Tb is undesirable/unjustified, but it might still be the underlying truth. You are then claiming that the regularization makes more information available, but Tikhonov regularization is a reasonable bias towards coarser structure rather than anything fundamental or inevitable. At the same time, having finer resolution observations could permit more structure (in Tb, not C) to be determined and that could be desirable. In other words, fitting the data using the least structure sounds a decent aim given the ill-posed nature of the problem (but what does it mean 'to fit', when the misfit is suboptimal?), but equally, one could claim to seek the most structure given the information content of the data (accept a suboptimal misfit in the light of knowledge that the problem is ill-posed so that certain types of observational error will be amplified and so certain aspects of the solution will be dominated by error).

**Specific Comments**

**Abstract.**

L-curve analysis is not the only way to select the regularization parameter, and so it cannot be used to select the optimal regularization level in general. It can be used to the select the optimal parameter if one accepts the idea behind the L-curve, but an alternative approach, which has been explored at least once in glaciology (Martin and Monnier, 2013), is Morozov's discrepancy principle.

I'm not saying you need to review this in the abstract but rephrase to be clear that L curve analysis is heuristic (but as you say, should be done properly).

**Review**

L 85. Stopping an iterative optimization method 'early' can be a type of regularization, depending on the iterative method of course. See e.g Hansen 1994.( "the CG process has some inherent regularization effect where the number of iterations plays the role of the regularization parameter")

**Methods**

L148: Eq 1 is wrong – but I see the authors already note this in a TC comment.

L162 (3.3): This is reasonable way to estimate 'B-bar' (and I also find that you have to reduce shelf temperatures by about 10 C, even with advection, unless I take care with the bottom boundary condition in the shelf). However, I think you should note that the full inverse problem includes estimation of 'B-bar' because Glen's flow law depends on unknown thermal conditions and even then is not the whole of large-scale ice rheology (e.g damage in shear margins might be important) This is obviously a problem because it makes the inverse problem even more ill-posed (i.e underdetermined as well as ill-conditioned) , and so assumptions such as the one made here are needed.

L212. Is this method entirely invented by the authors or is there a reference? It seems strange to me to call velocity a 'grid', when it is 'data on a grid'. At any rate, I can't see from what you say how the technique works.

314. This paragraph can be confusing at first, because 'smoothing' can apply to both 'C(x,y)' and the L-curve. Perhaps introduce $J_{r,c}$ and $J_d, c$ first.

L325. I like that you explore the region around the optimal $\lambda$ but the bounds you choose are arbitrary and so should not be called 'minimum/maximum acceptable', and you don't 'bracket' the full range of the corner.

**Results**

L365 : A very interesting point.

L370: Why does the curvature of the inidividual components matter? This is a long paper, so perhaps this detail could be trimmed.

L450. These are good points, but I am not sure that the points about Vogel 1996 help argue them (Vogel calls a method convergent if the regularized solution tends to the exact solution as either the upper bound of the magnitude on error tends to zero, or the number of observations of a random 'white noise' variable tends to infinity, model node count is secondary). The real point is that the 'forward' models are approximations that neglect fine structure.

L470 'but in addition coarse meshes are also unable to fit the data as well as fine meshes' or indeed, approximate solutions solve the stress balance equations.

L478: 'e-folding mesh size': define.

L500 (while paragraph). This is not convincing to me: what is meant by approximate bounds?

L555- (nonlinear sliding section). This is for me the most interesting result. What happens if you take your m=1 coefficient fields ($C_1(\lambda)$) use the formula $C_3|u|^{\frac{1}{3}} = C_1|u|$ to work out the corresponding fields for $C_3(\lambda)$ and then compute $J_{reg}(\lambda)$? Do the points lie on top of the m = 3 L curve (so that the difference is all in the cost function analysis) or not (so there is an effect in the individual problems, presumably does to the way the regularization term affects the optimization)

L627; "We feel that it is more appropriate to produce a consensus view of drag, τb". Agreed.

**Discussion**

L862-905 – this is a long section for what is at the end of the day opinion/ speculation. I don't think it is wrong, just not part of the work that has been done here. Same regarding 970 onwards.

**References**

Martin, Nathan & Monnier, Jerome. (2013). Of the gradient accuracy in Full-Stokes ice flow model: basal slipperiness inference. The Cryosphere Discussions. 7. 3853-3897. 10.5194/tcd-7-3853-2013.

Hansen, P.C., 2007. Regularization tools version 4.0 for Matlab 7.3. *Numerical algorithms*, *46*, pp.189-194.

---

## Author Comment (AC2)

**Response to Reviewer 1**

Original reviewer comments are in black, our responses are in blue.

We would like to thank both of the reviewers for their thorough and thoughtful reviews. The tone and recommendations of the two reviews were quite different, with Reviewer 1 recommending rejection and Reviewer 2 recommending acceptance with minor revisions. In addition to disagreeing on their overall recommendations for publication, there were also several instances in which the reviewers gave conflicting recommendations for specific changes or revisions. For instance, Reviewer 1 recommended that we should have used synthetic examples for our inversion, while Reviewer 2 praised our decision to use real data. Nonetheless, we have attempted to construct two replies that harmonize the various recommendations, and we have also given our rebuttal to Reviewer 1's reasons for recommending against publication.

While the official instructions we received explicitly called for us not to prepare a revised manuscript at this stage, we found that it was easier to deconflict our responses to the two reviews through reference to a common revised text. As a result, our revised manuscript is already partially finished, and the remaining requested changes should be relatively minor. Not counting smaller wording changes and other minor corrections, the biggest changes to the new manuscript are:

1. In order to shorten the manuscript, we removed the paragraphs speculating on future 3D inversions (lines 868-906), the paragraph speculating on future coupled hydrology/drag inversions (lines 785-800), and the final two paragraphs of the discussion about nonlinear sliding (lines 716-745).
2. In order to further shorten the manuscript, we moved several parts of the methods from the main text to a supplemental section. In particular, the methods sections on thermal structure and rheology (Section 3.3), mesh generation (Section 3.4) and the description of our curve-fitting procedure (Section 3.7) have all been moved from the main text to a supplemental methods section.
3. In response to comments from both reviewers, we moved the section explaining our experimental design (Section 3.8) from the end of the Methods section to the beginning.
4. In response to comments from both reviewers, we will add a supplemental section explaining our multi-wavelength smoothing procedure for interpolating gridded data onto our mesh.
5. In response to a comment from Reviewer 1, we will add a second panel to Figure 14 showing the uncertainty about our best combined drag estimate. We have already computed this uncertainty field and it is included in our data release on Zenodo, but we did not visualize the uncertainty field in the first draft of the manuscript.
6. In response to a comment from Reviewer 1, we will add a supplemental figure showing the distribution of mesh element anisotropy.
7. In response to a comment from Reviewer 2, we will add a supplemental section showing the results of a test where we convert the drag coefficient from the linear Weertman inversion into a nonlinear sliding law, and then compare the resulting L-curve with the L-curve obtained for the nonlinear sliding law itself. We already performed preliminary

versions of this test on our mid-resolution meshes (Meshes 4 and 5), although we chose not to show those tests in the first manuscript. We will repeat the test on our highest resolution mesh (Mesh 1) for the revision.

General comments

Wolovick et al. proposes best-practices for achieving a well-balanced regularization in the context of basal drag inversion. Inversions are commonly used to initialize (and calibrate boundary conditions on) ice sheet models before running numerical experiments (e.g., transient simulations). Inversions based on minimizing a cost function (as is the case of this manuscript) is an ill-posed problem. Adding some regularization tends to make the minimization problem less ill-posed and converge to a global minimum. The type of regularization is somewhat subjective. The commonly employed one is the Tikhonov regularization of first order (as used in this manuscript), but many others have been proposed. The weight of the regularization term itself may be adjusted depending on the level of smoothness that the user wants to enforce. One way to determine this level of smoothness is to perform an L-curve analysis, which is frequently used in the glaciological community. In this paper, the authors performed several numerical experiments (inversions) to infer the friction coefficient varying some modeling choices, such as the mesh resolution, type of friction law, stress balance model etc.

Overall, the paper is well-written, the figures are well done and their captions are clear. However, the description of the numerical experiments could be written right in the introduction - it is hard to follow what exactly was done and why before digging into the methodology and results sections. Also, there are unnecessary technical details, discussions and speculations in several sections, which makes the text unnecessarily long. Most of them could be removed or moved to a supporting information file.

Thank you.  The suggestion to move the description of the numerical experiments closer to the start of the paper is a good one.  Reviewer 2 also suggested that we move this section to the beginning of the methods rather than the end.  We will move this section forward in the revision. We will also make an effort to shorten the manuscript, including moving some material to an appendix or supplemental material section.

The biggest issue with the manuscript is the authors claiming they have "the best" recommendation for performing the friction coefficient inversion (by minimization). I would argue that it is simply impossible to validate any inversion without any observation of what we are trying to invert for.

This manuscript is not about validation of the friction coefficient, and indeed, the friction coefficient is not observable; even if we somehow had direct access to the bed at a broad spatial scale, the observable variable would be basal drag, not the friction coefficient. Regularized inversions are performed precisely because the true friction coefficient is unknown,

and L-curves are a widespread (if imperfectly utilized) tool for guiding regularized inversions. Our Recommended Best Practices (Section 6) is meant to guide future inverse modelers in the use of L-curves with regularized inverse problems, not to claim that the particular friction coefficient fields that we found are the best possible. We will add caveats where appropriate to be clear that there are alternatives to Tikhonov regularization and to L-curve analysis, but as these methods remain quite common we feel that there is a need for a paper giving best practices for their use and exploring what they can teach us about the ice sheet base.

How can one claim that an inversion is better if there is no information about the "real" friction coefficient or basal drag?

This goes to the crucial point we are making, that it is not only the quality of observational fit that matters, but also the complexity of inverted structure that is required to obtain the fit. We defined what we mean by "best" or "better" at multiple points in the manuscript (for instance lines 668-673); the key point is that we can claim that one inversion is "better" than another if it fits the data using less structure. Occam's Razor, or the *a priori* assumption that, all else being equal, simplicity is preferable to complexity, can be used to distinguish the quality of alternative inversions even in the absence of information about the true field.

Affirming that they have the "best" inversion results (or knowledgeable friction coefficient, or Best Friction Map) only based on the L-curve shape alone is problematic. One could change the norm of the cost function (e.g., L1 instead of L2 - the norm used by the authors), the regularization term, the descent algorithm, line-search scheme, the inversion method, and find completely different recommendations based on the shape of the L-curve (with or not minimal values of cost functions), whatever "better" means here.

Our Recommended Best Practices (Section 6) are all independent of any particular choices of norm, solver, or numerical methods. These recommendations apply so long as the modeler using them makes the choice to use L-curve analysis to quantify the tradeoff between observational fit and inverted complexity. We agree that changing the norm used in our cost function or changing the form of the regularization term would probably change the specific results obtained in the inversion. The L2 norm that we used in this manuscript is by far the most common norm in glaciological inverse modeling, but it is not the only norm that could be used, and other aspects of the cost function could be changed as well (for instance, it is common to compute the L2 norm of relative observational misfit or the L2 norm of logarithmic velocity misfit in addition to the L2 norm of absolute misfit). While other methods exist for managing the tradeoff between observational fit and inverted complexity (Reviewer 2 mentioned Morozov's discrepancy principle, for example), L-curve analysis remains the most common method used for managing this tradeoff, and L-curve analysis can be used with other norms or other ways of constructing the cost function. Our Recommended Best Practices would not, in fact, change when other things are changed in the inversion method; good L-curves should still be monotonic and curved regardless of how they are produced, and normalizing the individual components of the cost function is still useful regardless of which norm is used to compute those components. There is therefore a need for a detailed analysis of the glaciological L-curve, along with a set of

best practices and recommendations for using L-curves in glaciological inversions. We have added a sentence to the start of the Recommended Best Practices section explicitly stating that our recommendations still apply if the L-curve is constructed in other ways (e.g., other norms or misfit metrics).

Or, to summarize the point in a pithy way: we are not claiming that L-curves are best, we are recommending the best way to do an L-curve.

Also, minimization problems have lots of local minima; so if the descent algorithm constantly achieves global minima, the results are biased, and not the "best".

Is it not the opposite? One wants the descent algorithm to achieve the global minimum, not local minima. The whole point of regularization is to turn an ill-posed problem with many local minima into a well-posed problem with a single global minimum.

There is not much to say about the methodology itself without comparing its results with a known friction coefficient and other methodologies. It would be better if the authors employed a simpler ice sheet domain, with a flat bed, and a known friction coefficient (e.g., based on the ISMIP-HOM setups for example) to compare with their inversion results. The resolution of BedMachine is 450 m. It means that any element of size h > 450 m (considering linear finite elements) will not catch the complete geometry (i.e., bed elevation), which adds another factor to the current results. Changing mesh resolution also changes geometry (bed), so you are not solving the same mathematical problem (for any h > 450 m).

This is true, changing the mesh resolution also changes the ice geometry, so that we are not solving quite the same problem. The multi-wavelength smoothing technique that we used to interpolate gridded data onto our mesh alleviates the most disruptive effects of this problem, by ensuring that every mesh node receives a data value that is representative of the local average over a region comparable to the local element size. Reviewer 2 also asked for more detail about this method, so we will add a supplementary section or appendix explaining how it works. Yet you are correct that even with this method, different mesh resolutions do still have different geometry data. However, the true resolution of BedMachine is substantially coarser than the grid resolution of 450 m, because the underlying radar data have a flight line spacing that is typically several kilometers or more (Morlighem et al., 2020). Our finest mesh has a median size in the fast-flow domain of less than 1 km, and our first 5 meshes have median sizes less than 3 km, so we feel that we have passed the point at which further mesh refinement would reveal substantially new features in the basal topography.

We will add a brief discussion of this point in the revised manuscript during the section on convergence.

Also, it is hard to say the regularization parameter \lambda goes to a nonzero number based on the extrapolation of a fitted curve (Figure 8b). Is there any guarantee of that? Could one have similar results if using another inversion scheme, descent algorithm, regularization

term, cost function norm (L1, L2, Linf), cost function (log vel) or numerical setup?

What other method could we use to address the question of convergence without using extrapolation? Even if we used a synthetic example, we would still not be able to run an inversion with an infinite number of mesh nodes. The only way to address the question of convergence without resorting to extrapolation would be from a theoretical math standpoint, but that would be a completely different paper from what we are writing here. In practical numerical terms, the question of convergence with respect to mesh resolution is always going to involve extrapolation. Additionally, as we mentioned above, our best meshes have a resolution that is finer than the typical flight line spacing of the radar data used to produce BedMachine, so we are well below the scale at which additional mesh refinements change the problem by revealing substantially different features in the bed topography. Our best meshes are also finer than the mean ice thickness in the domain (2 km). Thus, we do not think that there are any other important lengthscales in the problem that we have not resolved, and we feel comfortable extrapolating our results towards zero mesh size.

In the revised manuscript we will include a brief discussion of the above points as they relate to extrapolation in the section on convergence. We will also mention the existence of alternatives to the L2 norm in the introduction.

Additionally, affirming that the exponent m (in Weertman's friction law) should be >1 based on the shape of L-curve alone is simply impossible: can we really select the correct physics without direct observations of that, based only on one kind of ill-posed inversion methodology?

We did not assert m>1 on the basis of L-curve shape, rather, we asserted that m>1 on the basis of total variance ratio. Total variance ratio compares observational fit and the quantity of inverted structure in two different optimal inversions selected from two different L-curves. It is true that the shape of the two individual L-curves was used to select lambda_best for each one, but the comparison between m=1 and m>1 was based on variance ratio, not L-curve shape. We explained our basis for comparison in Sections 4.6 and 5.1, and we showed the total variance ratio that we used to make the comparison in Table 2.

That being said, we agree that it would be inappropriate to definitively conclude that m>1 on the basis of inversion results alone. We will soften our wording in Section 5.1 to emphasize that we are not claiming that inversion results are definitive proof of a particular sliding law. However, as we also argue in our response to Reviewer 2, we hold to our assertion that inversion results are not neutral between different sliding laws either. Inversion results are supporting evidence for m>1, somewhere in between definitive proof and complete neutrality. Taken by themselves, all that the inversion results definitively demonstrate is that m=1 is not a particularly *useful* parameterization of the ice sheet bed: linear Weertman sliding requires a coefficient with greater dynamic range than either sliding velocity or basal drag individually. This result is almost certainly robust to changes in inversion methodology and choice of cost function norm, because this result is consistent both with the high dynamic range found in linear Weertman coefficients

in other ice sheet inverse models and with the poor correlation between observed velocity and driving stress, which does not depend on any inverse modeling at all.

This robust result, that linear Weertman sliding laws require an extremely high dynamic range in their free parameter in order to obtain a good fit to surface velocity observations, is (supporting) evidence against the validity of linear Weertman sliding.  Occam's Razor suggests that, when comparing different parameterizations with a similar quality of observational fit, we ought to favor simpler representations of reality over more complex representations.  Linear Weertman sliding requires a coefficient with over 5 orders of magnitude of dynamic range in order to relate two parameters, basal sliding and basal drag, that individually have only 2-3 orders of magnitude of dynamic range.  As Reviewer 2 acknowledged, the high dynamic range of the coefficient is problematic for linear Weertman sliding.  Of course, if other lines of evidence indicated that linear sliding was the correct representation of ice sheet basal processes, then we would be forced to accept the high dynamic range of the coefficient as reality.  However, other lines of evidence (which we cited on lines 707-715) indicate the opposite, that the best description of ice sheet sliding is Coulomb (or Coulomb-like).  None of these lines of evidence are definitive on their own, but taken together, the glaciological community has a convergence of evidence from multiple sources which indicate that linear Weertman is a very bad representation of reality.

In the revised manuscript we will soften our wording on this point to make clear that inversion results are supporting evidence that m>1, not definitive evidence by themselves.  However, we stand by our assertion that inversion results are not neutral between different sliding laws either.

A reasonable approach (but not the best) is matching the complexity of the model with the accuracy of the observed data (Smyl and Liu, 2019). But is it possible to be tested in a glaciological context considering the current lack of observed data below ice sheets?

Matching the complexity of the model with the complexity of the observed data was our explanation for the good performance of SSA inversions as compared to HO inversions (Sections 4.7 and 5.3).  That being said, it is beyond the scope of our paper to evaluate alternate data sources below ice sheets.  Inverse models are commonly used to learn something about the base of the ice sheet by making inferences from surface data.  Our goal was to show how this particular method could be leveraged to learn as much as possible.

With a simpler control model (synthetic model with a simple geometry and known friction coefficient) one could isolate and compare the effects of different practices and methodologies, regularization terms, cost function scaling, mesh resolution, errors in observed data, friction law, different wavelengths of friction coefficient, stress balance, ice temperature, etc, and then point out which approach achieves a reasonable result considering the known friction coefficient (see for example Smyl and Liu, 2019).

In contrast to this suggestion, Reviewer 2 agreed with our choice to use real data rather than a synthetic example in our experiments.

We feel that using real data ensures that our methods are more easily adopted by the glaciological community, and also that our conclusions have practical relevance to glaciological inverse modeling as it is actually performed. While synthetic examples absolutely do have their place in numerical modeling, there are many practical difficulties that arise when modelers turn their attention to real data.

For instance, our recommendation that cost function components should be normalized by an estimate of their characteristic scale would be easy to dismiss if we had used synthetic examples in our experiments, as in a synthetic example the modeler always knows the characteristic scale of the inputs that they themselves have created. By contrast, by using real data we have demonstrated that cost function normalization works even when modelers must estimate the characteristic scale of unknown parameters using indirect means. Our experiments with mesh resolution are also complicated by the use of real data rather than synthetic data. When testing the effect of mesh resolution for real data, we are forced to confront the fact that finite element ice sheet models typically employ spatially variable mesh resolution, and the question of "convergence with respect to mesh resolution" is thus a more complex one than it would be in simplified synthetic domains. Because we employed real data from a domain that contains a variety of glaciological settings, our demonstrations are more powerful, and also more directly useful to the ongoing work of glaciological inverse modeling, than they would be if we relied on synthetic examples.

Additionally, using real data allows us to learn things about the real ice sheet base and about the convergence properties of the glaciological L-curve. We devote two subsections of the Results section to analyzing spatial and spectral structure of ice sheet basal drag. We spend two additional subsections analyzing sliding nonlinearity and the utility of different representations of effective pressure. These results are only obtainable because we used real data. Finally, the convergence characteristics of L-curves- specifically, whether the corner lambda value approaches zero or a nonzero limit- depends on the roughness characteristics of the true solution (Vogel, 1996). Therefore, we would not have been able to meaningfully explore the convergence properties of the glaciological L-curve without using real data.

At the end of our discussion section we include an itemized list of recommended best practices for ice sheet inverse modelers. These recommendations would be of little practical utility if they were based on synthetic examples rather than real data. By using real data from a region that contains a wide variety of ice dynamic settings, we have demonstrated that our best practices are robust to the myriad complexities and unexpected difficulties that arise when applying inverse models to real settings.

I do agree with the "fitting the data using the least amount of structure" message of the paper, but is it not already known by the glaciological community?

We agree that this message is widely known in principle, but as we attempted to show in our Review section, it is unfortunately not always followed in practice. The actual practice of

L-curve analysis in glaciological inverse modeling is highly inconsistent. Published L-curves are often noisy, nonmonotonic, or uncurved, and the corner is rarely selected on the basis of quantification of curvature. Often, L-curve analysis is left out of inverse modeling papers entirely, and when it is left out, it is usually not replaced with alternate methods of evaluating the tradeoff between observational fit and inverted structure. Additionally, the corollary of this "fitting the data with the least amount of structure" message- that inversion results could help discriminate between different sliding laws and effective pressure models- is not widely accepted by the glaciological community. Thus, we feel that our analysis and recommended best practices are both needed.

With that said, and based on the TC criteria (novelty, rigour, impact, presentation quality) it is not possible to support publication. However, I do feel the manuscript, after several changes to the text, removing unnecessary discussions and speculations and with focus on a single message, could be a good educational paper for beginners in ice sheet modeling (I am not sure if TC is the right journal). It brings some interesting approaches that could (but not necessarily guarantee) improve the way a corner in L-curve could be estimated, shows the impact of mesh resolution on the inversion (considering that bed elevation changes as practical mesh resolution changes), compares nonlinearly in friction laws, etc, which could help others to understand and run inversion problems in glaciology.

While we appreciate your forthright and honest criticism, we respectfully hope that our response and intended changes will convince the editor to approve publication in TC. We feel that several aspects of our manuscript are novel, including our method of normalizing cost function coefficients and estimating L-curve corner location, our investigation of the convergence properties of glaciological L-curves, our use of a combined measure of observational fit and inverted structure to discriminate between competing sliding laws or effective pressure models, and our use of comparative L-curve analysis to compare the performance of HO and SSA inversions. Additionally, our spatial and spectral analysis sections add to the literature on these particular glaciers and to the understanding of basal processes producing rib-shaped patterns in ice sheet basal drag. Finally, as we showed in our Review section, the actual practice of L-curve analysis in published glaciological inverse modeling is highly inconsistent, and thus we would argue that our tutorial and recommended best practices are not actually "beginner" material.

That being said, we agree that our manuscript is too long and contains unnecessary speculation in the Discussion. Reviewer 2 also felt that our manuscript could be shortened and tightened. If our paper is accepted for publication, we will remove some of the speculation in the revised manuscript, and we will also move some material to supplementary sections in order to shorten the main text. We outlined our plans for shortening the manuscript at the beginning of this response.

Specific comments

lines 8-12: For the abstract, I believe it should be a little bit clearer, mainly for a broad audience. The results or conclusions are written without any mention of the methodology - for instance, a hydrological model was used? So one can not catch the main results without reading the methodology part in the paper. A suggestion is to briefly describe the methodology, and then briefly take home messages. A shorter abstract in general is more readable than a long one.

There is a tension between including more information in the abstract and making the abstract shorter and more readable.  That being said, we will try to edit the abstract to make it both more informative and more concise.

line 13 velocity data. Do you mean ice surface velocities?

Yes.  We will clarify the wording in the revision.

line 14 new constraints. Do you have any examples of possible constraints?

We speculated on future possibilities for more advanced inversions with new constraints in the discussion (section 5.3).  However, adding this information here would conflict with the goal of having a short and easily readable abstract.  Additionally, Reviewer 2 suggested that we should remove the relevant part of the discussion in the interest of making the manuscript shorter, so in the revision we will remove the mention of new constraints in the abstract as well.

lines 29-30. Not sure if these studies show temporal variability of the bed. Do you mean basal drag or bed elevation?

We are referring to temporal variability in the target of the inversions, ie, basal drag.  We will clarify the wording in the revision.

line 34. continuous problem?

The equations of the forward problem (Eqs 1-2) sliding law (Eqs 3-4) and the equations defining the cost function of the inverse problem (Eqs 8-15) are all continuous equations.  Every aspect of ice sheet inverse problems is formulated in terms of continuous equations which we approximate using numerical discretization.  In this sentence and the next we are making a distinction between the underlying continuous inversion, which has an infinite number of free parameters, and specific numerical implementations of the problem, in which the number of free parameters is limited by the discretization chosen.  We will reword these sentences to improve clarity.

line 35-36. Note that this is true if the discretization of the basal drag is P1 over a mesh, for example. But one can model the basal drag using any other polynomial order, even

piecewise constant. Maybe changes to "the number of free parameters is finite and depends on the discretization method (for example, in a grid....)", or something like that.

We will change the wording here to accommodate different discretization choices.

line 40-44 1 believe this part "Regularization involves adding .. minimized in the inversion." could be shortened. There is "adding" and "combining", which basically is the same.

We will shorten and simplify this sentence.

line 51. What is the meaning of non-convergent here? There is no an L-shape or corner?

We discuss the meaning of convergent and non-convergent in greater detail later in sections 4.4 and 5.0. As the manuscript is already very long, we would rather not get into detail here in the introduction. That being said, we agree that these sentences are awkwardly worded, and Reviewer 2 also recommended that we change the wording here. We will rewrite these sentences in the revised manuscript.

line 52: solutions: solution of the PDE, or the inversion process (inverted field)?

In this case, it is referring to the inverted field. We can clarify the wording here.

line 52-53 the phrase "specifically, L-curve analyses are ... Fourier coefficients of the solution" is not clear. For example, which "operator"? Is it performed a Fourier analysis? If the technical detail here does not add relevant information, maybe remove it. Stick to the point that in some problems (which ones?) the L-curve does not converge (what means converge?).

The "operator" in this context would be the bed-to-surface transfer function (Gudmundsson, 2003). However, we agree that this technical detail is not necessary here in the introduction and we will remove it.

lines 56-65 The paragraph is what is expected to be (briefly) written in the abstract (please, see my comment about missing a few phrases about the methodology), so the reader can at a glance know what was done and the resulting conclusions.

We can add some of this information to the abstract.

line 80 "without the need to derive the adjoint". Maybe it is worth mentioning the meaning of adjoint here (some readers may not be familiar with).

We will add this information.

line 81-82 "they did not rely on a series expansion representation of the drag coefficient in

their derivation". Maybe it is worth adding after that phrase how the authors discretized the basal drag.

We can briefly add this information

lines 90-99 I think that paragraph should be shortened. There is a discussion that is beyond the scope of "review". There is not need to say that one group did something in the second paper and not in the first. My suggestion is to stick to what is relevant of the current manuscript: there is not much detail of the regularization? Does the choice of regularization impact the basal drag near the grounding line? (note that the meaning of "grounding line" was not mentioned before - it is important for readers not familiar with specific terms of the cryosphere community);

We will shorten this paragraph by removing the discussion about the later Morlighem paper (which we reference separately later in the review section).  The important thing for our discussion here is that the first paper drew conclusions about the detailed spatial structure of basal drag without providing any detail on regularization and without doing anything to demonstrate that their results were robust to the choice of regularization.

lines 100-104. The same — the paragraph should be shortened.

We agree, we will shorten this paragraph.

lines 109-121. The same — the paragraph should be shortened.

Good suggestion, we will shorten this paragraph.

line 138. The references "Rignot et al., 2017a" and "Rignot et al., 2017b" are the same.

We apologize that we overlooked this! We will fix these references.

line 150-151. Maybe it is worth mentioning that the operations ($\nabla$) are on the map-view plane. The same for $u = u_x, u_y$

We will add this information.

line 164. There is no detailed mention about how the hydrology model is used in the inversions. It is used to compute the effective pressure? Maybe it is worth mentioning that a hydrological model is employed in the last paragraph of the introduction.

We discuss how the hydrology model is used later on, in Section 3.5.  Following the suggestion of both reviewers, we are going to move the overview of our experimental procedure (Section 3.8) from the end of the methods to the beginning.

line 168. "of our model". Which one? Thermal model?

Yes, this refers to the thermal model.  We will clarify the wording here.

line 182. "by our model" => by the SSA model (or something similar)

We will clarify the wording here.

line 182. "varies by about" => varies spatially (or on map view plane) by about

We will change the wording here.

Figure 3c. What is hydraulic diameter (x-axis label of panel c)

We define hydraulic diameter on line 200 of the text.  We can mention this in the figure caption.

line 194. "of the mask". Which one?

This is the same reordered mask that we used for smoothing and interpolation.  The combined mask represents ocean, floating shelf, grounded sheet, and bare rock. In response to comments from both reviewers, we will add a short appendix or supplementary material explaining our multi-wavelength smoothing procedure, as well as our reordered mask that behaves in an ice dynamically reasonable manner under averaging and interpolation operations.  We will briefly describe this mask in a new supplemental section explaining how we interpolated gridded data onto our mesh.

lines 195- 199. Maybe it is worth mentioning that err, hmin and hmax are Bamg parameters.

We will add this information.

line 201. "anisotropic elements". Which anisotropy ratio? Note that anisotropic elements might introduce numerical errors.

The anisotropy is generally low, mostly less than 3.  We can add a figure showing the distribution of element anisotropy to the supplement.

line 202. "common diameter". What do you mean?

By "common diameter" we mean the regular diameter of a circle- the distance from one edge through the center to the other edge.  We called it the "common diameter" in this context to distinguish it from the hydraulic diameter.  Maybe we could rephrase this as the "regular diameter"?

Line 203-205. "We analyze fast-flowing ...to generate the weighting function". Maybe it is

better to invert those phrases: explain first how you compute the weighting function, then the nominal cutoff.

We will switch the order here.

line 206-2067. I am not sure what it means: "and also separating reliable velocity data from unreliable data". If it is not relevant, maybe cut that phrase.

INSAR-based velocity maps often contain stripes of correlated errors aligned with the satellite orbits with amplitudes on the order of 5-10 m/yr.  Unfortunately, the published error estimates often do not fully account for these correlated error stripes, which are nonetheless easily visible on visual inspection of the data products.  While more recent velocity products have less pronounced striping than older products, we still feel that velocity data below ~10 m/yr is not reliable, regardless of the published error estimates.

As it is not really productive for us to spend a paragraph here venting our spleen at our colleagues in the remote sensing field over their overly optimistic uncertainty estimates, we will simply remove this phrase.

line 214-215. "This technique is superior to simple interpolation". Well, I am not sure if it is superior, since you are performing more operations than a simple interpolation and smooth things around. Maybe just mention that "this technique prevents aliasing ...". Also, as said above, most of the details here could be moved to a supporting information file.

Reviewer 2 also requested elaboration on our multi-grid smoothing procedure.  We will add an appendix or supplemental material explaining how it works.

line 217-220 Does this smoothing procedure produce artifacts in terms of grounding line position? Any change (or parameterization) of the grounding line impacts the stress balance.

The benefit of using this particular mask reordering is that it always behaves in an ice dynamically reasonable manner under averaging and interpolation operations.  For instance, the mask order given in BedMachine can produce erroneous ice rises at the ice shelf front when subject to spatial smoothing, a change which has an enormous impact on the ice shelf force balance.  However, the mask order that we used here will always produce intermediate values that respect the force balance- for instance, interpolated values between the grounded ice sheet and the floating shelf are readily interpreted in terms of area percentage of grounded ice.  We will add a short appendix or supplemental material section explaining the benefit of mask reordering.

line 231-232 Equations (5) and (6). "B" here is the bed elevation, right? Note that you use the same letter for the rheological parameter.

True, we have reused B for two different variables. We will replace bed elevation with $z_b$ to remove ambiguity.

lines 286-293. It is not clear how the S_{reg} is obtained.

Sreg is obtained from the mean-square gradient of a sinusoid with an amplitude given by the standard deviation of kguess and a wavelength given by the mean ice thickness. We chose not to give each step of the derivation its own numbered equation because the manuscript was already very long, but we can elaborate further on this point in the revision.

line 306. "inverse model to convergence". Which descent algorithm did you employ? LBFGS (M1QN3)? Which parameters to stop? Maybe it is worth mentioning it in the text.

Correct, we used M1QN3. We can add this information to the text.

line 309-324. I think most of the details here could be moved to supporting information. Maybe just say you applied a fitting curve to the sample points and the reason you did that (needs of a smooth 2nd derivative etc).

This is a good idea to shorten the manuscript. We can move this information to a supplemental section.

line 327. "curvature drops to 1/2 of its maximum value". Is 1/2 a heuristic value?

Yes, this is a heuristic value. We will add a mention of that here, along with a brief discussion about how this affects the uncertainty in our estimated lambda_min and lambda_max values.

line 329-330 I think the phrase "and the equivalent uncertainty ratio on a linear scale is simply the exponential of that uncertainty" could be deleted.

We can remove this phrase.

lines 332-343. Entire paragraph. This is what I expected to read somewhere above, such that the reader could have a big picture of all the experiments you did. Not sure if moving it or writing something similar above would improve the manuscript. Nevertheless, my suggestion is to keep it as simple and clearer as possible, with few words.

Reviewer 2 also suggested that we move this section to an earlier position in the manuscript. We will put it at the beginning of the methods section.

line 350 "giving us confidence in our inversion procedure". I am note sure if achieving L-curve monotonicity will give you a confidence of the inversion itself (i.e., achieving global minima); there are other factors involved (eg. descent algorithm convergence criterion). What the monotonicity could say is that your algorithm is not introducing potential spurious

or numerical errors, as you mentioned below. Maybe remove or rephrase that phrase.

We can rephrase this to say that a smooth monotonic L-curve that runs close to the original model points gives us confidence in our curve-fitting procedure.

line 362-363 - is it not the contrary?

No, this statement is correct, the high-lambda limb is the part that is closer to vertical.  Or were you referring to the part that says "the cost function is dominated by observational misfit"?  We worded this poorly; what we should have said was, "the observational misfit is higher than the regularization misfit".

Line 364-369 — "The power-law nature... that includes both limbs". I am not sure if this discussion should be here. Maybe remove it.

We think that it is important to provide some context as to why previous authors may have found the results they did, especially if those results (an inability to find a corner of their L-curve in log-log space) conflict with ours.  Additionally, Reviewer 2 thought that this was "a very interesting point".  We are therefore going to leave this point in place.

line 385-386. I believe it is important to mention you also performed a spatial and spectral analysis of the inversion results somewhere above (maybe in the introduction).

We will add this to the introduction.

line 447-448. "These results emphasize the importance of choosing the correct regularization before using an inversion to draw conclusions about the short-wavelength structure in the ice sheet basal drag". I believe this phrase could be removed.

We will remove it.

line 480. "Once mesh resolution is better than the ice thickness" better you mean "finer"? Maybe it is worth rephrasing it.

Yes, by "better" we mean "finer".  We will rephrase this.

line 504. lambda_best, lambda_min (is it the opposite? lambda_best. lambda_max) decreasing resolution, increasing lambda

The order of lambda_best and lambda_min is not relevant in this context.  The sentence was, "... lambda_best and lambda_min represent the approximate boundary between spatial structure that has converged with respect to mesh resolution and spatial structure that has not converged."  We could perhaps rephrase this to, "... the approximate boundary between spatial

structure that has converged with respect to mesh resolution and spatial structure that has not converged lies somewhere in between lambda_best and lambda_min".

line 515 What exactly do you mean by "no geological variation in the substrate"?

We mean no variation in the material properties of the substrate. We will rephrase this to clarify our meaning.

line 624 Best Drag Map. Combining several inversions to produce a single basal drag map seems a reasonable approach; however, I would not say "best" map. Unfortunately, we do not have measures of the "real" basal drag to compare with. Better say a combined map (an averaged or mean map) that includes some modeling "uncertainties" or different physical parameters (note that uncertainties in data sets were not included - e.g., bed elevation, surface elevation, ice velocity etc.)

We included an uncertainty map in our data release, but we did not visualize it for the paper. We computed the uncertainty estimate by taking the weighted standard deviation of the models that contributed to the combined best estimate. We can add the uncertainty map as a second panel to Figure 14.

As far as the title "best" is concerned, this map is the best estimate that we can achieve given the results we have available to us. We think that the title "best" is appropriate in this context. No one reading this paper should be under the illusion that we can magically access the true basal drag; rather, it is clear from context that what we mean by "best" is, "the best that we can achieve". Additionally, in the Discussion (lines 668-677) we discuss the "best knowable basal drag field"; our best drag map is the closest approximation that we can achieve to this field. We can add wording here to clarify that "best" means "best that we can achieve".

line 632-633 "Each inversion is weighted according to the inverse of its total variance ratio."
Is there a reason to use that weight?

The total variance ratio represents a combined measure of both the quality of observational fit and the variance of the inverted coefficient. It is also independent of our choice of characteristic scales for the cost function components. We spend a considerable amount of time in Sections 4.5, 4.6, and 4.7 talking about this metric and using it to discriminate between the quality of different inversions.

line 663 "singular values of the operator". Which operator?

In the context of Vogel (1996), the "operator" they were referring to was the function that maps the (unknown) forcing field to the observations. In the glaciological context, the "operator" would therefore be the bed-to-surface transfer function (Gudmundsson, 2003). We will change the wording here to explicitly state that the "operator" in this context is the bed-to-surface transfer function.

line 675 by a mesh of that size => by a mesh of that resolution
As mesh size approaches zero => As element size approaches zero or As mesh resolution approaches infinity (or anything better)

We will rephrase this to remove ambiguity.

line 726. "future modelers have no excuse for continuing to use linear Weertman sliding laws." It sounds like the whole community is using m=1 nowadays. I am not sure if this is really true, although you mentioned some modelers employed m=1 in the ISMIP6 effort. The most common is m=3 and new studies say that m>3 seems more appropriate. I would rephrase it appropriately.

We did not mean to imply that the entire community uses m=1, only that m=1 remains relatively common.  We will be removing this paragraph in the revised manuscript.

line 731-732 - I believe the delimiters "—" in that phrase is misplaced. Please check it.

The delimiters are correctly placed, but our sentence is awkwardly worded and difficult to parse. We will be removing this paragraph in the revised manuscript.

line 727-745. Note that in unconfined ice sheets, any change in the ice shelf will not propagate inland (Schoof, 2007). Probably the retreat of the grounding line is caused by loss of buttressing (which is caused by shelf thinning due to increased basal melting).

Yes, the observed connection between shelf thinning and grounding line retreat is almost certainly connected to the loss of buttressing.  However, in the interest of shortening the manuscript we are going to remove this paragraph in the revision.

Also, the phrase: "the nonlinearity of the sliding law also has a strong influence on our predictions of future ice sheet dynamics". From Barnes and Gudmundsson (2022): "In broad terms, the system always reacts in the same way to a particular change in forcing, no matter which sliding law or parameter values are used. The magnitude and speed of these reactions vary, but the response is clearly bounded." That said, I suggest rephrasing or removing that discussion, which I think is unnecessary here.

We would argue that the "magnitude and speed" of the ice sheet sea level response is an important societally-relevant prediction that glaciologists need to get right.  That being said, we agree that our manuscript is already quite long and it is really not necessary for us to get into details about the dynamics here.  We will remove this paragraph in the revised manuscript.

line 785-800. Again, not sure if this whole discussion is really needed. Better shortening or removing it.

We will remove this paragraph in the revised manuscript.  Reviewer 2 also recommended that we remove lines 863-906 for similar reasons.

Technical corrections

line 62 vertically integrated model => vertically integrated ice sheet model

We will change this in the revision.

line 63 our various model results => maybe "our various inversion results"

We will change this in the revision.

line 84 Rather than introduce => Rather than introducing

We will change this in the revision.

line 116. components very inversely => components vary inversely

We will change this in the revision.

line 139. driving stress resolved in the flow direction => driving stress projected on the flow direction

We will change this in the revision.

line 149. check the equation

We have already made an author comment committed to fixing this in the revision.

line 163 HO models => HO model

We will change this in the revision.

line 207. with mesh number => in all meshes (maybe?)

Mesh number refers to our Meshes 1-10 (the x-axis of Fig 3b).

line 208. the model places => the mesh generator (or mesher or Bamg or our meshing strategy) places

We will change this in the revision.

line 212. all relevant grids => all relevant fields (I know the fields are defined on grids, but

better saying fields (defined over equally spaced grids) )

We will change this in the revision.

line 224. please, un bold subscript "b" in "\tau_b"

We will change this in the revision.

line 264 analyze structure in C => analyze the structure in C

We will change this in the revision.

line 265 under-penalize structure => under-penalize C (maybe?)

"Under-penalize spatial structure in C"

line 281. We get this guess => We compute this guess

We will change this in the revision.

line 284. driving stress resolved in the flow direction => driving stress projected on the flow Direction

We will change this in the revision.

line 387. we tested differed by => we tested differ by

We will change this in the revision.

line 390: please, check if "\tau" should be bold here (same in lines 392, 411, and elsewhere)

Tau should be bold here.  We will change this in the revision.

line 431 driving stress and basal drag are locally balanced => driving stress and basal drag are virtually locally balanced (not sure if coherence in power spectra means exactly \tau_d = \tau_b)

We will reword this to "almost fully balanced" (because our coherence values do not actually reach 100%).

line 467 represent mesh size => represent element size (and elsewhere in this subsection)

We will change this in the revision.

line 483 we can get an approximate = > we can estimate an approximate

We will change this in the revision.

Figure 8. "with solid diamonds marking Abest for each mesh." it is hard to see the diamonds. Maybe changing their colors to a single color? (e.g., black) (same for Figures 10, 11 and 13)

We will make the diamonds more visible in the revision.

Table 1. "C_0" should it be C_W?

Yes.  We will change this in the revision.

line 519. geometry-based fields => geometry-based N fields. Same for "all three fields" => all three N fields

We will change this in the revision.

line 559 Fig. 10a => Fig. 11a

Good catch.  We will change this in the revision.

line 560 Fig. 10c,e,g => Fig. 11c,e,g

We will change this in the revision also.

line 625 descriptions of the ice sheet bed => descriptions of the ice sheet basal drag (or friction coefficient)

We will change this in the revision.

line 633. we use the version of total variance => we use the value of total variance

We will change this in the revision.

line 708. coulomb => Coulomb (and elsewhere)

We will change this in the revision.

line 962 mesh size => mesh resolution

We will change this in the revision.

References

Barnes, J. M. and Gudmundsson, G. H.: The predictive power of ice sheet models and the regional sensitivity of ice loss to basal sliding parameterisations: a case study of Pine Island and Thwaites (glaciers, West Antarctica, The Cryosphere, 16, 4291-4304, https://doi.org/10.5194/tc-16-4291-2022, 2022.

Schoof, C. (2007), Ice sheet grounding line dynamics: Steady states, stability, and hysteresis, J. Geophys. Res., 112, F03S28, doi:10.1029/2006JF000664.

Smyl, D., Liu, D. Less is often more: Applied inverse problems using hp-forward models, Journal of Computational Physics 399 (2019) 108949. https://www.sciencedirect.com/science/article/pii/S0021999119306540. doi:https://doi.org/10.1016/j.jcp.2019.108949.

Our References

Gudmundsson, G. H.: Transmission of basal variability to a glacier surface, J. Geophys. Res. Solid Earth, 108, https://doi.org/10.1029/2002JB002107, 2003.

Morlighem, M., Rignot, E., Binder, T., Blankenship, D., Drews, R., Eagles, G., Eisen, O., Ferraccioli, F., Forsberg, R., Fretwell, P., Goel, V., Greenbaum, J. S., Gudmundsson, H., Guo, J., Helm, V., Hofstede, C., Howat, I., Humbert, A., Jokat, W., Karlsson, N. B., Lee, W. S., Matsuoka, K., Millan, R., Mouginot, J., Paden, J., Pattyn, F., Roberts, J., Rosier, S., Ruppel, A., Seroussi, H., Smith, E. C., Steinhage, D., Sun, B., Broeke, M. R. van den, Ommen, T. D. van, Wessem, M. van, and Young, D. A.: Deep glacial troughs and stabilizing ridges unveiled beneath the margins of the Antarctic ice sheet, Nat. Geosci., 13, 132–137, https://doi.org/10.1038/s41561-019-0510-8, 2020.

Vogel, C. R.: Non-convergence of the L-curve regularization parameter selection method, Inverse Probl., 12, 535, https://doi.org/10.1088/0266-5611/12/4/013, 1996.

---

## Author Comment (AC3)

**Response to Reviewer 2**

Original reviewer comments are in black, our responses are in blue.

We would like to thank both of the reviewers for their thorough and thoughtful reviews. The tone and recommendations of the two reviews were quite different, with Reviewer 1 recommending rejection and Reviewer 2 recommending acceptance with minor revisions. In addition to disagreeing on their overall recommendations for publication, there were also several instances in which the reviewers gave conflicting recommendations for specific changes or revisions. For instance, Reviewer 1 recommended that we should have used synthetic examples for our inversion, while Reviewer 2 praised our decision to use real data. Nonetheless, we have attempted to construct two replies that harmonize the various recommendations, and we have also given our rebuttal to Reviewer 1's reasons for recommending against publication.

While the official instructions we received explicitly called for us not to prepare a revised manuscript at this stage, we found that it was easier to deconflict our responses to the two reviews through reference to a common revised text. As a result, our revised manuscript is already partially finished, and the remaining requested changes should be relatively minor. Not counting smaller wording changes and other minor corrections, the biggest changes to the new manuscript are:
1. In order to shorten the manuscript, we removed the paragraphs speculating on future 3D inversions (lines 868-906), the paragraph speculating on future coupled hydrology/drag inversions (lines 785-800), and the final two paragraphs of the discussion about nonlinear sliding (lines 716-745).
2. In order to further shorten the manuscript, we moved several parts of the methods from the main text to a supplemental section. In particular, the methods sections on thermal structure and rheology (Section 3.3), mesh generation (Section 3.4) and the description of our curve-fitting procedure (Section 3.7) have all been moved from the main text to a supplemental methods section.
3. In response to comments from both reviewers, we moved the section explaining our experimental design (Section 3.8) from the end of the Methods section to the beginning.
4. In response to comments from both reviewers, we will add a supplemental section explaining our multi-wavelength smoothing procedure for interpolating gridded data onto our mesh.
5. In response to a comment from Reviewer 1, we will add a second panel to Figure 14 showing the uncertainty about our best combined drag estimate. We have already computed this uncertainty field and it is included in our data release on Zenodo, but we did not visualize the uncertainty field in the first draft of the manuscript.
6. In response to a comment from Reviewer 1, we will add a supplemental figure showing the distribution of mesh element anisotropy.
7. In response to a comment from Reviewer 2, we will add a supplemental section showing the results of a test where we convert the drag coefficient from the linear Weertman inversion into a nonlinear sliding law, and then compare the resulting L-curve with the L-curve obtained for the nonlinear sliding law itself. We already performed preliminary

versions of this test on our mid-resolution meshes (Meshes 4 and 5), although we chose not to show those tests in the first manuscript. We will repeat the test on our highest resolution mesh (Mesh 1) for the revision.

General Comments

Wolovick et al present a detailed study centered on the regularization of an ill-posed problem in glaciology, namely the estimation of a basal traction field from surface velocity data. As the authors say, many ice sheet modellers rely on these techniques and improved discussion is welcome. I approve of the choice to confront real-world data rather than only recover pre-defined fields. The numerical experiments are sound and the analysis thoughtful. The demonstrations of scaling the cost function are particularly useful.

The paper is quite long: I don't think much is added by the inclusion of Budd sliding experiments, which are not the pressure dependent laws of interest today. I recommend removing these sections (they could go in a supplement?)

While we agree that the manuscript is quite long and needs to be shortened, it is not necessarily true that Budd laws are no longer of interest in the glaciological community.  The Budd sliding law continues to be used in recent glaciological publications (e.g. Kazmierczak et al., 2022; Brondex et al., 2019; Barnes and Gudmundsson, 2022).  In addition, at least two of the models in ISMIP6 used Budd sliding laws (Seroussi et al., 2020).  Furthermore, the use of a Budd sliding law in the context of our experiments makes sense, as a Budd law is easily comparable to a Weertman law with the same stress exponent.  By using both Budd and Weertman sliding laws, we can set up experiments to directly test the influence of effective pressure with everything else in the inversion held constant.

This is a good paper and should be published on the basis of its results and the detailed exploration of interesting aspects of the inverse problem, but sometimes seems to insist that its methods are optimal without rigorous proof, while using language that seems quite acerbic when referring to the work of others. I think the authors should word their points more carefully. I don't think that will require much work or change the paper in a major scientific sense.

Thank you.  We agree that our wording was too harsh in places.  We will rewrite the relevant parts in the revised manuscript.

For example, starting in L680, "It is common in the inverse modeling literature to read some version of the sentiment [I object to this word] that we … cannot use inverse models to distinguish between different sliding laws.", and says that the "the purpose of inverse modeling,… is not merely to fit the data, but to fit the data using the least amount of structure". In my view Joughin is correct to say that even complete knowledge of Tb alone at a single time provides no information on the relationship between Tb and u: it is not a sentiment and does not fundamentally misunderstand anything. Note that Joughin 2010 and Joughin 2019 make use of

data at multiple times to determine that m is not 1, which you should cite in the paragraph around line 710. I do agree that having more structure in C than Tb is undesirable/unjustified, but it might still be the underlying truth.

Thank you for pointing out those references. We will add these citations to the paragraph beginning on line 707, and we will soften our criticism of Joughin (who has made many important contributions to the field of glaciological inverse modeling over the past two decades), as well as dropping the word "sentiment".

However, we hold to our assertion that inversion results at a single snapshot in time can be used to learn something about the form of the sliding law. As we discussed in our reply to Reviewer 1, Occam's Razor suggests that, when two competing parameterizations obtain a similar fit to the data, we should prefer the parameterization that requires less complexity to do so.

Of course, this is not definitive proof that the sliding law has any one particular form. As you rightly point out, it could be the underlying truth that slip and drag have a linear relationship, and the extreme dynamic range of C could just be the unfortunate reality. However, in order to conclude that this is so, we would need positive evidence from other sources that a linear relationship between slip and drag is the correct one. The inversion results themselves may not be definitive proof of nonlinear sliding, but they are not neutral between all sliding laws either. If you accept Occam's Razor as a useful organizing principle for guiding scientific inference, then the inversion results (even for a single time slice) are evidence in favor of nonlinear sliding. That evidence could certainly be overridden if other lines of evidence favored linear sliding, but, as we discussed on lines 707-715, other lines of evidence agree with the inversion results in favoring nonlinear sliding. The inversion results add to a convergence of multiple lines of evidence favoring nonlinear sliding laws. We will add the two references that you suggest to this discussion.

As promised in our reply to Reviewer 1, we will soften our wording in this section to make clear that inversion results are supporting evidence in favor of nonlinear sliding, not definitive proof. However, we do disagree with your (and Joughin's) statement that "even complete knowledge of Tb alone at a single time provides no information on the relationship between Tb and u". Our point here is that competing parameterizations can be judged on more than just the quality of their observational fit, they can also be judged on the complexity of the free parameter(s) needed to obtain that fit. The new advance we are proposing here is that the complexity of the inverted free parameter required to fit the data provides additional information that can be used to discriminate between competing parameterizations. It may not be definitive proof, but it is not "no information" either.

You are then claiming that the regularization makes more information available, but Tikhonov regularization is a reasonable bias towards coarser structure rather than anything fundamental or inevitable. At the same time, having finer resolution observations could permit more structure (in Tb, not C) to be determined and that could be desirable. In other words, fitting the data using

the least structure sounds a decent aim given the ill-posed nature of the problem (but what does it mean 'to fit', when the misfit is suboptimal?), but equally, one could claim to seek the most structure given the information content of the data (accept a suboptimal misfit in the light of knowledge that the problem is ill-posed so that certain types of observational error will be amplified and so certain aspects of the solution will be dominated by error).

This is a good point. Lambda_best provides the best trade-off between observational fit and inverted complexity according to L-curve analysis; this could be thought of as finding the minimum necessary structure to explain the observations. However, a modeler could reasonably choose to use a smaller value of lambda if they valued obtaining the highest possible resolution of the basal drag more than they valued minimizing complexity in the inverted field. Alternatively, a modeler could reasonably choose to use a larger value of lambda if they had little interest in detailed basal structure and only wanted to include the structure that is absolutely required to produce an approximately reasonable velocity field.

This is why we bracketed the range of acceptable lambda values with lambda_min and lambda_max, rather than giving lambda_best alone. As both reviewers pointed out, bracketing the range of acceptable lambda in this way involves the selection of an arbitrary "heuristic" threshold for L-curve curvature (we used a value of ½ the maximum curvature to define lambda_min and lambda_max). Nonetheless, the drop-off in total curvature below lambda_best is quite steep, so this threshold has little influence on selecting lambda_min (it has a bigger effect on the selection of lambda_max, which we will explicitly mention in the revision). Thus, modelers who are interested in obtaining the highest resolution picture possible can still use this method, and the L-curve method will give them confidence that lambda_min will allow them to find the highest-resolution picture of the ice sheet basal drag that can reliably be obtained from the given observations.

Specific Comments

Abstract.
L-curve analysis is not the only way to select the regularization parameter, and so it cannot be used to select the optimal regularization level in general. It can be used to the select the optimal parameter if one accepts the idea behind the L-curve, but an alternative approach, which has been explored at least once in glaciology (Martin and Monnier, 2013), is Morozov's discrepancy principle.I'm not saying you need to review this in the abstract but rephrase to be clear that L curve analysis is heuristic (but as you say, should be done properly).

Reviewer 1 also mentioned alternate choices that we could have made in quantifying misfit and structure, such as using the L1 norm instead of the L2 norm. The L2 norm is by far the most common in glaciological inversions, and L-curve analysis is the most common method for calibrating regularization, so we feel that our choices here are defensible. Nonetheless, we will modify the text to acknowledge that alternatives to both of these choices exist.

Review

L 85. Stopping an iterative optimization method 'early' can be a type of regularization, depending on the iterative method of course. See e.g Hansen 1994.( "the CG process has some inherent regularization effect where the number of iterations plays the role of the regularization parameter")

That is true, but the drawback of regularizing an inversion by stopping the iterative algorithm early is that you are then dependent on the specific form of your iterative method for your regularization. This contrasts to explicit regularization, where the modeler can directly write down an equation for their regularization term and say, "I am penalizing this thing". We will rephrase this part to emphasize the difference between implicit and explicit regularization.

Methods

L148: Eq 1 is wrong – but I see the authors already note this in a TC comment.

Yes, we will change this in the revision.

L162 (3.3): This is reasonable way to estimate 'B-bar' (and I also find that you have to reduce shelf temperatures by about 10 C, even with advection, unless I take care with the bottom boundary condition in the shelf). However, I think you should note that the full inverse problem includes estimation of 'B-bar' because Glen's flow law depends on unknown thermal conditions and even then is not the whole of large-scale ice rheology (e.g damage in shear margins might be important). This is obviously a problem because it makes the inverse problem even more ill-posed (i.e underdetermined as well as ill-conditioned) , and so assumptions such as the one made here are needed.

Note that we did not shift the entire shelf column temperature by 10 C, only the surface temperature. The shelf base remained fixed to the melting point, so the shift in the average shelf temperature was less than 10 C.

You are quite right that the full inverse problem (for both rheology and drag) is underdetermined, as there are two unknowns at every point in the ice sheet. We wanted to make the problem more tractable by fixing rheology and only solving for drag. As we alluded to in the introduction and discussion, some authors have used inverse methods to solve for ice column rheology as well. However, the paper was quite long already, so we wanted to limit ourselves to only solving for a single variable.

L212. Is this method entirely invented by the authors or is there a reference? It seems strange to me to call velocity a 'grid', when it is 'data on a grid'. At any rate, I can't see from what you say how the technique works.

This method is our own invention.  Reviewer 1 also requested elaboration on this point, especially regarding the behavior of the reordered mask under averaging and interpolation operations.  We can add a short appendix explaining how the method works.

As far as terminology goes, we do not think it is strange to refer to velocity as a grid; we use "velocity grid" as shorthand for "a velocity dataset which is available in gridded format".

314. This paragraph can be confusing at first, because 'smoothing' can apply to both 'C(x,y)' and the L-curve. Perhaps introduce $J_{r,c}$ and $J_{d,c}$ first.

We will introduce Jr,c and Jd,c in the previous paragraph and reword this section in order to avoid confusion between smoothing in physical space and smoothing of the L-curve.

L325. I like that you explore the region around the optimal $\lambda$ but the bounds you choose are arbitrary and so should not be called 'minimum/maximum acceptable', and you don't 'bracket' the full range of the corner.

That is true, the value of ½ of maximum curvature that we have chosen to bound the optimal lambda is arbitrary.  Reviewer 1 referred to this choice as a "heuristic".  In practice this will probably have a bigger influence on our ability to constrain lambda_max than on lambda_min.  The drop-off in curvature below lambda_best is quite steep (Fig 5d), so alternate values of this heuristic threshold (say, 0.25 x max or 0.75 x max) will produce only small changes in the location of lambda_min.  On the high side, however, alternate values of this heuristic threshold could potentially produce bigger changes in lambda_max.

We will edit the text to acknowledge that this threshold is an arbitrary choice, and to discuss the uncertainty that this introduces, especially in lambda_max.

Results

L365 : A very interesting point.

Thank you.  Reviewer 1 suggested that we remove this point.  However, we think that it is important to provide some context as to why previous authors may have found the results they did, especially if those results (an inability to find a corner of their L-curve in log-log space) conflict with ours.  We are therefore going to leave this point in place.

L370: Why does the curvature of the individual components matter? This is a long paper, so perhaps this detail could be trimmed.

We will remove this detail.

L450. These are good points, but I am not sure that the points about Vogel 1996 help argue them (Vogel calls a method convergent if the regularized solution tends to the exact solution as either the upper bound of the magnitude on error tends to zero, or the number of observations of a random 'white noise' variable tends to infinity, model node count is secondary). The real point is that the 'forward' models are approximations that neglect fine structure.

It is true that model node count is secondary in the (Vogel, 1996) definition, however, in practical terms, the requirement that the regularized solution tends to the exact solution is also a requirement that the number of model nodes tends to infinity, since mesh resolution must approach zero in order to resolve the exact (continuous) solution.  Furthermore, the requirement that the number of observations approaches infinity is also a requirement that the spatial resolution of the observations approaches zero, since mechanical glaciological inversions represent a single snapshot in time.  Thus, the only way to increase the number of data points is for the velocity observations to have finer resolution.  The Vogel definition of convergence applied to glaciological inversions can be summarized as, "in the limit that observational error and/or observational resolution approach zero, and in the limit that model resolution also approaches zero, then the inverted solution tends to the true solution".

Note that this definition of convergence or nonconvergence is not dependent on the forward models being a complete representation of the physics.  Even Full Stokes inversions can be nonconvergent in the Vogel sense if the bed-to-surface transfer function (the "operator" in Vogel's terms) is sufficiently strong at attenuating short wavelengths.  Conversely, SSA inversions could be Vogel-convergent if their transfer function only weakly attenuates short wavelengths.  The key factor for this type of convergence is not the accuracy of the physics of the forward model; indeed, the Vogel analysis takes it for granted that the forward model is a complete representation of the physics.  The key factor for this type of analysis is the roughness of the true solution in comparison to the attenuating effect of the operator.

We will rewrite and clarify this paragraph.  In addition, this paragraph represents discussion more than results, so we will move it from Section 4.4 to Section 5.0.

L470 'but in addition coarse meshes are also unable to fit the data as well as fine meshes' or indeed, approximate solutions solve the stress balance equations.

Indeed.

L478: 'e-folding mesh size': define.

We computed best-fit exponentials to the graphs of lambda value vs mesh size.  An exponential function can be written either as y=exp(kx), or y=exp(x/x0), with the e-folding length x0 related to the growth/decay parameter k by x0=1/k.  In this case, the independent variable "x" is mesh

resolution, so our x0 is an "e-folding mesh size".  We will rephrase this section in the revised manuscript to improve clarity.

L500 (whole paragraph). This is not convincing to me: what is meant by approximate bounds?

We use the phrase "approximate bounds" to indicate the fact that the misfit metrics shown in Figure 9 worsen gradually as lambda is reduced between lambda_best and lambda_min, rather than having a single sharp threshold beyond which the misfit suddenly increases.

The point of this paragraph is that an inversion using a coarse mesh, evaluated at its own lambda_best, will produce structure that is similar to the structure produced by an inversion using a fine mesh at the same lambda value (even though this is not the lambda_best of the fine mesh).  Thus, we can regard the inverted structure in the coarse mesh as having converged with respect to mesh resolution.  The misfit increases as lambda is reduced from lambda_best to lambda_min, and then continues increasing below that.  Thus, we can conclude that the space between lambda_best and lambda_min represents the approximate boundary between inversion results that have converged and results that have not converged.  This analysis is separate from the convergence of the L-curve analysis, as the L-curve analysis involves changing lambda_best as a function of mesh resolution and lambda is held constant in this comparison.

We admit that this paragraph is awkwardly written and difficult to understand.  We will rewrite it in the revision.

L555- (nonlinear sliding section). This is for me the most interesting result. What happens if you take your m=1 coefficient fields ($C_1$ ($\lambda$)) use the formula $C_3 |u|_3 = C_1 |u|$ to work out the corresponding fields for $C_3$ ($\lambda$) and then compute $J_{reg}$ ($\lambda$)? Do the points lie on top of the m = 3 L curve (so that the difference is all in the cost function analysis) or not (so there is an effect in the individual problems, presumably does to the way the regularization term affects the optimization)

This is a very interesting question.  During preparation of the initial manuscript we performed these tests with our mid-resolution meshes (4 and 5) but we chose not to include them.  We will run this test on our highest resolution mesh (1) and add this analysis in a supplemental results section.

L627; "We feel that it is more appropriate to produce a consensus view of drag, тb". Agreed.

Thank you.

Discussion

L862-905 – this is a long section for what is at the end of the day opinion/ speculation. I don't think it is wrong, just not part of the work that has been done here. Same regarding 970 onwards.

That is true, these two sections are our speculations about future work, rather than a discussion of the results we presented in this paper.  As the manuscript is already quite long, we can remove these sections in the revised version.

References
Martin, Nathan & Monnier, Jerome. (2013). Of the gradient accuracy in Full-Stokes ice flow model: basal slipperiness inference. The Cryosphere Discussions. 7. 3853-3897. 10.5194/tcd-7-3853-2013.
Hansen, P.C., 2007. Regularization tools version 4.0 for Matlab 7.3. Numerical algorithms, 46, pp.189-194.

Our References

Barnes, J. M. and Gudmundsson, G. H.: The predictive power of ice sheet models and the regional sensitivity of ice loss to basal sliding parameterisations: a case study of Pine Island and Thwaites glaciers, West Antarctica, The Cryosphere, 16, 4291–4304, https://doi.org/10.5194/tc-16-4291-2022, 2022.

Brondex, J., Gillet-Chaulet, F., and Gagliardini, O.: Sensitivity of centennial mass loss projections of the Amundsen basin to the friction law, The Cryosphere, 13, 177–195, https://doi.org/10.5194/tc-13-177-2019, 2019.

Kazmierczak, E., Sun, S., Coulon, V., and Pattyn, F.: Subglacial hydrology modulates basal sliding response of the Antarctic ice sheet to climate forcing, The Cryosphere, 16, 4537–4552, https://doi.org/10.5194/tc-16-4537-2022, 2022.

Seroussi, H., Nowicki, S., Payne, A. J., Goelzer, H., Lipscomb, W. H., Abe-Ouchi, A., Agosta, C., Albrecht, T., Asay-Davis, X., Barthel, A., Calov, R., Cullather, R., Dumas, C., Galton-Fenzi, B. K., Gladstone, R., Golledge, N. R., Gregory, J. M., Greve, R., Hattermann, T., Hoffman, M. J., Humbert, A., Huybrechts, P., Jourdain, N. C., Kleiner, T., Larour, E., Leguy, G. R., Lowry, D. P., Little, C. M., Morlighem, M., Pattyn, F., Pelle, T., Price, S. F., Quiquet, A., Reese, R., Schlegel, N.-J., Shepherd, A., Simon, E., Smith, R. S., Straneo, F., Sun, S., Trusel, L. D., Van Breedam, J., van de Wal, R. S. W., Winkelmann, R., Zhao, C., Zhang, T., and Zwinger, T.: ISMIP6 Antarctica: a multi-model ensemble of the Antarctic ice sheet evolution over the 21st century, The Cryosphere, 14, 3033–3070, https://doi.org/10.5194/tc-14-3033-2020, 2020.

Vogel, C. R.: Non-convergence of the L-curve regularization parameter selection method, Inverse Probl., 12, 535, https://doi.org/10.1088/0266-5611/12/4/013, 1996.